# Subspace Optimization for Large Language Models with Convergence Guarantees

Yutong He [1 2]   Pengrui Li [3]   Yipeng Hu [1]   Chuyan Chen [1]   Kun Yuan [1 4 5]

## Abstract

Subspace optimization algorithms, such as Ga-Lore (Zhao et al., 2024), have gained attention for pre-training and fine-tuning large language models (LLMs) due to their memory efficiency. However, their convergence guarantees remain unclear, particularly in stochastic settings. In this paper, we reveal that GaLore does not always converge to the optimal solution and provide an explicit counterexample to support this finding. We further explore the conditions under which GaLore achieves convergence, showing that it does so when either (i) a sufficiently large mini-batch size is used or (ii) the gradient noise is isotropic. More significantly, we introduce **GoLore** (**G**radient rand**o**m **Lo**w-**r**ank proj**e**ction), a novel variant of GaLore that provably converges in typical stochastic settings, even with standard batch sizes. Our convergence analysis extends naturally to other subspace optimization algorithms. Finally, we empirically validate our theoretical results and thoroughly test the proposed mechanisms. Codes are available at `https://github.com/pkumelon/Golore`.

## 1. Introduction

Large Language Models (LLMs) have demonstrated impressive performance across a variety of tasks, including language processing, planning, and coding. However, LLMs require substantial computational resources and memory due to their large model size and the extensive amounts of training data. Consequently, recent advancements in stochastic optimization have focused on developing memory-efficient strategies to pre-train or fine-tune LLMs with significantly reduced computing resources. Most approaches (Vyas et al., 2024; Ramesh et al., 2024; Luo et al., 2023; Liu et al., 2024; Bini et al., 2024; Hao et al., 2024; Zhao et al., 2024; Muhamed et al., 2024; Pan et al., 2024; Loeschcke et al., 2024; Hayou et al., 2024; Lialin et al., 2023; Han et al., 2024; Song et al., 2023) concentrate on reducing the memory of optimizer states, which are critical components of overall training memory. For instance, Adam (Kingma, 2014) and AdamW (Loshchilov, 2017) maintain first and second-order momentum terms for gradients as optimizer states, leading to significant memory overhead for large models.

Among the most popular memory-efficient fine-tuning algorithms is LoRA (Hu et al., 2021), which decreases the number of trainable parameters by employing low-rank model adapters. However, the low-rank constraint on weight updates can result in substantial performance degradation for tasks that require full-rank updates, particularly in the pre-training of LLMs. To address this issue, several LoRA variants have been proposed, including ReLoRA (Lialin et al., 2023) and SLTrain (Han et al., 2024). Recently, Ga-Lore (Zhao et al., 2024) has emerged as an effective solution, significantly reducing optimizer states by projecting full-parameter gradients into periodically recomputed subspaces. By retaining optimizer states in low-rank subspaces, GaLore can reduce memory usage by over 60%, enabling the pre-training of a 7B model on an NVIDIA RTX 4090 with 24GB of memory. In contrast, the vanilla 8-bit Adam without low-rank projection requires over 40GB of memory.

### 1.1. Fundamental open questions and main results

While GaLore's memory efficiency has been well established, its convergence guarantees remain unclear. This raises the following fundamental open question:

*Q1. Can GaLore converge to stationary solutions, under standard and mild assumptions?*

By *stationary solutions*, we refer to first-order stationary points $x \in \mathbb{R}^d$ such that $\nabla f(x) = 0$ for $f : \mathbb{R}^d \to \mathbb{R}$. By *standard and mild assumptions*, we refer to common conditions in non-convex smooth optimization, including lower boundedness, $L$-smoothness and unbiased stochastic gradients with bounded variances, as in Assumptions 1-3.

[1]Peking University [2]Zhongguancun Academy [3]Beihang University [4]AI for Science Institute, Beijing, China [5]National Engineering Laboratory for Big Data Analytics and Applications. Correspondence to: Kun Yuan <kunyuan@pku.edu.cn>.

*Proceedings of the 42$^{nd}$ International Conference on Machine Learning*, Vancouver, Canada. PMLR 267, 2025. Copyright 2025 by the author(s).

Contrary to expectations, our investigation reveals that Ga-Lore is **NOT** theoretically guaranteed to converge precisely with standard and mild assumptions. The intuition behind this finding is straightforward: GaLore projects the stochastic gradient matrix onto a low-rank subspace spanned by the top $r$ singular vectors obtained via Singular Value Decomposition (SVD), effectively capturing the dominant components of the stochastic gradient matrix. However, the stochastic gradient comprises two components: the true gradient and gradient noise. When the true gradient dominates, the SVD-identified subspace primarily captures the gradient component. *In contrast, as the algorithm approaches a local minimum so that the true gradient diminishes while noise persists, the greedy and biased SVD-derived subspace captures only the noise component, ultimately leading to non-convergence.* To validate this intuition, we construct a counter-example demonstrating that GaLore fails to converge to stationary solutions, see the illustration in Fig. 1. This leads us to a subsequent open question:

> *Q2. Under what additional assumptions can GaLore converge to stationary solutions?*

Based on the preceding discussion, we conclude that the SVD-identified subspace aligns well with the descent direction when the true gradient component dominates the gradient noise. This observation naturally leads to several additional assumptions under which GaLore can converge:

- **Noise-Free Assumption.** We theoretically establish that GaLore converges at a rate of $\mathcal{O}(1/T)$ in the deterministic and non-convex setting.

- **Large-Batch Assumption.** We theoretically demonstrate that GaLore converges at a rate of $\mathcal{O}(1/\sqrt{T})$ in the stochastic and non-convex setting, provided that the batch size is extremely large and increases with the number of iterations $T$, *e.g.*, a batch size of $\Theta(\sqrt{T})$.

We further investigate GaLore's convergence under the **Isotropic-Noise Assumption**, wherein the noise is assumed to be evenly distributed across all directions, mitigating the bias introduced by the top-$K$ selection.

However, none of the aforementioned assumptions apply to the practical pre-training and fine-tuning of LLMs, where gradient noise is not assumed isotropic (Zhu et al., 2018; HaoChen et al., 2021; Mori et al., 2022; Wu et al., 2022; Wang & Wu) and fixed batch sizes are commonly employed. This observation raises a fundamental open question:

> *Q3. Under what modifications can GaLore provably converge in LLM settings, where possibly anisotropic gradient noise presents and the batch size is constant?*

It is evident that GaLore's SVD-based projections cannot extract meaningful information from noise-dominant matrices. To address this issue, this paper proposes modifying the SVD projection to a **G**radient Rand**o**m **Lo**w-**R**ank projection, resulting in the **GoLore** algorithm. This random projection can effectively capture gradient information even when gradient noise predominates, allowing for convergence in the stochastic and non-convex setting with normal batch sizes. We establish that GoLore converges at a rate of $\mathcal{O}(1/\sqrt{T})$ under standard assumptions.

In our empirical experiments, we implement GaLore during the primary phases of pre-training or fine-tuning LLMs due to its efficacy in capturing the gradient component using SVD-based projection. In contrast, we employ GoLore in the final phase, leveraging its ability to extract the gradient component from noise-dominant stochastic gradients using random projection. This approach enhances performance compared to employing GaLore throughout all stages.

While our analysis primarily focuses on GaLore, it also has significant connections to other memory-efficient algorithms. We demonstrate that a ReLoRA-like implementation is equivalent to GaLore, which is more computational efficient with little additional memory overhead. Furthermore, our theoretical results can be easily adapted to sparse subspace descent algorithms with minimal effort.

**Contributions.** In summary, our contributions include:

- We find that GaLore is not theoretically guaranteed to converge to stationary solutions under Assumptions 1-3. The key insight is that GaLore's SVD projection is biased and greedy; it may completely lose the true gradient information when the gradient noise is anisotropic and dominates the true gradient. We validate the non-convergence of GaLore by providing an explicit counterexample. This addresses Question Q1.

- Inspired by the aforementioned insight, we propose different additional assumptions under which GaLore can provably converge to stationary solutions. Under the noise-free assumption, we establish that GaLore converges at a rate of $\mathcal{O}(1/T)$. Under the large-batch assumption or isotropic noise assumptions, we demonstrate that GaLore converges at a rate of $\mathcal{O}(1/\sqrt{T})$. This addresses Question Q2.

- When possibly anisotropic gradient noise persists and the batch size maintains constant, we modify GaLore's SVD projection to a random projection, resulting in GoLore that provably converges to stationary solutions at a rate of $\mathcal{O}(1/\sqrt{T})$. This addresses Question Q3.

- We present an equivalent yet more computationally efficient implementation of GaLore/GoLore, and extend our analysis to sparse subspace descent algorithms. We

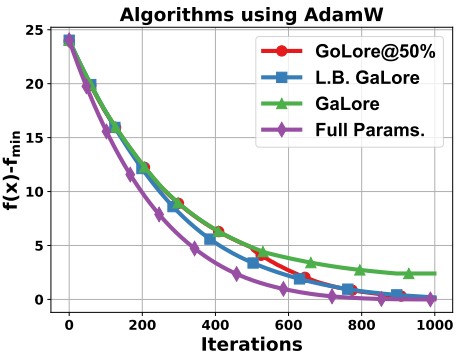 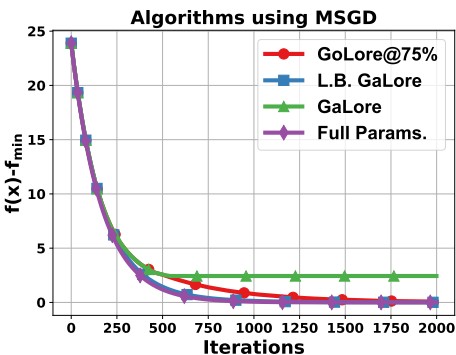

*Figure 1.* Loss curves of algorithms using AdamW (left) and Momentum SGD (right) on problem (1), where *L.B. GaLore* stands for large-batch GaLore, *GoLore@x%* applies GaLore for the beginning $(100 - x)\%$ iterations and GoLore for the last $x\%$ iterations.

conduct experiments across various tasks to validate our theoretical findings. Alternately using GaLore and GoLore in different phases achieves enhanced empirical performance in LLMs pre-training and fine-tuning.

### 1.2. Related work

**Memory-efficient training.** In LLM training, the primary memory consumption arises not only from the model parameters but also from activation values and optimizer states. Jiang et al. (2022) and Yu et al. (2024) have proposed methods to compress activation values into sparse vectors to alleviate memory usage. Other approaches primarily focus on reducing optimizer states. A notable work, LoRA (Hu et al., 2021) reparameterizes the weight matrix $\boldsymbol{W} \in \mathbb{R}^{m \times n}$ as $\boldsymbol{W} = \boldsymbol{W}_0 + \boldsymbol{BA}$, where $\boldsymbol{W}_0 \in \mathbb{R}^{m \times n}$ remains frozen as the pre-trained weights, and $\boldsymbol{B} \in \mathbb{R}^{m \times r}$ and $\boldsymbol{A} \in \mathbb{R}^{r \times n}$ are learnable low-rank adapters. Variants of LoRA, such as those proposed by Liu et al. (2024) and Hayou et al. (2024), aim to enhance training performance. However, constrained to low-rank updates, LoRA and its variants are primarily effective for fine-tuning tasks and struggle with pre-training tasks that require high-rank updates. To address this limitation, ReLoRA (Lialin et al., 2023) enables high-rank updates by accumulating multiple LoRA updates, while LISA (Pan et al., 2024) learns full-parameter updates on dynamically selected trainable layers. GaLore (Zhao et al., 2024) and FLORA (Hao et al., 2024) achieve high-rank updates by accumulating low-rank updates in periodically recomputed subspaces, and SLTrain (Han et al., 2024) employs additional sparse adapters for high-rank updates. SIFT (Song et al., 2023) also utilizes sparse updates. Although these algorithms have demonstrated comparable empirical performance to full-parameter training methods, theoretical guarantees regarding their convergence have not been established. Recently, Liang et al. (2024) proposes an online subspace decent algorithm with continuous-time convergence results; LDAdam (Robert et al., 2024) improves

GaLore by incorporating error-feedback; Fira (Chen et al., 2024a) and APOLLO (Zhu et al., 2024) adapt learning rates using optimizer states in the subspace.

**Convergence for lossy algorithms.** Many optimization algorithms utilize lossy compression on training dynamics, such as gradients, particularly in the realm of distributed optimization with communication compression. Researchers have established convergence properties for these algorithms based on either unbiased (Li et al., 2020; Li & Richtárik, 2021; Condat et al., 2024; Huang & Pu, 2023; He et al., 2024a;b; Mishchenko et al., 2019; Gorbunov et al., 2021; Alistarh et al., 2017; He et al., 2023) or contractive (Richtárik et al., 2021; Xie et al., 2020; Fatkhullin et al., 2024; He et al., 2023) compressibility. Kozak et al. (2019) provides a convergence analysis for subspace compression under Polyak-Lojasiewicz or convex conditions, where the subspace compression adheres contractive compressibility at each iteration. Despite these extensive findings, analyzing the convergence properties of subspace descent algorithms like GaLore remains challenging, as illustrated in Sec. 1.3.

### 1.3. Challenges in theoretical analysis

**Neither unbiased nor contractive compression.** Gradient projection onto the subspace can be viewed as gradient compression. Traditional analyses of optimization algorithms with lossy compression typically rely on either unbiased compressibility, *i.e.*, the compressor $\mathcal{C}$ satisfies

$$\mathbb{E}[\mathcal{C}(\boldsymbol{x})] = \boldsymbol{x}, \quad \mathbb{E}[\|\mathcal{C}(\boldsymbol{x}) - \boldsymbol{x}\|_2^2] \le \omega \|\boldsymbol{x}\|_2^2, \quad \forall \boldsymbol{x} \in \mathbb{R}^d,$$

for some $\omega \ge 0$, or contractive compressibility, *i.e.*,

$$\mathbb{E}[\|\mathcal{C}(\boldsymbol{x}) - \boldsymbol{x}\|_2^2] \le (1 - \delta)\|\boldsymbol{x}\|_2^2, \quad \forall \boldsymbol{x} \in \mathbb{R}^d,$$

for some $\delta \in (0, 1]$. However, GaLore's subspace compression is neither unbiased nor contractive due to the reuse of projection matrices. For example, consider a precomputed projection matrix $\boldsymbol{P} \in \mathbb{R}^{m \times r}$. There exists a

full-parameter gradient $\boldsymbol{G} \in \mathbb{R}^{m \times n}$ such that $\boldsymbol{G} \neq 0$ and $\mathcal{C}(\boldsymbol{G}) := \boldsymbol{P}\boldsymbol{P}^\top \boldsymbol{G} = 0$, violating both compressibilities.

**Periodically projected optimizer states.** When GaLore changes the subspace, the retained momentum terms must be adjusted to track the gradients in the new subspace. Since these momentum terms were initially aligned with the gradients in the original subspace, such adjustments inevitably introduce additional errors, especially when the two subspaces differ significantly. In the extreme case where the two subspaces are entirely orthogonal, the momentum from the previous subspace becomes largely irrelevant for optimization in the new one.

## 2. Preliminaries and assumptions

**Full-parameter training.** Neural network training can be formulated as the following optimization problem:

$$\min_{\boldsymbol{x}} f(\boldsymbol{x}) := \mathbb{E}_{\xi \sim \mathcal{D}} F(\boldsymbol{x}; \xi).$$

Here, $\boldsymbol{x} = (\text{vec}(\boldsymbol{X}_1)^\top, \cdots, \text{vec}(\boldsymbol{X}_{N_L})^\top)^\top$ collects all trainable parameters, $N_L$ is the number of layers, $\boldsymbol{X}_\ell \in \mathbb{R}^{m_\ell \times n_\ell}$ denotes the weight matrix in the $\ell$-th layer. $F(\boldsymbol{x}; \xi)$ computes the loss with respective to data point $\xi$, $\mathcal{D}$ denotes the training data distribution. In full-parameter training, we directly apply the optimizer to the full-parameter $\boldsymbol{x}$:

$$\boldsymbol{G}_\ell^{(t)} = \nabla_\ell F(\boldsymbol{x}^{(t)}; \xi^{(t)}),$$
$$\boldsymbol{X}_\ell^{(t+1)} = \boldsymbol{X}_\ell^{(t)} + \rho_\ell^{(t)}(\boldsymbol{G}_\ell^{(t)}), \quad \ell = 1, \cdots, N_L;$$

where $\nabla_\ell$ computes the gradient with respective to the $\ell$-th weight matrix $\boldsymbol{X}_\ell$, superscript $(t)$ denotes the variable in the $t$-th iteration, and $\rho_\ell^{(t)}$ is an entry-wise stateful gradient operator, such as Adam or Momentum SGD (MSGD). Specifically, using MSGD leads to the following $\rho_\ell^{(t)}(\cdot)$:

$$\boldsymbol{M}_\ell^{(t)} = (1 - \beta_1)\boldsymbol{M}_\ell^{(t-1)} + \beta_1 \boldsymbol{G}_\ell^{(t)};$$
$$\rho_\ell^{(t)}(\boldsymbol{G}_\ell^{(t)}) = -\eta \boldsymbol{M}_\ell^{(t)};$$

where $\eta$ is the learning rate, $\beta_1 \in (0, 1]$ is the momentum coefficient, and $\boldsymbol{M}_\ell^{(t)}$ is the momentum state. In full-parameter LLMs pre-training/fine-tuning, the memory requirements for storing momentum in MSGD and the additional variance state in Adam are highly demanding. According to Zhao et al. (2024), pre-training LLaMA 7B with a single batch size requires 58 GB of memory, with 42 GB allocated to Adam optimizer states and weight gradients.

**GaLore algorithm.** To address the memory challenge, (Zhao et al., 2024) proposes a novel Gradient Low-Rank Projection (GaLore) approach that allows much more memory-efficient full-parameter learning. The key idea is to project each stochastic gradient $\boldsymbol{G}_\ell \in \mathbb{R}^{m_\ell \times n_\ell}$ onto a low-rank subspace, yielding a low-dimensional gradient approximation.

Specifically, GaLore performs SVD on $\boldsymbol{G}_\ell^{(t)} = \boldsymbol{U}\boldsymbol{\Sigma}\boldsymbol{V}^\top$ and obtains rank-$r_\ell$ projection matrices $\boldsymbol{P}_\ell^{(t)} = \boldsymbol{U}[:, :r_\ell] \in \mathbb{R}^{m_\ell \times r_\ell}$ and $\boldsymbol{Q}_\ell^{(t)} = \boldsymbol{V}[:, :r_\ell] \in \mathbb{R}^{n_\ell \times r_\ell}$, where $[:, :r]$ denotes the selection of the matrix's first $r$ columns. When $m_\ell \leq n_\ell$, GaLore projects $\boldsymbol{G}_\ell$ onto $\boldsymbol{P}_\ell$, yielding a low-rank gradient representation $(\boldsymbol{P}_\ell^{(t)})^\top \boldsymbol{G}_\ell^{(t)} \in \mathbb{R}^{r_\ell \times n_\ell}$. Conversely, when $m_\ell > n_\ell$, GaLore projects $\boldsymbol{G}_\ell$ onto $\boldsymbol{Q}_\ell$, resulting in $\boldsymbol{G}_\ell^{(t)}\boldsymbol{Q}_\ell^{(t)} \in \mathbb{R}^{m_\ell \times r_\ell}$. In either scenarios, the memory cost of optimizer states associated with these low-rank representations can be significantly reduced, leading to memory-efficient LLMs pre-training or fine-tuning:

$$\boldsymbol{X}_\ell^{(t+1)} = \begin{cases} \boldsymbol{X}_\ell^{(t)} + \boldsymbol{P}_\ell^{(t)}\rho_\ell^{(t)}((\boldsymbol{P}_\ell^{(t)})^\top \boldsymbol{G}_\ell^{(t)}), & \text{if } m_\ell \leq n_\ell; \\ \boldsymbol{X}_\ell^{(t)} + \rho_\ell^{(t)}(\boldsymbol{G}_\ell^{(t)}\boldsymbol{Q}_\ell^{(t)})(\boldsymbol{Q}_\ell^{(t)})^\top, & \text{if } m_\ell > n_\ell. \end{cases}$$

Typically, GaLore selects $\rho_\ell(\cdot)$ as the Adam gradient operator, as illustrated in Alg. 1. However, GaLore can also choose $\rho_\ell(\cdot)$ to be gradient operators in either vanilla SGD or MSGD. Since SVD is computationally expensive, GaLore updates $\boldsymbol{P}_\ell^{(t)}$ or $\boldsymbol{Q}_\ell^{(t)}$ periodically. In other words, GaLore computes $\boldsymbol{P}_\ell^{(t)}$ or $\boldsymbol{Q}_\ell^{(t)}$ when iteration step $t \equiv 0 \pmod{\tau}$ where $\tau > 0$ is the period, otherwise $\boldsymbol{P}_\ell^{(t)} = \boldsymbol{P}_\ell^{(t-1)}$ and $\boldsymbol{Q}_\ell^{(t)} = \boldsymbol{Q}_\ell^{(t-1)}$ remain unchanged. Both the gradient subspace projection and periodic switches between different low-rank subspaces pose significant challenges to the convergence analysis for GaLore-like algorithms.

**Stiefel manifold.** Stiefel manifold is the set of low-rank projection matrices to use in subspace optimization. An $m \times r$ Stiefel manifold $(r \leq m)$ is defined as

$$\text{St}_{m,r} = \{\boldsymbol{P} \in \mathbb{R}^{m \times r} \mid \boldsymbol{P}^\top \boldsymbol{P} = I_r\}.$$

In GaLore, we have $\boldsymbol{P}_\ell^{(t)} \in \text{St}_{m_\ell, r_\ell}$ and $\boldsymbol{Q}_\ell^{(t)} \in \text{St}_{n_\ell, r_\ell}$.

**Basic assumptions.** We introduce the basic assumptions used throughout our theoretical analysis. Each of these assumptions is standard for stochastic optimization.

**Assumption 1** (Lower boundedness). The objective function $f : \mathbb{R}^d \to \mathbb{R}$ satisfies $\inf_{\boldsymbol{x} \in \mathbb{R}^d} f(\boldsymbol{x}) > -\infty$, where $d = \sum_{\ell=1}^{N_\ell} m_\ell n_\ell$ represents the total number of parameters.

**Assumption 2** (L-smoothness). Function $f : \mathbb{R}^d \to \mathbb{R}$ satisfies $\|\nabla f(\boldsymbol{x}) - \nabla f(\boldsymbol{y})\|_2 \leq L\|\boldsymbol{x} - \boldsymbol{y}\|_2, \forall \boldsymbol{x}, \boldsymbol{y} \in \mathbb{R}^d$.

**Assumption 3** (Stochastic gradient). It holds that

$$\mathbb{E}_{\xi \sim \mathcal{D}}[\nabla_\ell F(\boldsymbol{x}; \xi)] = \nabla_\ell f(\boldsymbol{x}), \quad \text{and}$$
$$\mathbb{E}_{\xi \sim \mathcal{D}}[\|\nabla_\ell F(\boldsymbol{x}; \xi) - \nabla_\ell f(\boldsymbol{x})\|_F^2] \leq \sigma_\ell^2, \quad \forall \boldsymbol{x} \in \mathbb{R}^d,$$

where $(F, \mathcal{D})$ represents the gradient oracle, $\sigma_\ell > 0$ is a scalar. Summing all weight matrices we obtain

$$\mathbb{E}_{\xi \sim \mathcal{D}}[\nabla F(\boldsymbol{x}; \xi)] = \nabla f(\boldsymbol{x}), \quad \text{and}$$
$$\mathbb{E}_{\xi \sim \mathcal{D}}[\|\nabla F(\boldsymbol{x}; \xi) - \nabla f(\boldsymbol{x})\|_2^2] \leq \sigma^2, \quad \forall \boldsymbol{x} \in \mathbb{R}^d,$$

where $\sigma = \sqrt{\sum_{\ell=1}^{N_\ell} \sigma_\ell^2}$.

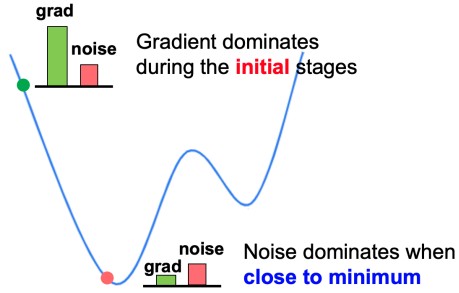

Gradient dominates during the **initial** stages

Noise dominates when **close to minimum**

Figure 2. Gradient noise dominates when close to local minimum.

## 3. Non-convergence of GaLore

In this section, we demonstrate why GaLore cannot guarantee exact convergence under Assumptions 1-3. We first illustrate the insight, then present the formal conclusion.

**Insight behind non-convergence.** As reviewed in Sec. 2, GaLore performs SVD on stochastic gradient $G = U\Sigma V^\top$ and obtains rank-$r$ projection matrices $P = U[:,:r] \in \mathbb{R}^{m\times r}$. GaLore projects $G$ onto $P$, yielding a low-rank gradient representation $P^\top G \in \mathbb{R}^{r\times n}$. In other words, Ga-Lore projects the stochastic gradient matrix onto a low-rank subspace spanned by the top $r$ singular vectors, capturing the dominant components of the stochastic gradient matrix. However, the stochastic gradient comprises two components: the true gradient and gradient noise, as shown in Fig. 3. When the true gradient significantly exceeds the gradient noise, typically at the start of training (see Fig. 2), the low-rank subspace obtained via SVD effectively preserves the true gradient information. As training progresses and the true gradient diminishes to zero, especially near a local minimum (see Fig. 2), the subspace may become increasingly influenced by gradient noise. When gradient noise is not isotropic, the noise-dominated subspace captured by SVD may become orthogonal to the true gradient subspace due to its greedy nature, leading to non-convergence.

**Counter-Example.** We consider the following quadratic problem with gradient noise:

$$f(X) = \frac{1}{2}\|AX\|_F^2 + \langle B, X\rangle_F, \quad (1)$$
$$\nabla F(X;\xi) = \nabla f(X) + \xi\sigma C,$$

where $A = \begin{pmatrix} I_{n-r} & 0 \end{pmatrix} \in \mathbb{R}^{(n-r)\times n}$, $B = \begin{pmatrix} D & 0 \\ 0 & 0 \end{pmatrix} \in \mathbb{R}^{n\times n}$ with $D \in \mathbb{R}^{(n-r)\times(n-r)}$ generated randomly, $C = \begin{pmatrix} 0 & 0 \\ 0 & I_r \end{pmatrix} \in \mathbb{R}^{n\times n}$, $\xi$ is a random variable uniformly sampled from $\{1, -1\}$ per iteration, and $\sigma$ is used to control the gradient noise. It is straightforward to verify that problem (1) satisfies Assumptions 1-3. Moreover, as $X$ approaches the global minimum of $f(X)$, the true gradient $\nabla f(X) \to 0$, while the gradient noise persists with a variance on the order of $\sigma^2$. Fig. 1 illustrates the performance

of GaLore when solving problem (1). It is observed that GaLore fails to converge to the optimal solution, regardless of whether the AdamW or MSGD optimizer is used.

**Non-convergence of GaLore.** The following theorem depicts GaLore's non-convergence based on the above insight.

**Theorem 4** (Non-convergence of GaLore)**:** There exists an objective function $f : \mathbb{R}^d \to \mathbb{R}$ satisfying Assumptions 1, 2, a stochastic gradient oracle $(F, \mathcal{D})$ satisfying Assumption 3, an initial point $x^{(0)} \in \mathbb{R}^d$, a constant $\epsilon_0 > 0$ such that for any rank $r_\ell < \min\{m_\ell, n_\ell\}$, subspace changing frequency $\tau$, any optimizer $\rho$ that inputs a subspace gradient of shape $r_\ell \times n_\ell$ and outputs a subspace update direction of the same shape and any $t > 0$, it holds that

$$\|\nabla f(x^{(t)})\|_2^2 \geq \epsilon_0.$$

## 4. Conditions for GaLore to converge

**GaLore provably converges in the noise-free setting.** According to Sec. 2, GaLore fails to converge when gradient noise dominates the true gradient in magnitudes. This motivates us to examine the deterministic scenario where true gradient $\nabla f(x)$ can be accessed without any gradient noise. GaLore with noise-free gradients is presented in Alg. 1 (or Alg. 3 in Appendix B.3), where the true gradient oracle is highlighted with label (det.) . Without gradient noise, the projection matrix $P_\ell^{(t)}$ obtained by SVD can effectively capture the true gradient even when approaching a local minimum. For simplicity, we analyze GaLore with MSGD and the following momentum updating mechanism:

$$M_\ell^{(t)} = \begin{cases} (1-\beta_1)P_\ell^{(t)\top}P_\ell^{(t-1)}M_\ell^{(t-1)} + \beta_1 P_\ell^{(t)\top}G_\ell^{(t)}, & m_\ell \leq n_\ell; \\ (1-\beta_1)M_\ell^{(t-1)}Q_\ell^{(t-1)\top}Q_\ell^{(t)} + \beta_1 G_\ell^{(t)}Q_\ell^{(t)}, & m_\ell > n_\ell; \end{cases}$$
$$(2)$$

If the subspace does not change at iteration $t$, it holds that $(P_\ell^{(t)})^\top P_\ell^{(t-1)} = (Q_\ell^{(t-1)})^\top Q_\ell^{(t)} = I_{r_\ell}$ and (2) reduces to regular momentum updates. If the subspace changes at iteration $t$, we inherit $M_\ell^{(t-1)}$ by first projecting back to the original space and then to the new subspace. We use *momentum projection (MP)* to refer to mechanism (2). When MP is used in the algorithm, we label the corresponding with (w/ MP) in Alg. 1 otherwise (w/o MP) . The following theorem provides convergence guarantees for GaLore using deterministic gradients and MSGD with MP.

**Theorem 5** (Convergence rate of deterministic GaLore)**:** Under Assumptions 1-2, if the number of iterations $T \geq 64/(3\underline{\delta})$ and we choose hyperparameters $\beta_1, \tau, \eta$ according to Appendix B.3, GaLore using deterministic gradient and momentum gradient descent with MP converges as

$$\frac{1}{T}\sum_{t=0}^{T-1}\|\nabla f(x^{(t)})\|_2^2 = \mathcal{O}\left(\frac{L\Delta}{\underline{\delta}^{5/2}T}\right),$$

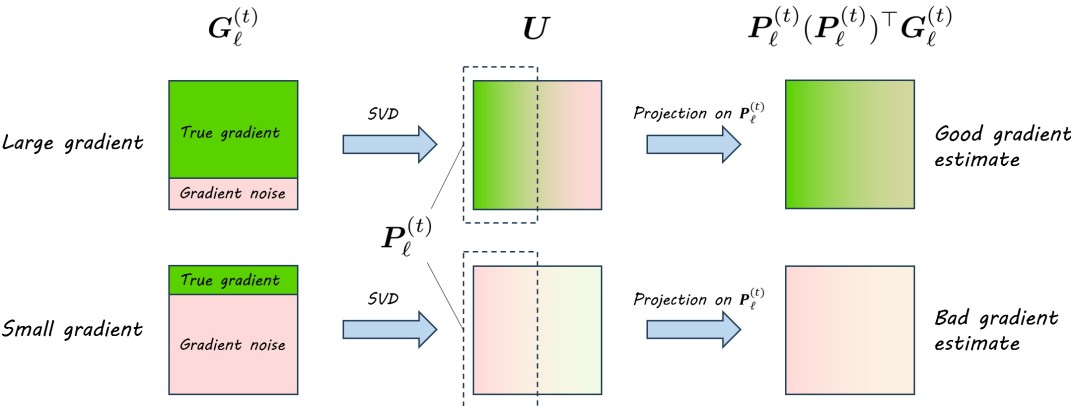

*Figure 3.* An illustration of the insight on why GaLore fails to converge in small-gradient scenarios. We use color green for true gradient and red for gradient noise.

where $\Delta = f(\boldsymbol{x}^{(0)}) - \inf_{\boldsymbol{x}} f(\boldsymbol{x})$ and $\underline{\delta} := \min_{\ell} \frac{r_{\ell}}{\min\{m_{\ell}, n_{\ell}\}}$.

**Remark 1.** Theorem 5 demonstrates that GaLore converges at a rate of $\mathcal{O}(1/T)$ in the deterministic scenario, which is on the same order as deterministic full-space gradient descent. More details are presented in Theorem 10 in Appendix B.3. However, in deep learning tasks with exceptionally large training datasets, computing the true gradient becomes impractical due to significant computational and memory costs. Therefore, we will next focus on the stochastic setting.

**Remark 2.** When $r_{\ell} \to \min\{m_{\ell}, n_{\ell}\}$, $\underline{\delta} \to 1$, and the convergence rate in Theorem 5 reduces to that of full-parameter gradient descent. This implies the sharpness of our analysis.

**GaLore provably converges with large-batch stochastic gradients.** Inspired by the insight presented in Sec. 2, GaLore converges in cases where the true gradient dominates the gradient noise. This convergence can be ensured by reducing the gradient noise through an increased batch size, particularly as the algorithm approaches a local minimum. Specifically, we replace the stochastic gradient $\boldsymbol{G}_{\ell}^{(t)} = \nabla_{\ell} F(\boldsymbol{x}^{(t)}; \xi^{(t)})$ with large-batch gradient $\boldsymbol{G}_{\ell}^{(t)} = \frac{1}{\mathcal{B}} \sum_{b=1}^{\mathcal{B}} \nabla_{\ell} F(\boldsymbol{x}^{(t)}; \xi^{(t,b)})$, which reduces the variance of gradient noise by $\mathcal{B}$ times. The GaLore algorithm with large-batch stochastic gradients is presented in Alg. 1 (or Alg. 4 in Appendix B.4), where the large-batch stochastic gradient oracle is highlighted with the label (l.b.). It is worth noting that the non-convergence of GaLore primarily stems from the erroneous subspace dominated by gradient noise. Therefore, we compute a large-batch gradient only for the SVD step while maintaining a smaller batch size for other computations, see Alg. 1. As the batch size $\mathcal{B}$ increases with iteration $T$, GaLore provably converge to stationary solutions, as established in the following theorem:

**Theorem 6** (Convergence rate of large-batch GaLore)**:** Under Assumptions 1-3, if the number of iterations $T \geq 2 + 256/(3\underline{\delta}) + (256\sigma)^2/(9\sqrt{\underline{\delta}}L\Delta)$ and we choose hyperparameters $\tau, \mathcal{B}, \beta_1, \eta$ according to Appendix B.4, GaLore using

large-batch MSGD with MP converges as

$$\frac{1}{T} \sum_{t=0}^{T-1} \mathbb{E}[\|\nabla f(\boldsymbol{x}^{(t)})\|_2^2] = \mathcal{O}\left(\frac{L\Delta}{\underline{\delta}^{5/2}T} + \sqrt{\frac{L\Delta\sigma^2}{\underline{\delta}^{7/2}T}}\right),$$

where $\Delta = f(\boldsymbol{x}^{(0)}) - \inf_{\boldsymbol{x}} f(\boldsymbol{x})$ and $\underline{\delta} := \min_{\ell} \frac{r_{\ell}}{\min\{m_{\ell}, n_{\ell}\}}$.

**Remark 3.** A more detailed result is presented in Theorem 12 in Appendix B.4. The large batch size $\mathcal{B} = \Theta(\sqrt{T})$ grows with iteration $T$, leading to increased memory overhead, making it less practical than small batch sizes. With gradient accumulation, an additional variable is needed to track the gradient, complicating compatibility with per-layer weight updates. Otherwise, larger batch sizes raise the memory for activation values. Therefore, exploring algorithms that converge with constant batch sizes becomes essential.

**GaLore provably converges under isotropic noise assumptions.** In Appendix C, we demonstrate that under specific isotropic noise assumptions, the SVD-induced subspace reliably preserves the true gradient information. Consequently, GaLore, even with constant batch sizes, achieves a guaranteed convergence rate of $\mathcal{O}(1/\sqrt{T})$. However, isotropic gradient noise is rarely considered in the convergence analysis of machine learning or deep learning algorithms (Zhu et al., 2018; HaoChen et al., 2021; Mori et al., 2022; Wu et al., 2022; Wang & Wu; Koloskova et al., 2020).

**Empirical validation.** Fig. 1 illustrates the convergence of large-batch GaLore (blue curve) in solving problem (1). It demonstrates that large-batch GaLore effectively corrects the bias present in small-batch stochastic GaLore (green curve), achieving convergence to stationary solutions.

## 5. GoLore: random low-rank projection

**GoLore algorithm.** The main issue with SVD-based projection in GaLore is that it aims to capture the dominant component in the stochastic gradient matrix. Consequently, when gradient noise overshadows the true gradient as the

**Algorithm 1** GaLore / GoLore algorithm framework using stochastic / deterministic / large-batch gradients with / without momentum projection

**Input:** Initial point $\boldsymbol{x}^{(0)}$, data distribution $\mathcal{D}$, learning rate $\eta$, subspace changing frequency $\tau$, rank $\{r_\ell\}_{\ell=1}^{N_L}$, optimizer hyperparameters $\beta_1$, $\beta_2$, $\epsilon$, large batch size $\mathcal{B}$.

**Output:** $\{\boldsymbol{x}^{(t)}\}_{t=0}^T$.

Initialize optimizer state $\{\boldsymbol{M}_\ell^{(-1)}\}_{\ell=1}^{N_L}$ and $\{\boldsymbol{V}_\ell^{(-1)}\}_{\ell=1}^{N_L}$ to zero;

**for** $t = 0, 1, \cdots, T-1$ **do**

  **for** $\ell = 1, 2, \cdots, N_L$ **do**

    **if** $t \equiv 0 \pmod{\tau}$ **then**

      $\boldsymbol{G}_\ell^{(t)} \leftarrow \nabla_\ell F(\boldsymbol{x}^{(t)}; \xi^{(t)})$;   (sto.)

      $\boldsymbol{G}_\ell^{(t)} \leftarrow \nabla_\ell f(\boldsymbol{x}^{(t)})$;   (det.)

      $\boldsymbol{G}_\ell^{(t)} \leftarrow \frac{1}{\mathcal{B}} \sum_{b=1}^{\mathcal{B}} \nabla_\ell F(\boldsymbol{x}^{(t)}; \xi^{(t,b)})$;   (l.b.)

      $\boldsymbol{U}, \boldsymbol{\Sigma}, \boldsymbol{V} \leftarrow \mathrm{SVD}(\boldsymbol{G}_\ell^{(t)})$, $\boldsymbol{P}_\ell^{(t)} \leftarrow \boldsymbol{U}[:, : r_\ell]$,

      $\boldsymbol{Q}_\ell^{(t)} \leftarrow \boldsymbol{V}[:, : r_\ell]$;   (GaLore)

      Sample $\boldsymbol{P}_\ell^{(t)} \sim \mathcal{U}(\mathrm{St}_{m_\ell, r_\ell})$,

      $\boldsymbol{Q}_\ell^{(t)} \sim \mathcal{U}(\mathrm{St}_{n_\ell, r_\ell})$;   (GoLore)

    **else**

      $\boldsymbol{G}_\ell^{(t)} \leftarrow \nabla_\ell F(\boldsymbol{x}^{(t)}; \xi^{(t)})$;   (sto.)

      $\boldsymbol{G}_\ell^{(t)} \leftarrow \nabla_\ell f(\boldsymbol{x}^{(t)})$;   (det.)

      $\boldsymbol{G}_\ell^{(t)} \leftarrow \nabla_\ell F(\boldsymbol{x}^{(t)}; \xi^{(t)})$;   (l.b.)

      $\boldsymbol{P}_\ell^{(t)} \leftarrow \boldsymbol{P}_\ell^{(t-1)}, \boldsymbol{Q}_\ell^{(t)} \leftarrow \boldsymbol{Q}_\ell^{(t-1)}$;

    **end if**

    $\boldsymbol{R}_\ell^{(t)} \leftarrow \begin{cases} (\boldsymbol{P}_\ell^{(t)})^\top \boldsymbol{G}_\ell^{(t)}, & \text{if } m_\ell \le n_\ell; \\ \boldsymbol{G}_\ell^{(t)} \boldsymbol{Q}_\ell^{(t)}, & \text{if } m_\ell > n_\ell; \end{cases}$

    Compute $\boldsymbol{M}_\ell^{(t)}$ via (2)   (w/ MP)

    $\boldsymbol{M}_\ell^{(t)} \leftarrow (1 - \beta_1) \boldsymbol{M}_\ell^{(t-1)} + \beta_1 \boldsymbol{R}_\ell^{(t)}$;   (w/o MP)

    $\boldsymbol{V}_\ell^{(t)} \leftarrow (1 - \beta_2) \boldsymbol{V}_\ell^{(t-1)} + \beta_2 \boldsymbol{R}_\ell^{(t)} \odot \boldsymbol{R}_\ell^{(t)}$;

    **if** using Adam **then**

      $\boldsymbol{M}_\ell^{(t)} \leftarrow \boldsymbol{M}_\ell^{(t)} / (1 - (1 - \beta_1)^t)$;

      $\boldsymbol{V}_\ell^{(t)} \leftarrow \boldsymbol{V}_\ell^{(t)} / (1 - (1 - \beta_2)^t)$;

      $\boldsymbol{N}_\ell^{(t)} \leftarrow \boldsymbol{M}_\ell^{(t)} / (\sqrt{\boldsymbol{V}_\ell^{(t)}} + \epsilon)$;

    **else if** using MSGD **then**

      $\boldsymbol{N}_\ell^{(t)} \leftarrow \boldsymbol{M}_\ell^{(t)}$;

    **end if**

    $\boldsymbol{X}_\ell^{(t+1)} \leftarrow \begin{cases} \boldsymbol{X}_\ell^{(t)} - \eta \boldsymbol{P}_\ell^{(t)} \boldsymbol{N}_\ell^{(t)}, & \text{if } m_\ell \le n_\ell; \\ \boldsymbol{X}_\ell^{(t)} - \eta \boldsymbol{N}_\ell^{(t)} (\boldsymbol{Q}_\ell^{(t)})^\top, & \text{if } m_\ell > n_\ell; \end{cases}$

  **end for**

**end for**

**return** $\{\boldsymbol{x}^{(t)}\}_{t=0}^T$.

---

algorithm approaches a local minimum, the SVD-based projection fails to identify valuable gradient information.

To address this, we propose replacing the SVD-based projection with a random projection, which captures components of the stochastic gradient matrix randomly without any preference. This results in the GoLore algorithm presented in Alg. 1 (or Alg. 5 in Appendix B.5). In Alg. 1, the GaLore method highlighted with the label (GaLore) samples the projection matrix $\boldsymbol{P}_\ell^{(t)}$ via SVD. In contrast, the GoLore method highlighted with the label (GoLore) samples $\boldsymbol{P}_\ell^{(t)}$ from $\mathcal{U}(\mathrm{St}_{m_\ell, r_\ell})$, a uniform distribution on the $m_\ell \times r_\ell$ Stiefel manifold. The following proposition provides a practical strategy to sample from distribution $\mathcal{U}(\mathrm{St}_{m,r})$.

**Proposition 7** (Chikuse (2012), Theorem 2.2.1). *A random matrix $\boldsymbol{X}$ uniformly distributed on $\mathrm{St}_{m,r}$ is expressed as $\boldsymbol{X} = \boldsymbol{Z}(\boldsymbol{Z}^\top \boldsymbol{Z})^{-1/2}$, where the elements of $\boldsymbol{Z} \in \mathbb{R}^{m \times r}$ are independent and identically distributed as normal $\mathcal{N}(0, 1)$.*

**Convergence guarantee.** Unlike SVD used in GaLore, the random sampling strategy in GoLore prevents the subspace from being dominated by gradient noise. The theorem below provides convergence guarantees for GoLore when using small-batch stochastic gradients and MSGD with MP.

**Theorem 8** (Convergence rate of GoLore): Under Assumptions 1-3, for any $T \ge 2 + 128/(3\underline{\delta}) + (128\sigma)^2/(9\sqrt{\underline{\delta}}L\Delta)$, if we choose hyperparameters $\beta_1$, $\tau$, $\eta$ according to Appendix B.5, GoLore using small-batch stochastic gradients and MSGD with MP converges as

$$\frac{1}{T} \sum_{t=0}^{T-1} \mathbb{E}[\|\nabla f(\boldsymbol{x}^{(t)})\|_2^2] = \mathcal{O}\left( \frac{L\Delta}{\underline{\delta}^{5/2} T} + \sqrt{\frac{L\Delta\sigma^2}{\underline{\delta}^{7/2} T}} \right),$$

where $\Delta = f(\boldsymbol{x}^{(0)}) - \inf_{\boldsymbol{x}} f(\boldsymbol{x})$ and $\underline{\delta} := \min_\ell \frac{r_\ell}{\min\{m_\ell, n_\ell\}}$.

**Remark 4.** Theorem 8 demonstrates that GoLore converges at a rate of $\mathcal{O}(1/\sqrt{T})$, which is consistent with the convergence rate of full-parameter pre-training using standard MSGD. A more detailed result is presented in Theorem 14 in Appendix B.5, where we established convergence for more general hyperparameter choices. Unlike deterministic GaLore and low-rank GaLore discussed in Sec. 2, the newly proposed GoLore algorithm converges in the nonconvex stochastic setting with constant batch sizes. Furthermore, GoLore converges without assuming isotropic gradient noise, and it remains effective whether the gradient noise is anisotropic or not, making it significantly more suitable for LLM pre-training and fine-tuning.

**Remark 5.** Notably, this paper presents the first discrete-time convergence analysis for GaLore-like algorithms under standard assumptions. Among the few GaLore-like studies providing convergence guarantees, Liang et al. (2024) establishes continuous-time convergence, while Robert et al.

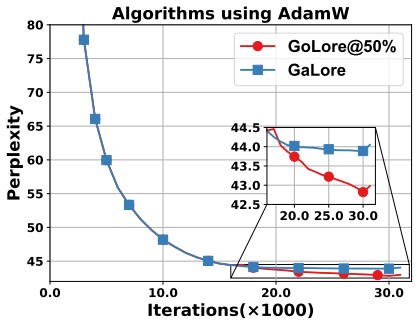
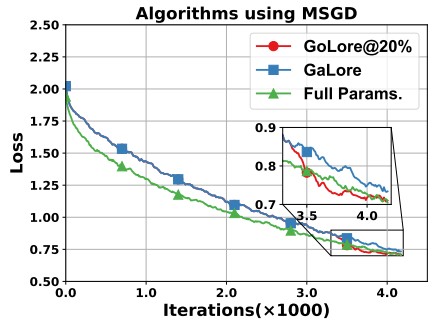

*Figure 4.* Pre-training curves of various approaches using AdamW with BF16 precision.

*Figure 5.* Fine-tuning curves of various approaches using MSGD with BF16 precision.

*Table 1.* Fine-tuning results on GLUE benchmark using pre-trained RoBERTa-Base.

| Algorithm | CoLA | STS-B | MRPC | RTE | SST2 | MNLI | QNLI | QQP | Avg |
|-----------|------|-------|------|-----|------|------|------|-----|-----|
| Full Params. | 62.07 | 90.18 | 92.25 | 78.34 | 94.38 | 87.59 | 92.46 | 91.90 | 86.15 |
| GaLore | 61.32 | 90.24 | 92.55 | 77.62 | **94.61** | 86.92 | 92.06 | 90.84 | 85.77 |
| FLORA | 57.71 | 89.59 | 91.96 | 76.17 | 94.50 | 85.42 | 91.93 | 90.49 | 84.72 |
| GoLore@20% | **61.66** | **90.55** | **92.93** | **78.34** | **94.61** | 87.02 | **92.20** | 90.91 | **86.03** |

*Table 2.* Results for fine-tuning pre-trained OPT-13B models on BoolQ. *OOM* stands for "out of memory".

| Algorithm | Memory | Accuracy |
|-----------|--------|----------|
| Full Params. | OOM | - |
| GaLore | 77.68 GB | 79.79 |
| GoLore@30% | 77.68 GB | **81.96** |

(2024) demonstrates convergence using a more complex error feedback technique, relying on a contractive assumption on the projection matrix that becomes stronger as the subspace recomputing period $\tau > 1$. In contrast, GoLore guarantees convergence by replacing the SVD projection with a random projection, without significantly altering the algorithmic structure or introducing restrictive assumptions.

**Practical application of GoLore in LLMs.** While GoLore have theoretical convergence guarantees, directly applying GoLore in LLM tasks may not be ideal. The advantage of using randomly sampled projection matrices becomes evident in the later stages of training, where stochastic gradients are primarily dominated by gradient noise. However, in the early stages, projection matrices derived from GaLore's SVD retain more gradient information, leading to more effective subspaces, see Fig. 2. Therefore, we recommend a *hybrid* approach: initially using GaLore to converge toward the neighborhood of the solution, then switching to GoLore for refinement and achieving more accurate results.

**Empirical validation.** Fig. 1 shows the convergence of the hybrid algorithm (red curve) applied to problem (1), which employs GaLore in the early training phase and switches to GoLore in the later stage. It is observed that the hybrid algorithm successfully converges to stationary solutions.

# 6. Experiments

We evaluate GaLore and GoLore on several different tasks, including a counter-example problem (1), pre-training and fine-tuning LLMs with real benchmarks. Throughout our experiments, *GoLore@x%* uses GaLore in the first $(100-x)\%$ iterations and GoLore in the last $x\%$ iterations, *L.B. GaLore* denotes large-batch GaLore, and *Full Params.* denotes full-parameter training. Further results and detailed experimental specifications including the hyperparameter choices and computing resources are deferred to Appendix E and F.

**GaLore's non-convergence.** In order to validate the non-convergence of GaLore and the convergence properties of GoLore and large-batch GaLore, we compare them with full-parameter training on the constructed quadratic problem defined in (1). Fig. 1 shows that, regardless of whether AdamW or MSGD is employed as the subspace optimizer, GaLore does not converge to the desired solution. In contrast, both GoLore and large-batch GaLore, along with full-parameter training, achieve exact convergence, thereby validating our theoretical results.

**Pre-training.** To validate GoLore in LLM pre-training tasks, we pre-trained LLaMA-60M on the C4 (Raffel et al.,

*Table 3.* Memory and computation comparison between GaLore's original implementation and our ReLoRA-like version, both utilizing MSGD with batch size $b$. We assume the weight $\boldsymbol{W} \in \mathbb{R}^{m \times n}$ satisfies $m \leq n$.

| GaLore Implementation | Memory | Computation |
|---|---|---|
| (Zhao et al., 2024) | $mn + rm + rn + bm$ | $6bmn + 4rmn + 2mn + 3rn$ |
| Our ReLoRA-like version | $mn + rm + 2rn + bm + br$ | $4bmn + 4brm + 6brn + 5rn$ |

2020) dataset for 30,000 iterations using various algorithms, including GaLore, GoLore and full-parameter training. All implementations utilized the AdamW optimizer in BF16 format. As illustrated in Fig. 4, the perplexity dramatically decreases when shifting from GaLore to GoLore, demonstrating the effectiveness of our approach.

**Fine-tuning.** To validate the efficiency of GoLore in LLM fine-tuning tasks, we fine-tuned pre-trained RoBERTa models (Liu, 2019) on the GLUE benchmark (Wang, 2018), LLaMA2-7B models (Touvron et al., 2023) on the Wino-Grande dataset (Sakaguchi et al., 2021), and OPT-13B models (Zhang et al., 2022) on the BoolQ dataset (Clark et al., 2019) Fig. 5 displays the loss curves for fine-tuning on WinoGrande with rank 1024, while Table 1 and 2 present the task scores for GaLore/GoLore with rank 4. GoLore consistently outperforms GaLore in the above experiments.

## 7. Connections with other algorithms

**Connection with ReLoRA.** Optimization algorithms like GaLore/GoLore that optimizes in periodically recomputed subspaces can be implemented in an equivalent yet potentially more computational efficient, ReLoRA-like way. Consider a linear layer $\boldsymbol{y} = \boldsymbol{W}\boldsymbol{x}$ with $\boldsymbol{W} \in \mathbb{R}^{m \times n}$, where $m \leq n$, GaLore first computes the full-parameter gradient $\nabla_{\boldsymbol{W}}\mathcal{L} = (\nabla_{\boldsymbol{y}}\mathcal{L})\boldsymbol{x}^{\top}$ via back propagation and update $\boldsymbol{W}$ in the subspace as $\boldsymbol{W} \leftarrow \boldsymbol{W} + \boldsymbol{P}\rho(\boldsymbol{P}^{\top}(\nabla_{\boldsymbol{W}}\mathcal{L}))$, where $\boldsymbol{P} \in \mathbb{R}^{m \times r}$ is a low-rank projection matrix. If we use LoRA adaptation $\boldsymbol{W} = \boldsymbol{W}_0 + \boldsymbol{B}\boldsymbol{A}$ with $\boldsymbol{B} \in \mathbb{R}^{m \times r}$ and $\boldsymbol{A} \in \mathbb{R}^{r \times n}$, we compute $\boldsymbol{A}$'s gradient $\nabla_{\boldsymbol{A}}\mathcal{L} = (\nabla_{\boldsymbol{z}}\mathcal{L})\boldsymbol{x}^{\top} = \boldsymbol{B}^{\top}(\nabla_{\boldsymbol{y}}\mathcal{L})\boldsymbol{x}^{\top}$, where $\boldsymbol{z} = \boldsymbol{B}\boldsymbol{x}$ is the additional activation. If we fix $\boldsymbol{B} = \boldsymbol{P}$, update $\boldsymbol{A} \leftarrow \boldsymbol{A} + \rho(\nabla_{\boldsymbol{A}}\mathcal{L})$ is equivalent to $\boldsymbol{W} \leftarrow \boldsymbol{W} + \boldsymbol{P}\rho(\boldsymbol{P}^{\top}(\nabla_{\boldsymbol{W}}\mathcal{L}))$. The memory and computational costs of the two implementations are compared in Table 3, showing the potential of our ReLoRA-like implementation to reduce computation with little memory overhead. Detailed algorithm descriptions and calculations are in Appendix A.

**Connection with FLORA.** Aware of the equivalence of the two (GaLore/ReLoRA-like) implementations, the main difference between GoLore and FLORA lies in the choice of projection matrices. Though both algorithms sample $\boldsymbol{P} \in \mathbb{R}^{m \times r}$ randomly, GoLore uses a uniform distribution on the Stiefel manifold $\mathcal{U}(\mathrm{St}_{m,r})$, while FLORA uses a random Gaussian distribution where each element in $\boldsymbol{P}$

is independently sampled from $\mathcal{N}(0, 1/r)$, and thus $\boldsymbol{P}$ may not belongs to $\mathrm{St}_{m,r}$.

**Connection with SIFT.** SIFT fine-tunes LLMs with sparsified gradients, which can also be viewed as subspace descent. While GaLore projects gradient $\boldsymbol{G}$ to $\boldsymbol{P}^{\top}\boldsymbol{G}$ via a projection matrix $\boldsymbol{P}$, SIFT projects gradient $\boldsymbol{G}$ to $\boldsymbol{S} \odot \boldsymbol{G}$ via a sparse mask matrix $\boldsymbol{S}$. Our theoretical analysis can be directly transferred to sparse subspace descent with little effort, implying similar results as in low-rank subspace descent, see Appendix D.

**Connection with zero-th order methods.** Zero-th order methods (Malladi et al., 2023; Zhang et al., 2023; Chen et al., 2024b) are another line of works on memory-efficient training. While these algorithms randomly select a direction to estimate the directional derivatives by finite difference, GoLore computes subspace gradients via back propagation. The directions used in zero-th order methods change every iteration, while GoLore applies a more lazily strategy changing its subspace every $\tau$ iterations.

**Connection with gradient sketching methods.** Gradient sketching methods like Hanzely et al. (2018) and Wang et al. (2024) uses gradient sketches in algorithm iterates. These methods recover gradient estimates from projected gradients and retains full-size gradients and optimizer states. In comparison, GoLore directly updates with projected gradients and retains compressed gradients and optimizer states, which is more memory-efficient.

## 8. Conclusion and Limitations

This paper investigates subspace optimization approaches for LLM pre-training and fine-tuning. We demonstrate that GaLore fails to converge to the desired solution under regular assumptions, as the SVD-based projection often generates potentially anisotropic noise-dominated subspaces when the true gradient is relatively small. However, we establish that GaLore can achieve exact convergence when using deterministic or large-batch stochastic gradients. We further introduce GoLore—a variant of GaLore employing randomly sampled projection matrices—and establish its convergence rate even with small-batch stochastic gradients. A limitation of this paper is that our convergence analysis framework has not readily covered the use of the Adam optimizer and recent GaLore variants such as Fira.

## Acknowledgements

The original submission is supported by the National Natural Science Foundation of China (No. 124B2017, 92370121, 12301392) and the National Key Research and Development Program of China (No. 2024YFA1012902). The final revisions are additionally supported by Zhongguancun Academy Project (No. C20250205).

## Impact Statement

This paper studies the non-convergence and convergence properties of GaLore under different assumptions, whose goal is to gain better understanding of GaLore and design better memory-efficient optimization algorithms. Aiming to advance the field of Machine Learning, we do not feel any potential societal consequences of this work necessary to be discussed here.

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

## A. The ReLoRA-like implementation

An equivalent, ReLoRA-like implementation of Alg. 1 is as illustrated in Alg. 2, where we only present the case with small-batch stochastic gradients for convenience. In fact, applying ReLoRA with a fixed $A$ or $B$ is not our contribution, as it has already been used in several previous works(Hao et al., 2024; Loeschcke et al., 2024). While leading to the same results, this ReLoRA-like implementation (Alg. 2) can potentially save computation as it computes the subspace gradient directly without computing the full-parameter one. Consider the case where $m \leq n$ and we use MSGD and a batch size of $b$. The computation complexity of GaLore's original implementation is $2bmn$ for forward propagation, $4bmn$ for backward propagation, $4rmn$ for projection, $3rn$ for momentum update and $2mn$ for weight update. The computational complexity of our ReLoRA-like implementation is $2bmn + 2brm + 2brn$ for forward propagation, $2bmn + 2brm + 2brn$ for backward propagation, $3rn$ for momentum updates and $2rn$ for weight updates. As illustrated in Table 3, our implementation can potentially reduce computation with little memory overhead.

## B. Theoretical proofs

### B.1. Notations and useful lemmas

We assume the model parameters consist of $N_L$ weight matrices. We use $\boldsymbol{X}_\ell \in \mathbb{R}^{m_\ell \times n_\ell}$ to denote the $\ell$-th weight matrix and $\boldsymbol{x} \in \mathbb{R}^d = (\text{vec}(\boldsymbol{X}_1)^\top, \cdots, \text{vec}(\boldsymbol{X}_{N_L})^\top)^\top$ to denote the vector collecting all the parameters, $d = \sum_{\ell=1}^{N_L} m_\ell n_\ell$. We assume GaLore/GoLore applies rank-$r_\ell$ projection to the $\ell$-th weight matrix and denote

$$\delta_\ell = \frac{r_\ell}{\min\{m_\ell, n_\ell\}}, \quad \underline{\delta} = \min_{1 \leq \ell \leq N_L} \delta_\ell, \quad \overline{\delta} = \max_{1 \leq \ell \leq N_l} \delta_\ell.$$

We define $\tilde{\boldsymbol{M}}_\ell^{(t)}$ as

$$\tilde{\boldsymbol{M}}_\ell^{(t)} = \begin{cases} \boldsymbol{P}_\ell^{(t)} \boldsymbol{M}_\ell^{(t)}, & \text{if } m_\ell \leq n_\ell, \\ \boldsymbol{M}_\ell^{(t)} (\boldsymbol{Q}_\ell^{(t)})^\top, & \text{if } m_\ell > n_\ell, \end{cases}$$

and $\tilde{\boldsymbol{m}} = (\text{vec}(\tilde{\boldsymbol{M}}_1)^\top, \cdots, \text{vec}(\tilde{\boldsymbol{M}}_{N_L})^\top)^\top$. While using Alg. 1 with MSGD and MP, it holds for $m_\ell \leq n_\ell$ that

$$\tilde{\boldsymbol{M}}_\ell^{(t)} = \begin{cases} \beta_1 \boldsymbol{P}_\ell^{(0)} (\boldsymbol{P}_\ell^{(0)})^\top \boldsymbol{G}_\ell^{(0)}, & t = 0; \\ \boldsymbol{P}_\ell^{(t)} (\boldsymbol{P}_\ell^{(t)})^\top \left( (1 - \beta_1) \tilde{\boldsymbol{M}}_\ell^{(t-1)} + \beta_1 \boldsymbol{G}_\ell^{(t)} \right), & t = k\tau, \ k \in \mathbb{N}^*; \\ (1 - \beta_1) \tilde{\boldsymbol{M}}_\ell^{(t-1)} + \beta_1 \boldsymbol{P}_\ell^{(t)} (\boldsymbol{P}_\ell^{(t)})^\top \boldsymbol{G}_\ell^{(t)}, & t = k\tau + r, \ k \in \mathbb{N}, \ 1 \leq r < \tau; \end{cases}$$

for $m_\ell > n_\ell$ that

$$\tilde{\boldsymbol{M}}_\ell^{(t)} = \begin{cases} \beta_1 \boldsymbol{G}_\ell^{(0)} \boldsymbol{Q}_\ell^{(0)} (\boldsymbol{Q}_\ell^{(0)})^\top, & t = 0; \\ \left( (1 - \beta_1) \tilde{\boldsymbol{M}}_\ell^{(t-1)} + \beta_1 \boldsymbol{G}_\ell^{(t)} \right) \boldsymbol{Q}_\ell^{(t)} (\boldsymbol{Q}_\ell^{(t)})^\top, & t = k\tau, \ k \in \mathbb{N}^*; \\ (1 - \beta_1) \tilde{\boldsymbol{M}}_\ell^{(t-1)} + \beta_1 \boldsymbol{G}_\ell^{(t)} \boldsymbol{Q}_\ell^{(t)} (\boldsymbol{Q}_\ell^{(t)})^\top, & t = k\tau + r, \ k \in \mathbb{N}, \ 1 \leq r < \tau; \end{cases}$$

and for both cases that

$$\boldsymbol{X}_\ell^{(t+1)} = \boldsymbol{X}_\ell^{(t)} - \eta \tilde{\boldsymbol{M}}_\ell^{(t)}.$$

**Lemma 1** (Error of GaLore's projection)**:** Let $\boldsymbol{G} = \boldsymbol{U} \boldsymbol{\Sigma} \boldsymbol{V}^\top$ be the SVD of $\boldsymbol{G} \in \mathbb{R}^{m \times n}$, projection matrix $\boldsymbol{P} = \boldsymbol{U}[:, : r]$, $\boldsymbol{Q} = \boldsymbol{V}[:, : r]$, $r < \min\{m, n\}$. It holds for $m \leq n$ that

$$\|\boldsymbol{P}\boldsymbol{P}^\top \boldsymbol{G} - \boldsymbol{G}\|_F^2 \leq \left( 1 - \frac{r}{m} \right) \|\boldsymbol{G}\|_F^2,$$

and for $m > n$ that

$$\|\boldsymbol{G}\boldsymbol{Q}\boldsymbol{Q}^\top - \boldsymbol{G}\|_F^2 \leq \left( 1 - \frac{r}{n} \right) \|\boldsymbol{G}\|_F^2.$$

---

**Algorithm 2** ReLoRA-like implementation of GaLore / GoLore algorithm using stochastic gradients with / without momentum projection

---

**Input:** Initial point $\boldsymbol{x}^{(0)}$, data distribution $\mathcal{D}$, learning rate $\eta$, subspace changing frequency $\tau$, rank $\{r_\ell\}_{\ell=1}^{N_L}$, optimizer hyperparameters $\beta_1, \beta_2, \epsilon$, large batch size $\mathcal{B}$.

**Output:** $\{\boldsymbol{x}^{(t)}\}_{t=0}^{T}$.

    Initialize LoRA adaptation $\boldsymbol{X}_\ell = \boldsymbol{W}_\ell + \boldsymbol{B}_\ell \boldsymbol{A}_\ell$ for $\ell = 1, 2, \cdots, N_L$, where $\boldsymbol{W}_\ell^{(0)} = \boldsymbol{X}_\ell^{(0)}$, $\boldsymbol{A}_\ell^{(0)} = 0$ and $\boldsymbol{B}_\ell^{(0)} = 0$;

    Initialize optimizer state $\{\boldsymbol{M}_\ell^{(-1)}\}_{\ell=1}^{N_L}$ and $\{\boldsymbol{V}_\ell^{(-1)}\}_{\ell=1}^{N_L}$ to zero;

    **for** $t = 0, 1, \cdots, T - 1$ **do**

        **for** $\ell = 1, 2, \cdots, N_L$ **do**

            **if** $t \equiv 0 \pmod{\tau}$ **then**

                $\boldsymbol{G}_\ell^{(t)} \leftarrow \nabla_\ell F(\boldsymbol{x}^{(t)}; \xi^{(t)})$;

                $\boldsymbol{U}, \boldsymbol{\Sigma}, \boldsymbol{V} \leftarrow \mathrm{SVD}(\boldsymbol{G}_\ell^{(t)}), \boldsymbol{P}_\ell^{(t)} \leftarrow \boldsymbol{U}[:, :r_\ell], \boldsymbol{Q}_\ell^{(t)} \leftarrow \boldsymbol{V}[:, :r_\ell]$;  (GaLore)

                Sample $\boldsymbol{P}_\ell^{(t)} \sim \mathcal{U}(\mathrm{St}_{m_\ell, r_\ell}), \quad \boldsymbol{Q}_\ell^{(t)} \sim \mathcal{U}(\mathrm{St}_{n_\ell, r_\ell})$;  (GoLore)

$$\boldsymbol{R}_\ell^{(t)} \leftarrow \begin{cases} (\boldsymbol{P}_\ell^{(t)})^\top \boldsymbol{G}_\ell^{(t)}, & \text{if } m_\ell \leq n_\ell; \\ \boldsymbol{G}_\ell^{(t)} \boldsymbol{Q}_\ell^{(t)}, & \text{if } m_\ell > n_\ell; \end{cases}$$

            **else**

$$\boldsymbol{R}_\ell^{(t)} \leftarrow \begin{cases} \nabla_{\boldsymbol{A}_\ell} F(\boldsymbol{x}^{(t)}; \xi^{(t)}), & \text{if } m_\ell \leq n_\ell; \\ \nabla_{\boldsymbol{B}_\ell} F(\boldsymbol{x}^{(t)}; \xi^{(t)}), & \text{if } m_\ell > n_\ell; \end{cases}$$

            **end if**

$$M_\ell^{(t)} \leftarrow \begin{cases} (1 - \beta_1)(\boldsymbol{P}_\ell^{(t)})^\top \boldsymbol{B}_\ell^{(t)} \boldsymbol{M}_\ell^{(t-1)} + \beta_1 \boldsymbol{R}_\ell^{(t)}, & \text{if } m_\ell \leq n_\ell; \\ (1 - \beta_1) \boldsymbol{M}_\ell^{(t-1)} \boldsymbol{A}_\ell^{(t)} \boldsymbol{Q}_\ell^{(t)} + \beta_1 \boldsymbol{R}_\ell^{(t)}, & \text{if } m_\ell > n_\ell; \end{cases} \quad \text{(with MP)}$$

$$\boldsymbol{M}_\ell^{(t)} \leftarrow (1 - \beta_1) \boldsymbol{M}_\ell^{(t-1)} + \beta_1 \boldsymbol{R}_\ell^{(t)}; \quad \text{(without MP)}$$

            $\boldsymbol{V}_\ell^{(t)} \leftarrow (1 - \beta_2) \boldsymbol{V}_\ell^{(t-1)} + \beta_2 \boldsymbol{R}_\ell^{(t)} \odot \boldsymbol{R}_\ell^{(t)}$;

            **if** using Adam **then**

                $\boldsymbol{M}_\ell^{(t)} \leftarrow \boldsymbol{M}_\ell^{(t)}/(1 - \beta_1^t), \quad \boldsymbol{V}_\ell^{(t)} \leftarrow \boldsymbol{V}_\ell^{(t)}/(1 - \beta_2^t), \quad \boldsymbol{N}_\ell^{(t)} \leftarrow \boldsymbol{M}_\ell^{(t)}/(\sqrt{\boldsymbol{V}_\ell^{(t)}} + \epsilon)$;

            **else if** using MSGD **then**

                $\boldsymbol{N}_\ell^{(t)} \leftarrow \boldsymbol{M}_\ell^{(t)}$;

            **end if**

            **if** $t \equiv 0 \pmod{\tau}$ **then**

                $\boldsymbol{W}_\ell^{(t+1)} \leftarrow \boldsymbol{W}_\ell^{(t)} + \boldsymbol{B}_\ell^{(t)} \boldsymbol{A}_\ell^{(t)}$;

$$\boldsymbol{A}_\ell^{(t+1)} \leftarrow \begin{cases} -\eta \boldsymbol{N}_\ell^{(t)}, & \text{if } m_\ell \leq n_\ell; \\ (\boldsymbol{Q}_\ell^{(t)})^\top, & \text{if } m_\ell > n_\ell; \end{cases}$$

$$\boldsymbol{B}_\ell^{(t+1)} \leftarrow \begin{cases} \boldsymbol{P}_\ell^{(t)}, & \text{if } m_\ell \leq n_\ell; \\ -\eta \boldsymbol{N}_\ell^{(t)}, & \text{if } m_\ell > n_\ell; \end{cases}$$

            **else**

                $\boldsymbol{W}_\ell^{(t+1)} \leftarrow \boldsymbol{W}_\ell^{(t)}$;

$$\boldsymbol{A}_\ell^{(t+1)} \leftarrow \begin{cases} \boldsymbol{A}_\ell^{(t)} - \eta \boldsymbol{N}_\ell^{(t)}, & \text{if } m_\ell \leq n_\ell; \\ \boldsymbol{A}_\ell^{(t)}, & \text{if } m_\ell > n_\ell; \end{cases}$$

$$\boldsymbol{B}_\ell^{(t+1)} \leftarrow \begin{cases} \boldsymbol{B}_\ell^{(t)}, & \text{if } m_\ell \leq n_\ell; \\ \boldsymbol{B}_\ell^{(t)} - \eta \boldsymbol{N}_\ell^{(t)}, & \text{if } m_\ell > n_\ell; \end{cases}$$

            **end if**

        **end for**

    **end for**

*Proof.* Without loss of generality assume $m \leq n$ (the other case can be proved similarly). Let $\boldsymbol{Q} = \boldsymbol{U}[:, (r+1):]$, It holds that $\boldsymbol{I} = \boldsymbol{U}\boldsymbol{U}^\top = \boldsymbol{P}\boldsymbol{P}^\top + \boldsymbol{Q}\boldsymbol{Q}^\top$. Thus,

$$
\begin{aligned}
\|\boldsymbol{P}\boldsymbol{P}^\top\boldsymbol{G} - \boldsymbol{G}\|_F^2 &= \|(\boldsymbol{I} - \boldsymbol{P}\boldsymbol{P}^\top)\boldsymbol{U}\boldsymbol{\Sigma}\boldsymbol{V}^\top\|_F^2 \\
&= \mathrm{tr}(\boldsymbol{V}\boldsymbol{\Sigma}^\top\boldsymbol{U}^\top(\boldsymbol{I} - \boldsymbol{P}\boldsymbol{P}^\top)^2\boldsymbol{U}\boldsymbol{\Sigma}\boldsymbol{V}^\top) \\
&= \mathrm{tr}(\boldsymbol{\Sigma}^\top\boldsymbol{U}^\top\boldsymbol{Q}\boldsymbol{Q}^\top\boldsymbol{U}\boldsymbol{\Sigma}),
\end{aligned}
\tag{3}
$$

where the second equation uses $\|\boldsymbol{X}\|_F^2 = \mathrm{tr}(\boldsymbol{X}^\top\boldsymbol{X})$ and the last equation uses $\mathrm{tr}(\boldsymbol{A}\boldsymbol{B}) = \mathrm{tr}(\boldsymbol{B}\boldsymbol{A})$, $\boldsymbol{V}^\top\boldsymbol{V} = \boldsymbol{I}$ and $\boldsymbol{Q}^\top\boldsymbol{Q} = \boldsymbol{I}$. By $\boldsymbol{Q}^\top\boldsymbol{P} = 0$ and $\boldsymbol{P}^\top\boldsymbol{Q} = 0$, we have

$$
\boldsymbol{U}^\top\boldsymbol{Q}\boldsymbol{Q}^\top\boldsymbol{U} = \begin{pmatrix} \boldsymbol{P}^\top \\ \boldsymbol{Q}^\top \end{pmatrix} \boldsymbol{Q}\boldsymbol{Q}^\top \begin{pmatrix} \boldsymbol{P} & \boldsymbol{Q} \end{pmatrix} = \begin{pmatrix} 0_{r\times r} & 0_{r\times(m-r)} \\ 0_{(m-r)\times r} & \boldsymbol{I}_{m-r} \end{pmatrix}.
\tag{4}
$$

Let $\sigma_1 \geq \sigma_2 \geq \cdots \geq \sigma_m \geq 0$ denote the eigenvalues of $\boldsymbol{G}$, (4) implies

$$
\boldsymbol{\Sigma}^\top\boldsymbol{U}^\top\boldsymbol{Q}\boldsymbol{Q}^\top\boldsymbol{U}\boldsymbol{\Sigma} = \begin{pmatrix} 0_{r\times r} & 0_{r\times(m-r)} & 0_{r\times(n-m)} \\ 0_{(m-r)\times r} & \mathrm{diag}(\sigma_{r+1}, \cdots, \sigma_m) & 0_{(m-r)\times(n-m)} \\ 0_{(n-m)\times r} & 0_{(n-m)\times(m-r)} & 0_{(n-m)\times(n-m)} \end{pmatrix}.
\tag{5}
$$

Applying (5) to (3) yields

$$
\|\boldsymbol{P}\boldsymbol{P}^\top\boldsymbol{G} - \boldsymbol{G}\|_F^2 = \mathrm{tr}(\boldsymbol{\Sigma}^\top\boldsymbol{U}^\top\boldsymbol{Q}\boldsymbol{Q}^\top\boldsymbol{U}\boldsymbol{\Sigma}) = \sum_{i=r+1}^m \sigma_i^2 \leq \frac{m-r}{m}\|\boldsymbol{G}\|_F^2,
$$

where the inequality uses $\|\boldsymbol{G}\|_F^2 = \mathrm{tr}(\boldsymbol{G}^\top\boldsymbol{G}) = \mathrm{tr}(\boldsymbol{\Sigma}^\top\boldsymbol{\Sigma}) = \sum_{i=1}^m \sigma_i^2$. $\qquad\square$

**Lemma 2** (Gradient connections)**:** It holds for any $t, \tau > 0$ that

$$
\|\nabla_\ell f(\boldsymbol{x}^{(0)})\|_F^2 \leq \frac{2}{\tau}\sum_{r=0}^{\tau-1}\|\nabla_\ell f(\boldsymbol{x}^{(t+r)})\|_F^2 + (\tau-1)\sum_{r=0}^{\tau-2}\|\nabla_\ell f(\boldsymbol{x}^{(t+r+1)}) - \nabla_\ell f(\boldsymbol{x}^{(t+r)})\|_F^2.
\tag{6}
$$

*Proof.* For any $r = 1, \cdots, \tau-1$, it holds that

$$
\begin{aligned}
\|\nabla_\ell f(\boldsymbol{x}^{(t)})\|_F^2 &= \|\nabla_\ell f(\boldsymbol{x}^{(t+r)}) - (\nabla_\ell f(\boldsymbol{x}^{(t+r)}) - \nabla_\ell f(\boldsymbol{x}^{(t)}))\|_F^2 \\
&\leq 2\|\nabla_\ell f(\boldsymbol{x}^{(t+r)})\|_F^2 + 2\|\nabla_\ell f(\boldsymbol{x}^{(t+r)}) - \nabla_\ell f(\boldsymbol{x}^{(t)})\|_F^2.
\end{aligned}
\tag{7}
$$

For any $r = 2, \cdots, \tau-1$, it holds that

$$
\begin{aligned}
\|\nabla_\ell f(\boldsymbol{x}^{(t+r)}) - \nabla_\ell f(\boldsymbol{x}^{(t)})\|_F^2 &= \left\|\sum_{i=1}^r \nabla_\ell f(\boldsymbol{x}^{(t+i)}) - \nabla_\ell f(\boldsymbol{x}^{(t+i-1)})\right\|_F^2 \\
&\leq r\sum_{i=1}^r \|\nabla_\ell f(\boldsymbol{x}^{(t+i)}) - \nabla_\ell f(\boldsymbol{x}^{(t+i-1)})\|_F^2,
\end{aligned}
\tag{8}
$$

where the inequality uses Cauchy's inequality. Summing (7) from $r = 1$ to $\tau-1$ and applying (8) yields

$$
\begin{aligned}
\tau\|\nabla_\ell f(\boldsymbol{x}^{(t)})\|_F^2 &\leq 2\sum_{r=0}^{\tau-1}\|\nabla_\ell f(\boldsymbol{x}^{(t+r)})\|_F^2 + 2\sum_{i=1}^{\tau-1}\sum_{j=1}^i i\|\nabla_\ell f(\boldsymbol{x}^{(t+j)}) - \nabla_\ell f(\boldsymbol{x}^{(t+j-1)})\|_F^2 \\
&\leq 2\sum_{r=0}^{\tau-1}\|\nabla_\ell f(\boldsymbol{x}^{(t+r)})\|_F^2 + 2\sum_{j=1}^{\tau-1}\sum_{i=1}^{\tau-1} i\|\nabla_\ell f(\boldsymbol{x}^{(t+j)}) - \nabla_\ell f(\boldsymbol{x}^{(t+j-1)})\|_F^2 \\
&= 2\sum_{r=0}^{\tau-1}\|\nabla_\ell f(\boldsymbol{x}^{(t+r)})\|_F^2 + \tau(\tau-1)\sum_{j=1}^{\tau-1}\|\nabla_\ell f(\boldsymbol{x}^{(t+j)}) - \nabla_\ell f(\boldsymbol{x}^{(t+j-1)})\|_F^2,
\end{aligned}
$$

which is exactly (6). $\qquad\square$

**Lemma 3** (Projection orthogonality)**:** If $\boldsymbol{P} \in \mathrm{St}_{m,r}$, it holds for any $\boldsymbol{A}, \boldsymbol{B} \in \mathbb{R}^{m \times n}$ that

$$\|\boldsymbol{P}\boldsymbol{P}^\top\boldsymbol{A} + (\boldsymbol{I} - \boldsymbol{P}\boldsymbol{P}^\top)\boldsymbol{B}\|_F^2 = \|\boldsymbol{P}\boldsymbol{P}^\top\boldsymbol{A}\|_F^2 + \|(\boldsymbol{I} - \boldsymbol{P}\boldsymbol{P}^\top)\boldsymbol{B}\|_F^2. \tag{9}$$

*Proof.* By definition we have $\boldsymbol{P}^\top\boldsymbol{P} = \boldsymbol{I}$. It suffices to note that

$$\langle \boldsymbol{P}\boldsymbol{P}^\top\boldsymbol{A}, (\boldsymbol{I} - \boldsymbol{P}\boldsymbol{P}^\top)\boldsymbol{B} \rangle_F = \mathrm{tr}(\boldsymbol{A}^\top\boldsymbol{P}\boldsymbol{P}^\top(\boldsymbol{I} - \boldsymbol{P}\boldsymbol{P}^\top)\boldsymbol{B}) = \mathrm{tr}(0) = 0.$$

$\square$

**Lemma 4** (Descent lemma)**:** Under Assumption 2, for update

$$\boldsymbol{x}^{(t+1)} = \boldsymbol{x}^{(t)} - \eta\tilde{\boldsymbol{m}}^{(t)},$$

it holds that

$$\begin{aligned} f(\boldsymbol{x}^{(t+1)}) \leq & f(\boldsymbol{x}^{(t)}) - \left(\frac{1}{2\eta} - \frac{L}{2}\right)\|\boldsymbol{x}^{(t+1)} - \boldsymbol{x}^{(t)}\|_2^2 + \frac{\eta}{2}\|\tilde{\boldsymbol{m}}^{(t)} - \nabla f(\boldsymbol{x}^{(t)})\|_2^2 \\ & - \frac{\eta}{2}\|\nabla f(\boldsymbol{x}^{(t)})\|_2^2. \end{aligned} \tag{10}$$

*Proof.* By $L$-smoothness of $f$ (Assumption 2) we have

$$\begin{aligned} & f(\boldsymbol{x}^{(t+1)}) - f(\boldsymbol{x}^{(t)}) \\ \leq & \langle \nabla f(\boldsymbol{x}^{(t)}), \boldsymbol{x}^{(t+1)} - \boldsymbol{x}^{(t)} \rangle + \frac{L}{2}\|\boldsymbol{x}^{(t+1)} - \boldsymbol{x}^{(t)}\|_2^2 \\ = & \left\langle \frac{\tilde{\boldsymbol{m}}^{(t)}}{2}, \boldsymbol{x}^{(t+1)} - \boldsymbol{x}^{(t)} \right\rangle + \left\langle \nabla f(\boldsymbol{x}^{(t)}) - \frac{\tilde{\boldsymbol{m}}^{(t)}}{2}, \boldsymbol{x}^{(t+1)} - \boldsymbol{x}^{(t)} \right\rangle + \frac{L}{2}\|\boldsymbol{x}^{(t+1)} - \boldsymbol{x}^{(t)}\|_2^2 \\ = & -\left(\frac{1}{2\eta} - \frac{L}{2}\right)\|\boldsymbol{x}^{(t+1)} - \boldsymbol{x}^{(t)}\|_2^2 + \frac{\eta}{2}\|\nabla f(\boldsymbol{x}^{(t)}) - \tilde{\boldsymbol{m}}^{(t)}\|_2^2 - \frac{\eta}{2}\|\nabla f(\boldsymbol{x}^{(t)})\|_2^2, \end{aligned}$$

which is exactly (10). $\square$

**Lemma 5** (Error of GoLore's projection)**:** Let $\boldsymbol{P} \sim \mathcal{U}(\mathrm{St}_{m,r})$, $\boldsymbol{Q} \sim \mathcal{U}(\mathrm{St}_{n,r})$, it holds for all $\boldsymbol{G} \in \mathbb{R}^{m \times n}$ that

$$\mathbb{E}[\boldsymbol{P}\boldsymbol{P}^\top] = \frac{r}{m} \cdot \boldsymbol{I}, \quad \mathbb{E}[\boldsymbol{Q}\boldsymbol{Q}^\top] = \frac{r}{n} \cdot \boldsymbol{I}, \tag{11}$$

and

$$\mathbb{E}[\|\boldsymbol{P}\boldsymbol{P}^\top\boldsymbol{G} - \boldsymbol{G}\|_F^2] = \left(1 - \frac{r}{m}\right)\|\boldsymbol{G}\|_F^2, \quad \mathbb{E}[\|\boldsymbol{G}\boldsymbol{Q}\boldsymbol{Q}^\top - \boldsymbol{G}\|_F^2] = \left(1 - \frac{r}{n}\right)\|\boldsymbol{G}\|_F^2. \tag{12}$$

*Proof.* We refer the proof of (11) to Theorem 2.2.2 in Chikuse (2012). By $\boldsymbol{P}^\top\boldsymbol{P} = \boldsymbol{I}$, we have

$$\begin{aligned} \mathbb{E}[\|\boldsymbol{P}\boldsymbol{P}^\top\boldsymbol{G} - \boldsymbol{G}\|_F^2] = & \mathbb{E}[\mathrm{tr}(\boldsymbol{G}^\top(\boldsymbol{I} - \boldsymbol{P}\boldsymbol{P}^\top)^2\boldsymbol{G})] \\ = & \mathbb{E}[\mathrm{tr}(\boldsymbol{G}^\top(\boldsymbol{I} - \boldsymbol{P}\boldsymbol{P}^\top)\boldsymbol{G})] \\ = & \mathrm{tr}(\boldsymbol{G}^\top(\boldsymbol{I} - \mathbb{E}[\boldsymbol{P}\boldsymbol{P}^\top])\boldsymbol{G}). \end{aligned} \tag{13}$$

Applying (11) to (13) yields

$$\begin{aligned} \mathbb{E}[\|\boldsymbol{P}\boldsymbol{P}^\top\boldsymbol{G} - \boldsymbol{G}\|_F^2] = & \mathrm{tr}\left(\boldsymbol{G}^\top\left(\boldsymbol{I} - \frac{r}{m}\boldsymbol{I}\right)\boldsymbol{G}\right) \\ = & \left(1 - \frac{r}{m}\right)\mathrm{tr}(\boldsymbol{G}^\top\boldsymbol{G}) \\ = & \left(1 - \frac{r}{m}\right)\|\boldsymbol{G}\|_F^2. \end{aligned}$$

The other part of (12) can be proved similarly. $\square$

**B.2. Non-convergence of GaLore**

In this subsection, we present the proof for Theorem 4. We first restate Theorem 4 as follows:

**Theorem 9** (Non-convergence of GaLore)**:** There exists an objective function $f : \mathbb{R}^d \to \mathbb{R}$ satisfying Assumptions 1, 2, a stochastic gradient oracle $(F, \mathcal{D})$ satisfying Assumption 3, an initial point $\boldsymbol{x}^{(0)} \in \mathbb{R}^d$, a constant $\epsilon_0 > 0$ such that for GaLore with any rank $r_\ell < \min\{m_\ell, n_\ell\}$, subspace changing frequency $\tau$, any subspace optimizer $\rho$ with arbitrary hyperparameters and any $t > 0$, it holds that

$$\|\nabla f(\boldsymbol{x}^{(t)})\|_2^2 \geq \epsilon_0.$$

*Proof.* Consider target function $f(\boldsymbol{X}) = \frac{L}{2}\mathrm{tr}(\boldsymbol{X}^\top \boldsymbol{pp}^\top \boldsymbol{X})$ where $L > 0$, $\boldsymbol{X} \in \mathbb{R}^{n \times n}$ with $n > 1$ and $\boldsymbol{p} = (1, 0, \cdots, 0)^\top \in \mathbb{R}^n$. It holds that

$$f(\boldsymbol{X}) = \frac{L}{2}\|\boldsymbol{p}^\top \boldsymbol{X}\|_2^2 \geq 0,$$

thus $f$ satisfies Assumption 1. Since $\nabla f(\boldsymbol{X}) = L\boldsymbol{pp}^\top \boldsymbol{X}$, it holds that

$$\|\nabla f(\boldsymbol{X}) - \nabla f(\boldsymbol{Y})\|_F = L\|\boldsymbol{pp}^\top(\boldsymbol{X} - \boldsymbol{Y})\|_F \leq L\|\boldsymbol{pp}^\top\|_2\|\boldsymbol{X} - \boldsymbol{Y}\|_F = L\|\boldsymbol{X} - \boldsymbol{Y}\|_F,$$

thus $f$ satisfies Assumption 2.

Consider the following stochastic gradient oracle:

$$F(\boldsymbol{X}; \xi) = f(\boldsymbol{X}) + \xi\tilde{\sigma} \cdot \mathrm{tr}(\boldsymbol{QQ}^\top \boldsymbol{X}), \quad \text{and} \quad \mathbb{P}_{\xi \sim \mathcal{D}}[\xi = 1] = \mathbb{P}_{\xi \sim \mathcal{D}}[\xi = -1] = 0.5,$$

where $\tilde{\sigma} = \sigma/\sqrt{(n-1)n/2}$ and

$$\boldsymbol{Q} = \begin{pmatrix} 0 \\ \mathrm{diag}\left(1, \sqrt[4]{2}, \cdots, \sqrt[4]{n-1}\right) \end{pmatrix} \in \mathbb{R}^{n \times (n-1)}.$$

Note that $\nabla F(\boldsymbol{X}; \xi) = \nabla f(\boldsymbol{X}) + \xi\tilde{\sigma}\boldsymbol{QQ}^\top$, it holds for any $\boldsymbol{X} \in \mathbb{R}^{n \times n}$ that

$$\mathbb{E}_{\xi \sim \mathcal{D}}[\nabla F(\boldsymbol{X}; \xi)] = \nabla f(\boldsymbol{X})$$

$$\mathbb{E}_{\xi \sim \mathcal{D}}[\|\nabla F(\boldsymbol{X}; \xi) - \nabla f(\boldsymbol{X})\|_F^2] = \tilde{\sigma}^2\|\boldsymbol{QQ}^\top\|_F^2 = \frac{\sigma^2}{(n-1)n/2} \cdot \sum_{i=1}^{n-1} i = \sigma^2,$$

thus oracle $(F, \mathcal{D})$ satisfies Assumption 3.

Consider the following initial point:

$$\boldsymbol{X}^{(0)} = \begin{pmatrix} \lambda\boldsymbol{p}^\top \\ \boldsymbol{\Lambda} \end{pmatrix},$$

where $0 < \lambda < \tilde{\sigma}/L$ is a scalar and $\boldsymbol{\Lambda} \in \mathbb{R}^{(n-1) \times n}$ is an arbitrary matrix. We show that GaLore with the above objective function $f$, stochastic gradient oracle $(F, \mathcal{D})$, initial point $\boldsymbol{X}^{(0)}$, arbitrary rank $0 < r < n$, arbitrary subspace changing frequency $\tau$ and arbitrary subspace optimizer $\rho$, can only output points $\boldsymbol{X}^{(t)}$ with $\|\nabla f(\boldsymbol{X}^{(t)})\|_F^2 \geq \epsilon_0$ for $\epsilon_0 = L^2\lambda^2 > 0$.

When $\tau \mid t$, GaLore recomputes the subspace projection matrix at iteration $t$. If the first row of $\boldsymbol{X}^{(t)}$ equals $\lambda\boldsymbol{p}^\top$, *i.e.*, $\boldsymbol{X}^{(t)}[1, :] = \lambda\boldsymbol{p}^\top$, the stochastic gradient is given by

$$\boldsymbol{G}^{(t)} = L\boldsymbol{pp}^\top \boldsymbol{X} + \xi^{(t)}\tilde{\sigma}\boldsymbol{QQ}^\top = \mathrm{diag}\left(L\lambda, \xi^{(t)}\tilde{\sigma}, \sqrt{2}\xi^{(t)}\tilde{\sigma}, \cdots, \sqrt{n-1}\xi^{(t)}\tilde{\sigma}\right).$$

since $L\lambda < \tilde{\sigma}$, computing SVD yields

$$\boldsymbol{G}^{(t)} = \begin{pmatrix} L\lambda & 0 & \cdots & 0 \\ 0 & \xi^{(t)}\tilde{\sigma} & \cdots & 0 \\ \vdots & \vdots & \ddots & \vdots \\ 0 & 0 & \cdots & \sqrt{n-1}\xi^{(t)}\tilde{\sigma} \end{pmatrix}$$

$$= \underbrace{\begin{pmatrix} 0 & \cdots & 0 & \zeta_1 \\ 0 & \cdots & \zeta_2 & 0 \\ \vdots & \ddots & \vdots & \vdots \\ \zeta_n & \cdots & 0 & 0 \end{pmatrix}}_{:=U} \underbrace{\begin{pmatrix} \sqrt{n-1}\tilde{\sigma} & \cdots & 0 & 0 \\ \vdots & \ddots & \vdots & \vdots \\ 0 & \cdots & \tilde{\sigma} & 0 \\ 0 & \cdots & 0 & L\lambda \end{pmatrix}}_{:=\Sigma} \underbrace{\begin{pmatrix} 0 & 0 & \cdots & \zeta_n \xi^{(t)} \\ \vdots & \vdots & \ddots & \vdots \\ 0 & \zeta_2 \xi^{(t)} & \cdots & 0 \\ \zeta_1 & 0 & \cdots & 0 \end{pmatrix}}_{:=V^\top},$$

where $\zeta_1, \cdots, \zeta_n \in \{-1, 1\}$. For any rank $r < n$, the projection matrix is thus

$$\boldsymbol{P}^{(t)} = \begin{pmatrix} 0 & 0 & \cdots & 0 \\ \vdots & \vdots & \ddots & \vdots \\ 0 & 0 & \cdots & 0 \\ 0 & 0 & \cdots & \zeta_{n-r+1} \\ \vdots & \vdots & \ddots & \vdots \\ 0 & \zeta_{n-1} & \cdots & 0 \\ \zeta_n & 0 & \cdots & 0 \end{pmatrix} \in \mathbb{R}^{n \times r}.$$

Using this projection matrix, the subspace updates in the following $\tau$ iterations is as

$$\boldsymbol{X}^{(t+\Delta_t)} = \boldsymbol{X}^{(t)} + \boldsymbol{P}^{(t)} \sum_{s=0}^{\Delta_t - 1} \rho^{(t+s)}((\boldsymbol{P}^{(t)})^\top \boldsymbol{G}^{(t)}) \quad \Rightarrow \quad \boldsymbol{X}^{(t+\Delta_t)}[1,:] = \boldsymbol{X}^{(t)}[1,:] = \lambda \boldsymbol{p}^\top,$$

for $\Delta_t = 1, 2, \cdots, \tau$. Since $\boldsymbol{X}^{(0)}[1,:] = \lambda \boldsymbol{p}^\top$, it holds for all $t > 0$ that $\boldsymbol{X}^{(t)}[1,:] = \lambda \boldsymbol{p}^\top$ and thus

$$\|\nabla f(\boldsymbol{X}^{(t)})\|_F^2 = L^2 \lambda^2 = \epsilon_0.$$

$\square$

**Remark 6.** When setting $\boldsymbol{B} = 0$ in the quadratic problem setting (Sec. 6), the quadratic problem is equivalent to the counter-example we construct in the proof of Theorem 9. The illustration in Fig. 6 displays the loss curves for this problem.

### B.3. Convergence of deterministic GaLore

In this subsection, we present the proof for Theorem 5. GaLore using deterministic gradients and MSGD with MP is specified as Alg. 3.

**Lemma 6** (Momentum contraction): In deterministic GaLore using MSGD with MP (Alg. 3), if $0 < \beta_1 \le 1$, term $\tilde{\boldsymbol{M}}_\ell^{(t)}$ has the following contraction properties:

- When $t = 0$, it holds that

$$\|\tilde{\boldsymbol{M}}_\ell^{(0)} - \nabla_\ell f(\boldsymbol{X}^{(0)})\|_F^2 \le (\tau - 1)(1 - \delta_\ell \beta_1) \sum_{r=0}^{\tau-2} \|\nabla_\ell f(\boldsymbol{x}^{(r+1)}) - \nabla_\ell f(\boldsymbol{x}^{(r)})\|_F^2$$

$$+ \frac{2(1 - \delta_\ell \beta_1)}{\tau} \sum_{r=0}^{\tau-1} \|\nabla_\ell f(\boldsymbol{x}^{(r)})\|_F^2; \tag{14}$$

- When $t = k\tau$, $k \in \mathbb{N}^*$, it holds that

$$\|\tilde{\boldsymbol{M}}_\ell^{(t)} - \nabla_\ell f(\boldsymbol{x}^{(t)})\|_F^2 - \left(1 - \left(1 - \frac{\delta_\ell}{4}\right)\beta_1\right) \|\tilde{\boldsymbol{M}}_\ell^{(t-1)} - \nabla_\ell f(\boldsymbol{x}^{(t-1)})\|_F^2$$

$$\le \frac{2(1 - \delta_\ell)}{\tau} \sum_{r=0}^{\tau-1} \|\nabla_l f(\boldsymbol{x}^{(k\tau+r)})\|_F^2 + \frac{5(1 - \beta_1)}{\delta_\ell \beta_1} \|\nabla_\ell f(\boldsymbol{x}^{(t)}) - \nabla_\ell f(\boldsymbol{x}^{(t-1)})\|_F^2$$

$$+ (\tau - 1)(1 - \delta_\ell) \sum_{r=0}^{\tau-2} \|\nabla_\ell f(\boldsymbol{x}^{(k\tau+r+1)}) - \nabla_\ell f(\boldsymbol{x}^{(k\tau+r)})\|_F^2; \tag{15}$$

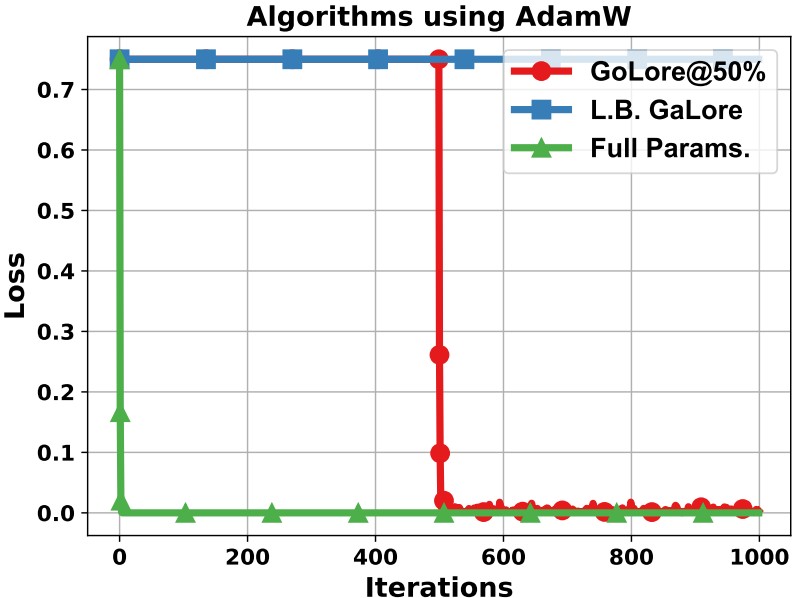

*Figure 6.* Loss curves of algorithms using AdamW. *GoLore@50%* uses GaLore in the first half and shifts to GoLore in the last half, *Full Params.* denotes full-parameter training.

- When $t = k\tau + r$, $k \in \mathbb{N}$, $1 \leq r < \tau$, it holds that

$$\|\tilde{\boldsymbol{M}}_\ell^{(t)} - \nabla_\ell f(\boldsymbol{x}^{(t)})\|_F^2 - \left(1 - \left(1 - \frac{\delta_\ell}{4}\right)\beta_1\right)\|\tilde{\boldsymbol{M}}_\ell^{(t-1)} - \nabla_\ell f(\boldsymbol{x}^{(t-1)})\|_F^2$$
$$\leq \left(1 - \frac{\delta_\ell}{2}\right)\beta_1\|\nabla_\ell f(\boldsymbol{x}^{(t)})\|_F^2 + \frac{5(1-\beta_1)}{\delta_\ell\beta_1}\|\nabla_\ell f(\boldsymbol{x}^{(t)}) - \nabla_\ell f(\boldsymbol{x}^{(t-1)})\|_F^2$$
$$+ \frac{10r\beta_1}{\delta_\ell}\sum_{i=1}^{r}\|\nabla_\ell f(\boldsymbol{x}^{(k\tau+i)}) - \nabla_\ell f(\boldsymbol{x}^{(k\tau+i-1)})\|_F^2. \tag{16}$$

*Proof.* Without loss of generality assume $m_\ell \leq n_\ell$ (the other case can be proved similarly). When $t = 0$, we have

$$\|\tilde{\boldsymbol{M}}_\ell^{(0)} - \nabla_\ell f(\boldsymbol{x}^{(0)})\|_F^2 = \|\beta_1(\boldsymbol{P}_\ell^{(0)}(\boldsymbol{P}_\ell^{(0)})^\top - \boldsymbol{I})\nabla_\ell f(\boldsymbol{x}^{(0)}) - (1-\beta_1)\nabla_\ell f(\boldsymbol{x}^{(0)})\|_F^2$$
$$\leq \beta_1(1-\delta_\ell)\|\nabla_\ell f(\boldsymbol{x}^{(0)})\|_F^2 + (1-\beta_1)\|\nabla_\ell f(\boldsymbol{x}^{(0)})\|_F^2$$
$$= (1 - \delta_\ell\beta_1)\|\nabla_\ell f(\boldsymbol{x}^{(0)})\|_F^2, \tag{17}$$

where the inequality uses Lemma 1 and Jensen's inequality. Applying Lemma 2 to (17) yields (14).

When $t = k\tau$, $k \in \mathbb{N}^*$, we have

$$\|\tilde{\boldsymbol{M}}_\ell^{(t)} - \nabla_\ell f(\boldsymbol{x}^{(t)})\|_F^2$$
$$= \|\boldsymbol{P}_\ell^{(t)}(\boldsymbol{P}_\ell^{(t)})^\top[(1-\beta_1)\tilde{\boldsymbol{M}}_\ell^{(t-1)} + \beta_1\boldsymbol{G}_\ell^{(t)} - \nabla_\ell f(\boldsymbol{x}^{(t)})] - (\boldsymbol{I} - \boldsymbol{P}_\ell^{(t)}(\boldsymbol{P}_\ell^{(t)})^\top)\nabla_\ell f(\boldsymbol{x}^{(t)})\|_F^2$$
$$= \|\boldsymbol{P}_\ell^{(t)}(\boldsymbol{P}_\ell^{(t)})^\top[(1-\beta_1)(\tilde{\boldsymbol{M}}_\ell^{(t-1)} - \nabla_\ell f(\boldsymbol{x}^{(t)}))]\|_F^2 + \|(\boldsymbol{I} - \boldsymbol{P}_\ell^{(t)}(\boldsymbol{P}_\ell^{(t)})^\top)\nabla_\ell f(\boldsymbol{x}^{(t)})\|_F^2$$
$$\leq \|(1-\beta_1)(\tilde{\boldsymbol{M}}_\ell^{(t-1)} - \nabla_\ell f(\boldsymbol{x}^{(t)}))\|_F^2 + (1-\delta_\ell)\|\nabla_\ell f(\boldsymbol{x}^{(t)})\|_F^2, \tag{18}$$

where the second equality uses Lemma 3 and $\boldsymbol{G}_\ell^{(t)} = \nabla_\ell f(\boldsymbol{x}^{(t)})$, the inequality uses Lemma 1 and $\|\boldsymbol{P}_\ell^{(t)}(\boldsymbol{P}_\ell^{(t)})^\top\|_2 = 1$. By Young's inequality, we have

$$\|\tilde{\boldsymbol{M}}_\ell^{(t-1)} - \nabla_\ell f(\boldsymbol{x}^{(t)})\|_F^2$$
$$= \|(\tilde{\boldsymbol{M}}_\ell^{(t-1)} - \nabla_\ell f(\boldsymbol{x}^{(t-1)})) - (\nabla_\ell f(\boldsymbol{x}^{(t)}) - \nabla_\ell f(\boldsymbol{x}^{(t-1)}))\|_F^2$$

---

**Algorithm 3** GaLore using deterministic gradients and MSGD with MP

---

**Input:** Initial point $\boldsymbol{x}^{(0)}$, learning rate $\eta$, subspace changing frequency $\tau$, rank $\{r_\ell\}_{\ell=1}^{N_L}$, momentum parameter $\beta_1$.
**Output:** $\{\boldsymbol{x}^{(t)}\}_{t=0}^T$.

  Initialize optimizer state $\{\boldsymbol{M}_\ell^{(-1)}\}_{\ell=1}^{N_L}$ to zero;
  **for** $t = 0, 1, \cdots, T-1$ **do**
    **for** $\ell = 1, 2, \cdots, N_L$ **do**
      $\boldsymbol{G}_\ell^{(t)} \leftarrow \nabla_\ell f(\boldsymbol{x}^{(t)})$;
      **if** $t \equiv 0 \pmod{\tau}$ **then**
        $\boldsymbol{U}, \boldsymbol{\Sigma}, \boldsymbol{V} \leftarrow \mathrm{SVD}(\boldsymbol{G}_\ell^{(t)})$;
        **if** $m_\ell \leq n_\ell$ **then**
          $\boldsymbol{P}_\ell^{(t)} \leftarrow \boldsymbol{U}[:, :r_\ell]$;
          $\boldsymbol{M}_\ell^{(t)} \leftarrow (1-\beta_1)(\boldsymbol{P}_\ell^{(t)})^\top \boldsymbol{P}_\ell^{(t-1)} \boldsymbol{M}_\ell^{(t-1)} + \beta_1 (\boldsymbol{P}_\ell^{(t)})^\top \boldsymbol{G}_\ell^{(t)}$;
          $\boldsymbol{X}_\ell^{(t+1)} \leftarrow \boldsymbol{X}_\ell^{(t)} - \eta \boldsymbol{P}_\ell^{(t)} \boldsymbol{M}_\ell^{(t)}$;
        **else**
          $\boldsymbol{Q}_\ell^{(t)} \leftarrow \boldsymbol{V}[:, :r_\ell]$;
          $\boldsymbol{M}_\ell^{(t)} \leftarrow (1-\beta_1) \boldsymbol{M}_\ell^{(t-1)} (\boldsymbol{Q}_\ell^{(t-1)})^\top \boldsymbol{Q}_\ell^{(t)} + \beta_1 \boldsymbol{G}_\ell^{(t)} \boldsymbol{Q}_\ell^{(t)}$;
          $\boldsymbol{X}_\ell^{(t+1)} \leftarrow \boldsymbol{X}_\ell^{(t)} - \eta \boldsymbol{M}_\ell^{(t)} (\boldsymbol{Q}_\ell^{(t)})^\top$;
        **end if**
      **else**
        **if** $m_\ell \leq n_\ell$ **then**
          $\boldsymbol{P}_\ell^{(t)} \leftarrow \boldsymbol{P}_\ell^{(t-1)}$;
          $\boldsymbol{M}_\ell^{(t)} \leftarrow (1-\beta_1) \boldsymbol{M}_\ell^{(t-1)} + \beta_1 (\boldsymbol{P}_\ell^{(t)})^\top \boldsymbol{G}_\ell^{(t)}$;
          $\boldsymbol{X}_\ell^{(t+1)} \leftarrow \boldsymbol{X}_\ell^{(t)} - \eta \boldsymbol{P}_\ell^{(t)} \boldsymbol{M}_\ell^{(t)}$;
        **else**
          $\boldsymbol{Q}_\ell^{(t)} \leftarrow \boldsymbol{Q}_\ell^{(t-1)}$;
          $\boldsymbol{M}_\ell^{(t)} \leftarrow (1-\beta_1) \boldsymbol{M}_\ell^{(t-1)} + \beta_1 \boldsymbol{G}_\ell^{(t)} \boldsymbol{Q}_\ell^{(t)}$;
          $\boldsymbol{X}_\ell^{(t+1)} \leftarrow \boldsymbol{X}_\ell^{(t)} - \eta \boldsymbol{M}_\ell^{(t)} (\boldsymbol{Q}_\ell^{(t)})^\top$;
        **end if**
      **end if**
    **end for**
  **end for**

---

$$\leq \left(1 + \frac{\delta_\ell \beta_1}{4}\right) \|\tilde{\boldsymbol{M}}_\ell^{(t-1)} - \nabla_\ell f(\boldsymbol{x}^{(t-1)})\|_F^2 + \left(1 + \frac{4}{\delta_\ell \beta_1}\right) \|\nabla_\ell f(\boldsymbol{x}^{(t)}) - \nabla_\ell f(\boldsymbol{x}^{(t-1)})\|_F^2. \tag{19}$$

Applying Lemma 2 and (19) to (18) yields (15).

When $t = k\tau + r$, $k \in \mathbb{N}$, $1 \leq r < \tau$, we have

$$\|\tilde{\boldsymbol{M}}_\ell^{(t)} - \nabla_\ell f(\boldsymbol{x}^{(t)})\|_F^2$$
$$= \|(1-\beta_1)(\tilde{\boldsymbol{M}}_\ell^{(t-1)} - \nabla_\ell f(\boldsymbol{x}^{(t)})) + \beta_1 (\boldsymbol{P}_\ell^{(t)} (\boldsymbol{P}_\ell^{(t)})^\top - \boldsymbol{I}) \nabla_\ell f(\boldsymbol{x}^{(t)})\|_F^2$$
$$\leq (1-\beta_1) \|\tilde{\boldsymbol{M}}_\ell^{(t-1)} - \nabla_\ell f(\boldsymbol{x}^{(t)})\|_F^2 + \beta_1 \|(\boldsymbol{I} - \boldsymbol{P}_\ell^{(k\tau)} (\boldsymbol{P}_\ell^{(k\tau)})^\top) \nabla_\ell f(\boldsymbol{x}^{(t)})\|_F^2, \tag{20}$$

where the inequality uses Jensen's inequality and $\boldsymbol{P}_\ell^{(t)} = \boldsymbol{P}_\ell^{(t-1)} = \cdots = \boldsymbol{P}_\ell^{(k\tau)}$. The first term can be similarly upper bounded as (19). For the second term, we have

$$(\boldsymbol{I} - \boldsymbol{P}_\ell^{(k\tau)} (\boldsymbol{P}_\ell^{(k\tau)})^\top) \nabla_\ell f(\boldsymbol{x}^{(t)})\|_F^2$$
$$\leq \left(1 + \frac{\delta_\ell}{4}\right) \|(\boldsymbol{I} - \boldsymbol{P}_\ell^{(k\tau)} (\boldsymbol{P}_\ell^{(k\tau)})^\top) \nabla_\ell f(\boldsymbol{x}^{(k\tau)})\|_F^2$$
$$+ \left(1 + \frac{4}{\delta_\ell}\right) \|(\boldsymbol{I} - \boldsymbol{P}_\ell^{(k\tau)} (\boldsymbol{P}_\ell^{(k\tau)})^\top)(\nabla_\ell f(\boldsymbol{x}^{(t)}) - \nabla_\ell f(\boldsymbol{x}^{(k\tau)})\|_F^2$$
$$\leq \left(1 + \frac{\delta_\ell}{4}\right) (1 - \delta_\ell) \|\nabla_\ell f(\boldsymbol{x}^{(k\tau)})\|_F^2 + \frac{5}{\delta_\ell} \|\nabla_\ell f(\boldsymbol{x}^{(t)}) - \nabla_\ell f(\boldsymbol{x}^{(k\tau)})\|_F^2, \tag{21}$$

where the first inequality uses Young's inequality and the second inequality uses Lemma 1. By Young's inequality, we have

$$\|\nabla_\ell f(\boldsymbol{x}^{(k\tau)})\|_F^2 \le \left(1 + \frac{\delta_\ell}{4}\right)\|\nabla_\ell f(\boldsymbol{x}^{(t)})\|_F^2 + \left(1 + \frac{4}{\delta_\ell}\right)\|\nabla_\ell f(\boldsymbol{x}^{(t)}) - \nabla_\ell f(\boldsymbol{x}^{(k\tau)})\|_F^2. \tag{22}$$

Note that $t = k\tau + r$, we further have

$$\|\nabla_\ell f(\boldsymbol{x}^{(t)}) - \nabla_\ell f(\boldsymbol{x}^{(k\tau)})\|_F^2 = \left\|\sum_{i=1}^r \nabla_\ell f(\boldsymbol{x}^{(k\tau+i)}) - \nabla_\ell f(\boldsymbol{x}^{(k\tau+i-1)})\right\|_F^2$$
$$\le r \sum_{i=1}^r \|\nabla_\ell f(\boldsymbol{x}^{(k\tau+i)}) - \nabla_\ell f(\boldsymbol{x}^{(k\tau+i-1)})\|_F^2, \tag{23}$$

where the inequality uses Cauchy's inequality. Applying (22)(23) to (21) yields

$$(\boldsymbol{I} - \boldsymbol{P}_\ell^{(k\tau)}(\boldsymbol{P}_\ell^{(k\tau)})^\top)\nabla_\ell f(\boldsymbol{x}^{(t)})\|_F^2$$
$$\le \left(1 - \frac{\delta_\ell}{2}\right)\|\nabla_\ell f(\boldsymbol{x}^{(t)})\|_F^2 + \frac{10r}{\delta_\ell}\sum_{i=1}^r \|\nabla_\ell f(\boldsymbol{x}^{(k\tau+i)}) - \nabla_\ell f(\boldsymbol{x}^{(k\tau+i-1)})\|_F^2. \tag{24}$$

Applying (19)(24) to (20) yields (16). $\qquad\square$

**Lemma 7** (Momentum error)**:** Under Assumption 2, if $0 < \beta_1 \le 1$ in deterministic GaLore using MSGD and MP (Alg. 3), it holds for any $K \ge 1$ that

$$\sum_{t=0}^{K\tau-1} \|\tilde{\boldsymbol{m}}^{(t)} - \nabla f(\boldsymbol{x}^{(t)})\|_2^2$$
$$\le \left(\frac{5(1-\beta_1)}{(1-\underline{\delta}/4)\underline{\delta}\beta_1^2} + \frac{5\tau(\tau-1)}{(1-\underline{\delta}/4)\underline{\delta}} + \frac{\tau-1}{(1-\overline{\delta}/4)\beta_1}\right) L^2 \sum_{t=0}^{K\tau-2} \|\boldsymbol{x}^{(t+1)} - \boldsymbol{x}^{(t)}\|_2^2$$
$$+ \left(\frac{1-\underline{\delta}/2}{1-\underline{\delta}/4} + \frac{2}{(1-\overline{\delta}/4)\tau\beta_1}\right) \sum_{t=0}^{K\tau-1} \|\nabla f(\boldsymbol{x}^{(t)})\|_2^2. \tag{25}$$

*Proof.* By Lemma 6 we have

$$\sum_{t=0}^{K\tau-1} \|\tilde{\boldsymbol{M}}_\ell^{(t)} - \nabla_\ell f(\boldsymbol{x}^{(t)})\|_F^2 - \left(1 - \left(1 - \frac{\delta_\ell}{4}\right)\beta_1\right)\sum_{t=0}^{K\tau-2} \|\tilde{\boldsymbol{M}}_\ell^{(t)} - \nabla_\ell f(\boldsymbol{x}^{(t)})\|_F^2$$
$$\le \left(\frac{5(1-\beta_1)}{\delta_\ell\beta_1} + \frac{5\tau(\tau-1)\beta_1}{\delta_\ell} + (\tau-1)\right)\sum_{t=0}^{K\tau-2} \|\nabla_\ell f(\boldsymbol{x}^{(t+1)}) - \nabla_\ell f(\boldsymbol{x}^{(t)})\|_F^2$$
$$+ \left(\frac{2}{\tau} + \left(1 - \frac{\delta_\ell}{2}\right)\beta_1\right)\sum_{t=0}^{K\tau-1} \|\nabla_\ell f(\boldsymbol{x}^{(t)})\|_F^2,$$

which implies

$$\sum_{t=0}^{K\tau-1} \|\tilde{\boldsymbol{M}}_\ell^{(t)} - \nabla_\ell f(\boldsymbol{x}^{(t)})\|_F^2$$
$$\le \left(\frac{5(1-\beta_1)}{(1-\delta_\ell/4)\delta_\ell\beta_1^2} + \frac{5\tau(\tau-1)}{(1-\delta_\ell/4)\delta_\ell} + \frac{\tau-1}{(1-\delta_\ell/4)\beta_1}\right)\sum_{t=0}^{K\tau-2} \|\nabla_\ell f(\boldsymbol{x}^{(t+1)}) - \nabla_\ell f(\boldsymbol{x}^{(t)})\|_F^2$$
$$+ \left(\frac{1-\delta_\ell/2}{1-\delta_\ell/4} + \frac{2}{(1-\delta_\ell/4)\tau\beta_1}\right)\sum_{t=0}^{K\tau-1} \|\nabla_\ell f(\boldsymbol{x}^{(t)})\|_F^2. \tag{26}$$

Summing (26) for $\ell = 1, \cdots, N_L$ and applying Assumption 2 yields (25). $\qquad\square$

Now we are ready to prove the convergence of Alg. 3.

**Theorem 10** (Convergence of deterministic GaLore): Under Assumptions 1-2, if hyperparameters

$$0 < \beta_1 \leq 1, \quad \tau \geq \frac{64}{3\beta_1 \underline{\delta}}, \quad 0 < \eta \leq \min\left\{\frac{1}{4L}, \sqrt{\frac{3\underline{\delta}\beta_1^2}{80L^2}}, \sqrt{\frac{3\underline{\delta}}{80\tau^2 L^2}}, \sqrt{\frac{3\beta_1}{16\tau L^2}}\right\}, \tag{27}$$

GaLore using deterministic gradients and MSGD with MP (Alg. 3) converges as

$$\frac{1}{K\tau} \sum_{t=0}^{K\tau-1} \|\nabla f(\boldsymbol{x}^{(t)})\|_2^2 \leq \frac{16\Delta}{\underline{\delta}\eta K\tau} \tag{28}$$

for any $K \geq 1$, where $\Delta = f(\boldsymbol{x}^{(0)}) - \inf_{\boldsymbol{x}} f(\boldsymbol{x})$.

*Proof.* By Lemma 4 we have

$$\sum_{t=0}^{K\tau-1} \|\nabla f(\boldsymbol{x}^{(t)})\|_2^2 \leq \frac{2[f(\boldsymbol{x}^{(0)}) - f(\boldsymbol{x}^{(K\tau)})]}{\eta} + \sum_{t=0}^{K\tau-1} \|\tilde{\boldsymbol{m}}^{(t)} - \nabla f(\boldsymbol{x}^{(t)})\|_2^2$$

$$- \left(\frac{1}{\eta^2} - \frac{L}{\eta}\right) \sum_{t=0}^{K\tau-1} \|\boldsymbol{x}^{(t+1)} - \boldsymbol{x}^{(t)}\|_2^2. \tag{29}$$

Applying Lemma 7 to (29) and using $\underline{\delta} \leq \overline{\delta} < 1$ yields

$$\left(\frac{\delta}{4} - \frac{8}{3\tau\beta_1}\right) \sum_{t=0}^{K\tau-1} \|\nabla f(\boldsymbol{x}^{(t)})\|_2^2$$

$$\leq \frac{2}{\eta} f(\boldsymbol{x}^{(0)}) - f(\boldsymbol{x}^{(K\tau)})$$

$$- \left(\frac{1}{\eta^2} - \frac{L}{\eta} - \frac{20(1-\beta_1)L^2}{3\underline{\delta}\beta_1^2} - \frac{20\tau(\tau-1)L^2}{3\underline{\delta}} - \frac{4(\tau-1)L^2}{3\beta_1}\right) \sum_{t=0}^{K\tau-1} \|\boldsymbol{x}^{(t+1)} - \boldsymbol{x}^{(t)}\|_2^2. \tag{30}$$

By (27) we have

$$\frac{\delta}{4} - \frac{8}{3\tau\beta_1} \geq \frac{\delta}{8}, \quad \text{and} \quad \frac{1}{4\eta^2} \geq \max\left\{\frac{L}{\eta}, \frac{20(1-\beta_1)L^2}{3\underline{\delta}\beta_1^2}, \frac{20\tau(\tau-1)L^2}{3\underline{\delta}}, \frac{4(\tau-1)L^2}{3\beta_1}\right\}. \tag{31}$$

Applying (31) to (30) yields (28). □

We now prove Theorem 5, which is restated as follows.

**Corollary 11** (Convergence complexity of deterministic GaLore). *Under Assumptions 1-2, if $T \geq 64/(3\underline{\delta})$ and we choose*

$$\beta_1 = 1$$

$$\tau = \left\lceil \frac{64}{3\underline{\delta}\beta_1} \right\rceil$$

$$\eta = \left(4L + \sqrt{\frac{80L^2}{3\underline{\delta}\beta_1^2}} + \sqrt{\frac{80\tau^2 L^2}{3\underline{\delta}}} + \sqrt{\frac{16\tau L^2}{3\beta_1}}\right)^{-1},$$

*GaLore using deterministic gradients and MSGD with MP (Alg. 3) converges as*

$$\frac{1}{T} \sum_{t=0}^{T-1} \|\nabla f(\boldsymbol{x}^{(t)})\|_2^2 = \mathcal{O}\left(\frac{L\Delta}{\underline{\delta}^{5/2} T}\right), \tag{32}$$

*where $\Delta = f(\boldsymbol{x}^{(0)}) - \inf_{\boldsymbol{x}} f(\boldsymbol{x})$. Consequently, the computation complexity to reach an $\varepsilon$-accurate solution $\boldsymbol{x}$ such that $\|\nabla f(\boldsymbol{x})\|_2^2 \leq \varepsilon$ is $\mathcal{O}\left(\frac{L\Delta}{\underline{\delta}^{5/2}\varepsilon} + \frac{1}{\underline{\delta}}\right)$.*

*Proof.* $T \geq 1 + 64/(3\underline{\delta})$ guarantees $T \geq \tau$. Let $T = K\tau + r$, where $K \in \mathbb{N}^*$ and $0 \leq r < \tau$. If $r = 0$, (32) is a direct result of Theorem 10. If $r > 0$, applying Theorem 10 to $\tilde{K} := K + 1$ yields

$$\frac{1}{T} \sum_{t=0}^{T-1} \|\nabla f(\boldsymbol{x}^{(t)})\|_2^2 \leq \frac{\tilde{K}\tau}{T} \cdot \frac{1}{\tilde{K}\tau} \sum_{t=0}^{\tilde{K}\tau-1} \|\nabla f(\boldsymbol{x}^{(t)})\|_2^2 = \mathcal{O}\left(\frac{L\Delta}{\underline{\delta}^{5/2}T}\right).$$

$\square$

### B.4. Convergence of large-batch GaLore

In this subsection, we present the proof for Theorem 6. GaLore using large-batch stochastic gradients and MSGD with MP is specified as Alg. 4.

**Lemma 8** (Momentum contraction)**:** Under Assumption 3, in large-batch GaLore using MSGD with MP (Alg. 4), if $0 < \beta_1 \leq 1$, term $\tilde{\boldsymbol{M}}_\ell^{(t)}$ has the following contraction properties:

- When $t = 0$, it holds that

$$\mathbb{E}[\|\tilde{\boldsymbol{M}}_\ell^{(0)} - \nabla_\ell f(\boldsymbol{X}^{(0)})\|_F^2] \leq 2(\tau-1)(1-\delta_\ell\beta_1) \sum_{r=0}^{\tau-2} \mathbb{E}[\|\nabla_\ell f(\boldsymbol{x}^{(r+1)}) - \nabla_\ell f(\boldsymbol{x}^{(r)})\|_F^2]$$

$$+ \frac{4(1-\delta_\ell\beta_1)}{\tau} \sum_{r=0}^{\tau-1} \mathbb{E}[\|\nabla_\ell f(\boldsymbol{x}^{(r)})\|_F^2] + \frac{4\beta_1\sigma_\ell^2}{\mathcal{B}}; \tag{33}$$

- When $t = k\tau$, $k \in \mathbb{N}^*$, it holds that

$$\mathbb{E}[\|\tilde{\boldsymbol{M}}_\ell^{(t)} - \nabla_\ell f(\boldsymbol{x}^{(t)})\|_F^2] - \left(1 - \left(1 - \frac{\delta_\ell}{4}\right)\beta_1\right) \mathbb{E}[\|\tilde{\boldsymbol{M}}_\ell^{(t-1)} - \nabla_\ell f(\boldsymbol{x}^{(t-1)})\|_F^2]$$

$$\leq \frac{4(1-\delta_\ell)}{\tau} \sum_{r=0}^{\tau-1} \mathbb{E}[\|\nabla_l f(\boldsymbol{x}^{(k\tau+r)})\|_F^2] + \frac{5(1-\beta_1)}{\delta_\ell\beta_1} \mathbb{E}[\|\nabla_\ell f(\boldsymbol{x}^{(t)}) - \nabla_\ell f(\boldsymbol{x}^{(t-1)})\|_F^2]$$

$$+ 2(\tau-1)(1-\delta_\ell) \sum_{r=0}^{\tau-2} \mathbb{E}[\|\nabla_\ell f(\boldsymbol{x}^{(k\tau+r+1)}) - \nabla_\ell f(\boldsymbol{x}^{(k\tau+r)})\|_F^2] + \frac{5\sigma_\ell^2}{\mathcal{B}}; \tag{34}$$

- When $t = k\tau + r$, $k \in \mathbb{N}$, $1 \leq r < \tau$, it holds that

$$\mathbb{E}[\|\tilde{\boldsymbol{M}}_\ell^{(t)} - \nabla_\ell f(\boldsymbol{x}^{(t)})\|_F^2] - \left(1 - \left(1 - \frac{\delta_\ell}{4}\right)\beta_1\right) \mathbb{E}[\|\tilde{\boldsymbol{M}}_\ell^{(t-1)} - \nabla_\ell f(\boldsymbol{x}^{(t-1)})\|_F^2]$$

$$\leq \left(1 - \frac{\delta_\ell}{2}\right)\beta_1 \mathbb{E}[\|\nabla_\ell f(\boldsymbol{x}^{(t)})\|_F^2] + \frac{5(1-\beta_1)}{\delta_\ell\beta_1} \mathbb{E}[\|\nabla_\ell f(\boldsymbol{x}^{(t)}) - \nabla_\ell f(\boldsymbol{x}^{(t-1)})\|_F^2]$$

$$+ \frac{15r\beta_1}{\delta_\ell} \sum_{i=1}^{r} \mathbb{E}[\|\nabla_\ell f(\boldsymbol{x}^{(k\tau+i)}) - \nabla_\ell f(\boldsymbol{x}^{(k\tau+i-1)})\|_F^2] + \left(\frac{11\beta_1}{\delta_\ell\mathcal{B}} + \beta_1^2\right)\sigma_\ell^2. \tag{35}$$

*Proof.* Without loss of generality assume $m_\ell \leq n_\ell$ (the other case can be proved similarly). When $t = 0$, we have

$$\mathbb{E}[\|\tilde{\boldsymbol{M}}_\ell^{(0)} - \nabla_\ell f(\boldsymbol{x}^{(0)})\|_F^2]$$

$$= \mathbb{E}[\|\beta_1 \boldsymbol{P}_\ell^{(0)}(\boldsymbol{P}_\ell^{(0)})^\top \boldsymbol{G}_\ell^{(0)} - \nabla_\ell f(\boldsymbol{x}^{(0)})\|_F^2]$$

$$= \mathbb{E}[\|\beta_1(\boldsymbol{P}_\ell^{(0)}(\boldsymbol{P}_\ell^{(0)})^\top - \boldsymbol{I})\boldsymbol{G}_\ell^{(0)} + \beta_1(\boldsymbol{G}_\ell^{(0)} - \nabla_\ell f(\boldsymbol{x}^{(0)})) - (1-\beta_1)\nabla_\ell f(\boldsymbol{x}^{(0)})\|_F^2]$$

$$\leq \beta_1 \mathbb{E}[\|(\boldsymbol{P}_\ell^{(0)}(\boldsymbol{P}_\ell^{(0)})^\top - \boldsymbol{I})\boldsymbol{G}_\ell^{(0)} + \boldsymbol{G}_\ell^{(0)} - \nabla_\ell f(\boldsymbol{x}^{(0)})\|_F^2] + (1-\beta_1)\|\nabla_\ell f(\boldsymbol{x}^{(0)})\|_F^2, \tag{36}$$

where the inequality uses Jensen's inequality. For the first term we have

$$\mathbb{E}[\|(\boldsymbol{P}_\ell^{(0)}(\boldsymbol{P}_\ell^{(0)})^\top - \boldsymbol{I})\boldsymbol{G}_\ell^{(0)} + \boldsymbol{G}_\ell^{(0)} - \nabla_\ell f(\boldsymbol{x}^{(0)})\|_F^2]$$

$$\leq 2\mathbb{E}[\|(\boldsymbol{I} - \boldsymbol{P}_\ell^{(0)}(\boldsymbol{P}_\ell^{(0)})^\top)\boldsymbol{G}_\ell^{(0)}\|_F^2] + 2\mathbb{E}[\|\boldsymbol{G}_\ell^{(0)} - \nabla_\ell f(\boldsymbol{x}^{(0)})\|_F^2]$$

$$\leq 2(1 - \delta_\ell)\mathbb{E}[\|\boldsymbol{G}_\ell\|_F^2] + 2\mathbb{E}[\|\boldsymbol{G}_\ell^{(0)} - \nabla_\ell f(\boldsymbol{x}^{(0)})\|_F^2]$$

$$\leq 2(1 - \delta_\ell)\|\nabla_\ell f(\boldsymbol{x}^{(0)})\|_F^2 + \frac{(4 - 2\delta_\ell)\sigma_\ell^2}{\mathcal{B}}, \tag{37}$$

where the first inequality uses Cauchy's inequality, the second inequality uses Lemma 1, the third inequality uses $\mathbb{E}[\|\boldsymbol{G}_\ell^{(0)} - \nabla_\ell f(\boldsymbol{x}^{(0)})\|_F^2] \leq \sigma_\ell^2/\mathcal{B}$ (Assumption 3). Applying (37) and Lemma 2 to (36) yields (33).

When $t = k\tau$, $k \in \mathbb{N}^*$, we have

$$\mathbb{E}[\|\tilde{\boldsymbol{M}}_\ell^{(t)} - \nabla_\ell f(\boldsymbol{x}^{(t)})\|_F^2]$$

$$= \mathbb{E}[\|\boldsymbol{P}_\ell^{(t)}(\boldsymbol{P}_\ell^{(t)})^\top[(1 - \beta_1)\tilde{\boldsymbol{M}}_\ell^{(t-1)} + \beta_1 \boldsymbol{G}_\ell^{(t)} - \nabla_\ell f(\boldsymbol{x}^{(t)})] - (\boldsymbol{I} - \boldsymbol{P}_\ell^{(t)}(\boldsymbol{P}_\ell^{(t)})^\top)\nabla_\ell f(\boldsymbol{x}^{(t)})\|_F^2]$$

$$= \mathbb{E}[\|\boldsymbol{P}_\ell^{(t)}(\boldsymbol{P}_\ell^{(t)})^\top[(1 - \beta_1)\tilde{\boldsymbol{M}}_\ell^{(t-1)} + \beta_1 \boldsymbol{G}_\ell^{(t)} - \nabla_\ell f(\boldsymbol{x}^{(t)})]\|_F^2]$$

$$+ \mathbb{E}[\|(\boldsymbol{I} - \boldsymbol{P}_\ell^{(t)}(\boldsymbol{P}_\ell^{(t)})^\top)\nabla_\ell f(\boldsymbol{x}^{(t)})\|_F^2], \tag{38}$$

where the second equality uses Lemma 3. By $\|\boldsymbol{P}_\ell^{(t)}(\boldsymbol{P}_\ell^{(t)})^\top\|_2 = 1$, we have

$$\mathbb{E}[\|\boldsymbol{P}_\ell^{(t)}(\boldsymbol{P}_\ell^{(t)})^\top[(1 - \beta_1)\tilde{\boldsymbol{M}}_\ell^{(t-1)} + \beta_1 \boldsymbol{G}_\ell^{(t)} - \nabla_\ell f(\boldsymbol{x}^{(t)})]\|_F^2]$$

$$\leq \mathbb{E}[\|(1 - \beta_1)\tilde{\boldsymbol{M}}_\ell^{(t-1)} + \beta_1 \boldsymbol{G}_\ell^{(t)} - \nabla_\ell f(\boldsymbol{x}^{(t)})\|_F^2]$$

$$= \mathbb{E}[\|(1 - \beta_1)(\tilde{\boldsymbol{M}}_\ell^{(t-1)} - \nabla_\ell f(\boldsymbol{x}^{(t)})) + \beta_1(\boldsymbol{G}_\ell^{(t)} - \nabla_\ell f(\boldsymbol{x}^{(t)}))\|_F^2]$$

$$\leq \mathbb{E}[\|(1 - \beta_1)(\tilde{\boldsymbol{M}}_\ell^{(t-1)} - \nabla_\ell f(\boldsymbol{x}^{(t)}))\|_F^2] + \beta_1^2 \mathbb{E}[\|\boldsymbol{G}_\ell^{(t)} - \nabla_\ell f(\boldsymbol{x}^{(t)})\|_F^2], \tag{39}$$

where the last inequality uses the unbiasedness of $\boldsymbol{G}_\ell^{(t)}$ (Assumption 3). By Young's inequality, we have

$$\mathbb{E}[\|\tilde{\boldsymbol{M}}_\ell^{(t-1)} - \nabla_\ell f(\boldsymbol{x}^{(t)})\|_F^2]$$

$$= \mathbb{E}[\|(\tilde{\boldsymbol{M}}_\ell^{(t-1)} - \nabla_\ell f(\boldsymbol{x}^{(t-1)})) - (\nabla_\ell f(\boldsymbol{x}^{(t)}) - \nabla_\ell f(\boldsymbol{x}^{(t-1)})\|_F^2]$$

$$\leq \left(1 + \frac{\delta_\ell \beta_1}{4}\right) \mathbb{E}[\|\tilde{\boldsymbol{M}}_\ell^{(t-1)} - \nabla_\ell f(\boldsymbol{x}^{(t-1)})\|_F^2] + \left(1 + \frac{4}{\delta_\ell \beta_1}\right) \mathbb{E}[\|\nabla_\ell f(\boldsymbol{x}^{(t)}) - \nabla_\ell f(\boldsymbol{x}^{(t-1)})\|_F^2]. \tag{40}$$

Applying (40) to (39) yields

$$\mathbb{E}[\|\boldsymbol{P}_\ell^{(t)}(\boldsymbol{P}_\ell^{(t)})^\top[(1 - \beta_1)\tilde{\boldsymbol{M}}_\ell^{(t-1)} + \beta_1 \boldsymbol{G}_\ell^{(t)} - \nabla_\ell f(\boldsymbol{x}^{(t)})]\|_F^2]$$

$$\leq \left(1 - \left(1 - \frac{\delta_\ell}{4}\right)\beta_1\right) \mathbb{E}[\|\tilde{\boldsymbol{M}}_\ell^{(t-1)} - \nabla_\ell f(\boldsymbol{x}^{(t-1)})\|_F^2] + \frac{\beta_1^2 \sigma^2}{\mathcal{B}}$$

$$+ \frac{5(1 - \beta_1)}{\delta_\ell \beta_1} \mathbb{E}[\|\nabla_\ell f(\boldsymbol{x}^{(t)}) - \nabla_\ell f(\boldsymbol{x}^{(t-1)})\|_F^2]. \tag{41}$$

For the second term in (38), we have

$$\mathbb{E}[\|(\boldsymbol{I} - \boldsymbol{P}_\ell^{(t)}(\boldsymbol{P}_\ell^{(t)})^\top)\nabla_\ell f(\boldsymbol{x}^{(t)})\|_F^2]$$

$$\leq 2\mathbb{E}[\|(\boldsymbol{I} - \boldsymbol{P}_\ell^{(t)}(\boldsymbol{P}_\ell^{(t)})^\top)\boldsymbol{G}_\ell^{(t)}\|_F^2] + 2\mathbb{E}[\|(\boldsymbol{I} - \boldsymbol{P}_\ell^{(t)}(\boldsymbol{P}_\ell^{(t)})^\top)(\boldsymbol{G}_\ell^{(t)} - \nabla_\ell f(\boldsymbol{x}^{(t)}))\|_F^2]$$

$$\leq 2(1 - \delta_\ell)\mathbb{E}[\|\boldsymbol{G}_\ell^{(t)}\|_F^2] + 2\mathbb{E}[\|\boldsymbol{G}_\ell^{(t)} - \nabla_\ell f(\boldsymbol{x}^{(t)})\|_F^2]$$

$$\leq 2(1 - \delta_\ell)\mathbb{E}[\|\nabla_\ell f(\boldsymbol{x}^{(t)})\|_F^2] + \frac{4\sigma_\ell^2}{\mathcal{B}}, \tag{42}$$

where the first inequality uses Cauchy's inequality, the second inequality uses Lemma 1 and $\|\boldsymbol{I} - \boldsymbol{P}_\ell^{(t)}(\boldsymbol{P}_\ell^{(t)})^\top\|_2 = 1$, the third inequality uses Assumption 3. Applying (41)(42) to (38) and using Lemma 2 yields (34).

When $t = k\tau + r$, $k \in \mathbb{N}$, $1 \leq r < \tau$, we have

$$\mathbb{E}[\|\tilde{\boldsymbol{M}}_\ell^{(t)} - \nabla_\ell f(\boldsymbol{x}^{(t)})\|_F^2]$$

$$=\mathbb{E}[\|(1-\beta_1)(\tilde{M}_\ell^{(t-1)}-\nabla_\ell f(\boldsymbol{x}^{(t)}))+\beta_1(\boldsymbol{P}_\ell^{(t)}(\boldsymbol{P}_\ell^{(t)})^\top\boldsymbol{G}_\ell^{(t)}-\nabla_\ell f(\boldsymbol{x}^{(t)}))\|_F^2]$$

$$=\mathbb{E}[\|(1-\beta_1)(\tilde{M}_\ell^{(t-1)}-\nabla_\ell f(\boldsymbol{x}^{(t)}))+\beta_1(\boldsymbol{P}_\ell^{(t)}(\boldsymbol{P}_\ell^{(t)})^\top-\boldsymbol{I})\nabla_\ell f(\boldsymbol{x}^{(t)})\|_F^2]$$

$$+\beta_1^2\mathbb{E}[\boldsymbol{P}_\ell^{(t)}(\boldsymbol{P}_\ell^{(t)})^\top(\boldsymbol{G}_\ell^{(t)}-\nabla_\ell f(\boldsymbol{x}^{(t)}))\|_F^2]$$

$$\le(1-\beta_1)\mathbb{E}[\|\tilde{M}_\ell^{(t-1)}-\nabla_\ell f(\boldsymbol{x}^{(t)})\|_F^2]+\beta_1\mathbb{E}[\|(\boldsymbol{I}-\boldsymbol{P}_\ell^{(t)}(\boldsymbol{P}_\ell^{(t)})^\top)\nabla_\ell f(\boldsymbol{x}^{(t)})\|_F^2$$

$$+\beta_1^2\mathbb{E}[\boldsymbol{P}_\ell^{(t)}(\boldsymbol{P}_\ell^{(t)})^\top(\boldsymbol{G}_\ell^{(t)}-\nabla_\ell f(\boldsymbol{x}^{(t)}))\|_F^2], \tag{43}$$

where the second equality uses the unbiasedness of $\boldsymbol{G}_\ell^{(t)}$ and the independence implied by $\boldsymbol{P}_\ell^{(t)}=\boldsymbol{P}_\ell^{(t-1)}$, the inequality uses Jensen's inequality. The first term is similarly bounded as (40). For the second term, we have

$$\mathbb{E}[\|(\boldsymbol{I}-\boldsymbol{P}_\ell^{(k\tau)}(\boldsymbol{P}_\ell^{(k\tau)})^\top)\nabla_\ell f(\boldsymbol{x}^{(t)})\|_F^2]$$

$$\le\left(1+\frac{\delta_\ell}{4}\right)\mathbb{E}[\|(\boldsymbol{I}-\boldsymbol{P}_\ell^{(k\tau)}(\boldsymbol{P}_\ell^{(k\tau)})^\top)\boldsymbol{G}_\ell^{(k\tau)}\|_F^2]$$

$$+\left(1+\frac{4}{\delta_\ell}\right)\mathbb{E}[\|(\boldsymbol{I}-\boldsymbol{P}_\ell^{(k\tau)}(\boldsymbol{P}_\ell^{(k\tau)})^\top)(\nabla_\ell f(\boldsymbol{x}^{(t)})-\boldsymbol{G}_\ell^{(k\tau)})\|_F^2]$$

$$\le\left(1-\frac{3\delta_\ell}{4}\right)\mathbb{E}[\|\boldsymbol{G}_\ell^{(k\tau)}\|_F^2]+2\left(1+\frac{4}{\delta_\ell}\right)\mathbb{E}[\|\boldsymbol{G}_\ell^{(k\tau)}-\nabla_\ell f(\boldsymbol{x}^{(k\tau)})\|_F^2]$$

$$+2\left(1+\frac{4}{\delta_\ell}\right)\mathbb{E}[\|\nabla_\ell f(\boldsymbol{x}^{(t)})-\nabla_\ell f(\boldsymbol{x}^{(k\tau)})\|_F^2], \tag{44}$$

where the first inequality uses Young's inequality, the second inequality uses Lemma 1 and Cauchy's inequality. We further have

$$\left(1-\frac{3\delta_\ell}{4}\right)\mathbb{E}[\|\boldsymbol{G}_\ell^{(k\tau)}\|_F^2]+2\left(1+\frac{4}{\delta_\ell}\right)\mathbb{E}[\|\boldsymbol{G}_\ell^{(k\tau)}-\nabla_\ell f(\boldsymbol{x}^{(k\tau)})\|_F^2]$$

$$\le\left(1-\frac{3\delta_\ell}{4}\right)\mathbb{E}[\|\nabla_\ell f(\boldsymbol{x}^{(k\tau)})\|_F^2]+\frac{11}{\delta_\ell}\mathbb{E}[\|\boldsymbol{G}_\ell^{(k\tau)}-\nabla_\ell f(\boldsymbol{x}^{(k\tau)})\|_F^2]$$

$$\le\left(1-\frac{3\delta_\ell}{4}\right)\mathbb{E}[\|\nabla_\ell f(\boldsymbol{x}^{(k\tau)})\|_F^2]+\frac{11\sigma_\ell^2}{\delta_\ell\mathcal{B}}$$

$$\le\left(1-\frac{\delta_\ell}{2}\right)\mathbb{E}[\|\nabla_\ell f(\boldsymbol{x}^{(t)})\|_F^2]+\left(1+\frac{4}{\delta_\ell}\right)\mathbb{E}[\|\nabla_\ell f(\boldsymbol{x}^{(t)})-\nabla_\ell f(\boldsymbol{x}^{(k\tau)})\|_F^2]+\frac{11\sigma_\ell^2}{\delta_\ell\mathcal{B}}, \tag{45}$$

where the first inequality uses unbiasedness of $\boldsymbol{G}_\ell^{(k\tau)}$, the second inequality uses Assumption 3, the third inequality uses Young's inequality.

Applying (45) to (44) and applying Cauchy's inequality yields

$$\mathbb{E}[\|(\boldsymbol{I}-\boldsymbol{P}_\ell^{(k\tau)}(\boldsymbol{P}_\ell^{(k\tau)})^\top)\nabla_\ell f(\boldsymbol{x}^{(t)})\|_F^2]$$

$$\le\left(1-\frac{\delta_\ell}{2}\right)\mathbb{E}[\|\nabla_\ell f(\boldsymbol{x}^{(t)})\|_F^2]+\frac{11\sigma_\ell^2}{\delta_\ell\mathcal{B}}+\frac{15r}{\delta_\ell}\sum_{i=1}^r\mathbb{E}[\|\nabla_\ell f(\boldsymbol{x}^{(k\tau+i)})-\nabla_\ell f(\boldsymbol{x}^{(k\tau+i-1)})\|_F^2]. \tag{46}$$

For the third term, we have

$$\mathbb{E}[\|\boldsymbol{P}_\ell^{(k\tau)}(\boldsymbol{P}_\ell^{(k\tau)})^\top(\boldsymbol{G}_\ell^{(t)}-\nabla_\ell f(\boldsymbol{x}^{(t)}))\|_F^2]\le\mathbb{E}[\|\boldsymbol{G}_\ell^{(t)}-\nabla_\ell f(\boldsymbol{x}^{(t)})\|_F^2]\le\sigma_\ell^2, \tag{47}$$

where the first inequality uses $\|\boldsymbol{P}_\ell^{(k\tau)}(\boldsymbol{P}_\ell^{(k\tau)})^\top\|_2=1$, the second inequality uses Assumption 3.

Applying (40)(46)(47) to (43) yields (35). □

**Lemma 9** (Momentum error)**:** Under Assumption 2-3, if $0<\beta_1\le1$ in large-batch GaLore using MSGD and MP (Alg. 4), it holds for any $K\ge1$ that

$$\sum_{t=0}^{K\tau-1}\mathbb{E}[\|\tilde{\boldsymbol{m}}^{(t)}-\nabla f(\boldsymbol{x}^{(t)})\|_2^2]$$

$$\leq \left( \frac{5(1-\beta_1)}{(1-\underline{\delta}/4)\underline{\delta}\beta_1^2} + \frac{15\tau(\tau-1)}{2(1-\underline{\delta}/4)\underline{\delta}} + \frac{2(\tau-1)}{(1-\overline{\delta}/4)\beta_1} \right) L^2 \sum_{t=0}^{K\tau-2} \mathbb{E}[\|\boldsymbol{x}^{(t+1)} - \boldsymbol{x}^{(t)}\|_2^2]$$

$$+ \left( \frac{1-\underline{\delta}/2}{1-\underline{\delta}/4} + \frac{4}{(1-\overline{\delta}/4)\tau\beta_1} \right) \sum_{t=0}^{K\tau-1} \mathbb{E}[\|\nabla f(\boldsymbol{x}^{(t)})\|_2^2]$$

$$+ \left( \frac{5K}{(1-\overline{\delta}/4)\beta_1\mathcal{B}} + \frac{11K\tau}{(1-\underline{\delta}/4)\underline{\delta}\mathcal{B}} + \frac{K\tau\beta_1}{1-\overline{\delta}/4} \right) \sigma^2. \tag{48}$$

*Proof.* By Lemma 8 we have

$$\sum_{t=0}^{K\tau-1} \mathbb{E}[\|\tilde{\boldsymbol{M}}_\ell^{(t)} - \nabla_\ell f(\boldsymbol{x}^{(t)})\|_F^2] - \left( 1 - \left( 1 - \frac{\delta_\ell}{4} \right) \beta_1 \right) \sum_{t=0}^{K\tau-2} \mathbb{E}[\|\tilde{\boldsymbol{M}}_\ell^{(t)} - \nabla_\ell f(\boldsymbol{x}^{(t)})\|_F^2]$$

$$\leq \left( \frac{5(1-\beta_1)}{\delta_\ell\beta_1} + \frac{15\tau(\tau-1)\beta_1}{2\delta_\ell} + 2(\tau-1) \right) \sum_{t=0}^{K\tau-2} \mathbb{E}[\|\nabla_\ell f(\boldsymbol{x}^{(t+1)}) - \nabla_\ell f(\boldsymbol{x}^{(t)})\|_F^2]$$

$$+ \left( \frac{4}{\tau} + \left( 1 - \frac{\delta_\ell}{2} \right) \beta_1 \right) \sum_{t=0}^{K\tau-1} \mathbb{E}[\|\nabla_\ell f(\boldsymbol{x}^{(t)})\|_F^2] + \left( \frac{5K}{\mathcal{B}} + \frac{11K\tau\beta_1}{\delta_\ell\mathcal{B}} + K\tau\beta_1^2 \right) \sigma_\ell^2,$$

which implies

$$\sum_{t=0}^{K\tau-1} \mathbb{E}[\|\tilde{\boldsymbol{M}}_\ell^{(t)} - \nabla_\ell f(\boldsymbol{x}^{(t)})\|_F^2]$$

$$\leq \left( \frac{5(1-\beta_1)}{(1-\delta_\ell/4)\delta_\ell\beta_1^2} + \frac{15\tau(\tau-1)}{2(1-\delta_\ell/4)\delta_\ell} + \frac{2(\tau-1)}{(1-\delta_\ell/4)\beta_1} \right) \sum_{t=0}^{K\tau-2} \mathbb{E}[\|\nabla_\ell f(\boldsymbol{x}^{(t+1)}) - \nabla_\ell f(\boldsymbol{x}^{(t)})\|_F^2]$$

$$+ \left( \frac{1-\delta_\ell/2}{1-\delta_\ell/4} + \frac{4}{(1-\delta_\ell/4)\tau\beta_1} \right) \sum_{t=0}^{K\tau-1} \mathbb{E}[\|\nabla_\ell f(\boldsymbol{x}^{(t)})\|_F^2]$$

$$+ \left( \frac{5K}{(1-\delta_\ell/4)\beta_1\mathcal{B}} + \frac{11K\tau}{(1-\delta_\ell/4)\delta_\ell\mathcal{B}} + \frac{K\tau\beta_1}{1-\delta_\ell/4} \right) \sigma_\ell^2. \tag{49}$$

Summing (49) for $\ell = 1, \cdots, N_L$ and applying Assumption 2-3 yields (48). $\qquad\square$

Now we are ready to prove the convergence of Alg. 4.

**Theorem 12** (Convergence of large-batch GaLore): Under Assumptions 1-3, if hyperparameters

$$0 < \beta_1 \leq 1, \quad \tau \geq \frac{128}{3\beta_1\underline{\delta}}, \quad 0 < \eta \leq \min\left\{ \frac{1}{4L}, \sqrt{\frac{3\underline{\delta}\beta_1^2}{80L^2}}, \sqrt{\frac{\underline{\delta}}{40\tau^2L^2}}, \sqrt{\frac{3\beta_1}{32\tau L^2}} \right\}, \tag{50}$$

GaLore using large-batch stochastic gradients and MSGD with MP (Alg. 4) converges as

$$\frac{1}{K\tau} \sum_{t=0}^{K\tau-1} \mathbb{E}\|\nabla f(\boldsymbol{x}^{(t)})\|_2^2 \leq \frac{16\Delta}{\underline{\delta}\eta K\tau} + \left( \frac{160}{3\beta_1\underline{\delta}\tau\mathcal{B}} + \frac{352}{3\underline{\delta}^2\mathcal{B}} + \frac{32\beta_1}{3\underline{\delta}} \right) \sigma^2 \tag{51}$$

for any $K \geq 1$, where $\Delta = f(\boldsymbol{x}^{(0)}) - \inf_{\boldsymbol{x}} f(\boldsymbol{x})$.

*Proof.* By Lemma 4 we have

$$\sum_{t=0}^{K\tau-1} \mathbb{E}[\|\nabla f(\boldsymbol{x}^{(t)})\|_2^2] \leq \frac{2[f(\boldsymbol{x}^{(0)}) - \mathbb{E}[f(\boldsymbol{x}^{(K\tau)})]}{\eta} + \sum_{t=0}^{K\tau-1} \mathbb{E}[\|\tilde{\boldsymbol{m}}^{(t)} - \nabla f(\boldsymbol{x}^{(t)})\|_2^2]$$

$$- \left( \frac{1}{\eta^2} - \frac{L}{\eta} \right) \sum_{t=0}^{K\tau-1} \mathbb{E}[\|\boldsymbol{x}^{(t+1)} - \boldsymbol{x}^{(t)}\|_2^2]. \tag{52}$$

Applying Lemma 9 to (52) and using $\underline{\delta} \leq \overline{\delta} < 1$ yields

$$\left( \frac{\delta}{4} - \frac{16}{3\tau\beta_1} \right) \sum_{t=0}^{K\tau-1} \mathbb{E}[\|\nabla f(\boldsymbol{x}^{(t)})\|_2^2]$$

$$\leq \frac{2}{\eta} \mathbb{E}[f(\boldsymbol{x}^{(0)}) - f(\boldsymbol{x}^{(K\tau)})] + \left( \frac{20K}{3\beta_1\mathcal{B}} + \frac{44K\tau}{3\underline{\delta}\mathcal{B}} + \frac{4K\tau\beta_1}{3} \right) \sigma^2$$

$$- \left( \frac{1}{\eta^2} - \frac{L}{\eta} - \frac{20(1-\beta_1)L^2}{3\underline{\delta}\beta_1^2} - \frac{10\tau(\tau-1)L^2}{\underline{\delta}} - \frac{8(\tau-1)L^2}{3\beta_1} \right) \sum_{t=0}^{K\tau-1} \mathbb{E}[\|\boldsymbol{x}^{(t+1)} - \boldsymbol{x}^{(t)}\|_2^2]. \tag{53}$$

By (50) we have

$$\frac{\delta}{4} - \frac{16}{3\tau\beta_1} \geq \frac{\delta}{8}, \quad \text{and} \quad \frac{1}{4\eta^2} \geq \max \left\{ \frac{L}{\eta}, \frac{20(1-\beta_1)L^2}{3\underline{\delta}\beta_1^2}, \frac{10\tau(\tau-1)L^2}{\underline{\delta}}, \frac{8(\tau-1)L^2}{3\beta_1} \right\}. \tag{54}$$

Applying (54) to (53) yields (51). $\qquad\square$

We now prove Theorem 6, which is restated as follows.

**Corollary 13** (Convergence complexity of large-batch GaLore)**.** *Under Assumptions 1-3, if* $T \geq 2 + 256/(3\underline{\delta}) + (256\sigma)^2/(9\sqrt{\underline{\delta}}L\Delta)$ *and we choose*

$$\beta_1 = \left( 1 + \sqrt{\frac{\underline{\delta}^{3/2}\sigma^2 T}{L\Delta}} \right)^{-1},$$

$$\tau = \left\lceil \frac{128}{3\underline{\delta}\beta_1} \right\rceil,$$

$$\eta = \left( 4L + \sqrt{\frac{80L^2}{3\underline{\delta}\beta_1^2}} + \sqrt{\frac{40\tau^2 L^2}{\underline{\delta}}} + \sqrt{\frac{32\tau L^2}{3\beta_1}} \right)^{-1},$$

$$\mathcal{B} = \left\lceil \frac{1}{\underline{\delta}\beta_1} \right\rceil,$$

*GaLore using large-batch stochastic gradients and MSGD with MP (Alg. 4) converges as*

$$\frac{1}{T} \sum_{t=0}^{T-1} \mathbb{E}[\|\nabla f(\boldsymbol{x}^{(t)})\|_2^2] = \mathcal{O}\left( \frac{L\Delta}{\underline{\delta}^{5/2}T} + \sqrt{\frac{L\Delta\sigma^2}{\underline{\delta}^{7/2}T}} \right), \tag{55}$$

*where* $\Delta = f(\boldsymbol{x}^{(0)}) - \inf_{\boldsymbol{x}} f(\boldsymbol{x})$. *Consequently, the computation complexity to reach an* $\varepsilon$-*accurate solution* $\boldsymbol{x}$ *such that* $\|\nabla f(\boldsymbol{x})\|_2^2 \leq \varepsilon$ *is* $\mathcal{O}\left( \frac{L\Delta\sigma^2}{\underline{\delta}^{7/2}\varepsilon^2} + \frac{L\Delta}{\underline{\delta}^{5/2}\varepsilon} + \frac{\sigma^2}{\underline{\delta}^{1/2}L\Delta} + \frac{1}{\underline{\delta}} \right)$.

*Proof.* $T \geq 2 + 128/(3\underline{\delta}) + (128\sigma)^2/(9\sqrt{\underline{\delta}}L\Delta)$ guarantees $T \geq \tau$. Let $T = K\tau + r$, where $K \in \mathbb{N}^*$ and $0 \leq r < \tau$. If $r = 0$, (55) is a direct result of Theorem 12. If $r > 0$, applying Theorem 12 to $\tilde{K} := K + 1$ yields

$$\frac{1}{T} \sum_{t=0}^{T-1} \mathbb{E}[\|\nabla f(\boldsymbol{x}^{(t)})\|_2^2] \leq \frac{\tilde{K}\tau}{T} \cdot \frac{1}{\tilde{K}\tau} \sum_{t=0}^{\tilde{K}\tau-1} \mathbb{E}[\|\nabla f(\boldsymbol{x}^{(t)})\|_2^2] = \mathcal{O}\left( \frac{L\Delta}{\underline{\delta}^{5/2}T} + \sqrt{\frac{L\Delta\sigma^2}{\underline{\delta}^{7/2}T}} \right).$$

$\qquad\square$

## B.5. Convergence of GoLore

In this subsection, we present the proof for Theorem 8. GoLore using small-batch stochastic gradients and MSGD with MP is specified as Alg. 5.

**Lemma 10** (Momentum contraction)**:** Under Assumption 3, in large-batch GoLore using MSGD with MP (Alg. 5), if $0 < \beta_1 \leq 1$, term $\tilde{\boldsymbol{M}}_\ell^{(t)}$ has the following contraction properties:

- When $t = 0$, it holds that

$$
\mathbb{E}[\|\tilde{\boldsymbol{M}}_\ell^{(0)} - \nabla_\ell f(\boldsymbol{X}^{(0)})\|_F^2] \leq (\tau - 1)(1 - \delta_\ell \beta_1) \sum_{r=0}^{\tau-2} \mathbb{E}[\|\nabla_\ell f(\boldsymbol{x}^{(r+1)}) - \nabla_\ell f(\boldsymbol{x}^{(r)})\|_F^2]
$$
$$
+ \frac{2(1 - \delta_\ell \beta_1)}{\tau} \sum_{r=0}^{\tau-1} \mathbb{E}[\|\nabla_\ell f(\boldsymbol{x}^{(r)})\|_F^2] + \delta_\ell \beta_1^2 \sigma_\ell^2; \tag{56}
$$

- When $t = k\tau$, $k \in \mathbb{N}^*$, it holds that

$$
\mathbb{E}[\|\tilde{\boldsymbol{M}}_\ell^{(t)} - \nabla_\ell f(\boldsymbol{x}^{(t)})\|_F^2] - \delta_\ell \left(1 - \left(1 - \frac{\delta_\ell}{4}\right)\beta_1\right) \mathbb{E}[\|\tilde{\boldsymbol{M}}_\ell^{(t-1)} - \nabla_\ell f(\boldsymbol{x}^{(t-1)})\|_F^2]
$$
$$
\leq \frac{2(1 - \delta_\ell)}{\tau} \sum_{r=0}^{\tau-1} \mathbb{E}[\|\nabla_l f(\boldsymbol{x}^{(k\tau+r)})\|_F^2] + \frac{5(1 - \beta_1)}{\beta_1} \mathbb{E}[\|\nabla_\ell f(\boldsymbol{x}^{(t)}) - \nabla_\ell f(\boldsymbol{x}^{(t-1)})\|_F^2]
$$
$$
+ (\tau - 1)(1 - \delta_\ell) \sum_{r=0}^{\tau-2} \mathbb{E}[\|\nabla_\ell f(\boldsymbol{x}^{(k\tau+r+1)}) - \nabla_\ell f(\boldsymbol{x}^{(k\tau+r)})\|_F^2] + \delta_\ell \beta_1^2 \sigma_\ell^2; \tag{57}
$$

- When $t = k\tau + r$, $k \in \mathbb{N}$, $1 \leq r < \tau$, it holds that

$$
\mathbb{E}[\|\tilde{\boldsymbol{M}}_\ell^{(t)} - \nabla_\ell f(\boldsymbol{x}^{(t)})\|_F^2] - \left(1 - \left(1 - \frac{\delta_\ell}{4}\right)\beta_1\right) \mathbb{E}[\|\tilde{\boldsymbol{M}}_\ell^{(t-1)} - \nabla_\ell f(\boldsymbol{x}^{(t-1)})\|_F^2]
$$
$$
\leq \left(1 - \frac{\delta_\ell}{2}\right)\beta_1 \mathbb{E}[\|\nabla_\ell f(\boldsymbol{x}^{(t)})\|_F^2] + \frac{5(1 - \beta_1)}{\delta_\ell \beta_1} \mathbb{E}[\|\nabla_\ell f(\boldsymbol{x}^{(t)}) - \nabla_\ell f(\boldsymbol{x}^{(t-1)})\|_F^2]
$$
$$
+ \frac{10 r \beta_1}{\delta_\ell} \sum_{i=1}^{r} \mathbb{E}[\|\nabla_\ell f(\boldsymbol{x}^{(k\tau+i)}) - \nabla_\ell f(\boldsymbol{x}^{(k\tau+i-1)})\|_F^2] + \beta_1^2 \sigma_\ell^2. \tag{58}
$$

*Proof.* Without loss of generality assume $m_\ell \leq n_\ell$ (the other case can be proved similarly). When $t = 0$, we have

$$
\mathbb{E}[\|\tilde{\boldsymbol{M}}_\ell^{(0)} - \nabla_\ell f(\boldsymbol{x}^{(0)})\|_F^2]
$$
$$
= \mathbb{E}[\|\beta_1 \boldsymbol{P}_\ell^{(0)}(\boldsymbol{P}_\ell^{(0)})^\top \boldsymbol{G}_\ell^{(0)} - \nabla_\ell f(\boldsymbol{x}^{(0)})\|_F^2]
$$
$$
= \mathbb{E}[\|(\beta_1 \boldsymbol{P}_\ell^{(0)}(\boldsymbol{P}_\ell^{(0)})^\top - \boldsymbol{I})\nabla_\ell f(\boldsymbol{x}^{(0)})\|_F^2] + \beta_1^2 \mathbb{E}[\|\boldsymbol{P}_\ell^{(0)}(\boldsymbol{P}_\ell^{(0)})^\top (\boldsymbol{G}_\ell^{(0)} - \nabla_\ell f(\boldsymbol{x}^{(0)}))\|_F^2]
$$
$$
= \text{tr}((\nabla_\ell f(\boldsymbol{x}^{(0)}))^\top \mathbb{E}[(\beta_1 \boldsymbol{P}_\ell^{(0)}(\boldsymbol{P}_\ell^{(0)})^\top - \boldsymbol{I})^2]\nabla_\ell f(\boldsymbol{x}^{(0)}))
$$
$$
+ \beta_1^2 \text{tr}(\mathbb{E}_{\xi^{(0)}\sim\mathcal{D}}[(\boldsymbol{G}_\ell^{(0)} - \nabla_\ell f(\boldsymbol{x}^{(0)}))^\top \mathbb{E}_{\boldsymbol{P}\sim\mathcal{U}(\text{St}_{m_\ell,r_\ell})}[(\boldsymbol{P}\boldsymbol{P}^\top)^2](\boldsymbol{G}_\ell^{(0)} - \nabla_\ell f(\boldsymbol{x}^{(0)}))]), \tag{59}
$$

where the second equality uses unbiasedness of $\boldsymbol{G}_\ell^{(0)}$. By Lemma 5 we have

$$
\mathbb{E}[(\beta \boldsymbol{P}_\ell^{(0)}(\boldsymbol{P}_\ell^{(0)})^\top - \boldsymbol{I})^2] = \boldsymbol{I} - (2\beta_1 - \beta_1^2)\mathbb{E}[\boldsymbol{P}_\ell^{(0)}(\boldsymbol{P}_\ell^{(0)})^\top]
$$
$$
= \boldsymbol{I} - (2\beta_1 - \beta_1^2)\delta_\ell \boldsymbol{I},
$$

thus

$$
\text{tr}((\nabla_\ell f(\boldsymbol{x}^{(0)}))^\top \mathbb{E}[(\beta_1 \boldsymbol{P}_\ell^{(0)}(\boldsymbol{P}_\ell^{(0)})^\top - \boldsymbol{I})^2]\nabla_\ell f(\boldsymbol{x}^{(0)})) = (1 - \delta_\ell(2\beta_1 - \beta_1^2))\|\nabla_\ell f(\boldsymbol{x}^{(0)})\|_F^2
$$

$$\leq (1 - \delta_\ell \beta_1) \|\nabla_\ell f(\boldsymbol{x}^{(0)})\|_F^2. \tag{60}$$

Similarly, by Lemma 5 we have

$$\begin{aligned}
&\mathrm{tr}(\mathbb{E}_{\xi^{(0)} \sim \mathcal{D}}[(\boldsymbol{G}_\ell^{(0)} - \nabla_\ell f(\boldsymbol{x}^{(0)}))^\top \mathbb{E}_{\boldsymbol{P} \sim \mathcal{U}(\mathrm{St}_{m_\ell, r_\ell})}[(\boldsymbol{P}\boldsymbol{P}^\top)^2](\boldsymbol{G}_\ell^{(0)} - \nabla_\ell f(\boldsymbol{x}^{(0)}))]) \\
=&\mathrm{tr}\left(\mathbb{E}\left[(\boldsymbol{G}_\ell^{(0)} - \nabla_\ell f(\boldsymbol{x}^{(0)}))^\top \left(\frac{r_\ell}{m_\ell} \cdot \boldsymbol{I}\right)(\boldsymbol{G}_\ell^{(0)} - \nabla_\ell f(\boldsymbol{x}^{(0)}))\right]\right) \\
=&\delta_\ell \mathbb{E}[\|\boldsymbol{G}_\ell^{(0)} - \nabla_\ell f(\boldsymbol{x}^{(0)})\|_F^2] \\
\leq&\delta_\ell \sigma_\ell^2,
\end{aligned} \tag{61}$$

where the inequality uses Assumption 3. Applying (60)(61) and Lemma 2 to (59) yields (56).

When $t = k\tau$, $k \in \mathbb{N}^*$, we have

$$\begin{aligned}
&\mathbb{E}[\|\tilde{\boldsymbol{M}}_\ell^{(t)} - \nabla_\ell f(\boldsymbol{x}^{(t)})\|_F^2] \\
=&\mathbb{E}[\|\boldsymbol{P}_\ell^{(t)}(\boldsymbol{P}_\ell^{(t)})^\top[(1-\beta_1)\tilde{\boldsymbol{M}}_\ell^{(t-1)} + \beta_1 \boldsymbol{G}_\ell^{(t)} - \nabla_\ell f(\boldsymbol{x}^{(t)})] - (\boldsymbol{I} - \boldsymbol{P}_\ell^{(t)}(\boldsymbol{P}_\ell^{(t)})^\top)\nabla_\ell f(\boldsymbol{x}^{(t)})\|_F^2] \\
=&\delta_\ell \mathbb{E}[\|(1-\beta_1)\tilde{\boldsymbol{M}}_\ell^{(t-1)} + \beta_1 \boldsymbol{G}_\ell^{(t)} - \nabla_\ell f(\boldsymbol{x}^{(t)})\|_F^2] + (1-\delta_\ell)\mathbb{E}[\|\nabla_\ell f(\boldsymbol{x}^{(t)})\|_F^2],
\end{aligned} \tag{62}$$

where the second equality uses Lemma 3 and Lemma 5. For the first term, we have

$$\begin{aligned}
&\mathbb{E}[\|(1-\beta_1)\tilde{\boldsymbol{M}}_\ell^{(t-1)} + \beta_1 \boldsymbol{G}_\ell^{(t)} - \nabla_\ell f(\boldsymbol{x}^{(t)})\|_F^2] \\
=&\mathbb{E}[\|(1-\beta_1)(\tilde{\boldsymbol{M}}_\ell^{(t-1)} - \nabla_\ell f(\boldsymbol{x}^{(t)})) + \beta_1(\boldsymbol{G}_\ell^{(t)} - \nabla_\ell f(\boldsymbol{x}^{(t)}))\|_F^2] \\
\leq&\mathbb{E}[\|(1-\beta_1)(\tilde{\boldsymbol{M}}_\ell^{(t-1)} - \nabla_\ell f(\boldsymbol{x}^{(t)}))\|_F^2] + \beta_1^2 \mathbb{E}[\|\boldsymbol{G}_\ell^{(t)} - \nabla_\ell f(\boldsymbol{x}^{(t)})\|_F^2] \\
\leq&(1-\beta_1)\mathbb{E}[\|\tilde{\boldsymbol{M}}_\ell^{(t-1)} - \nabla_\ell f(\boldsymbol{x}^{(t)})\|_F^2] + \beta_1^2 \sigma_\ell^2,
\end{aligned} \tag{63}$$

where both inequalities use Assumption 3. By Young's inequality, we have

$$\begin{aligned}
&\mathbb{E}[\|\tilde{\boldsymbol{M}}_\ell^{(t-1)} - \nabla_\ell f(\boldsymbol{x}^{(t)})\|_F^2] \\
=&\mathbb{E}[\|(\tilde{\boldsymbol{M}}_\ell^{(t-1)} - \nabla_\ell f(\boldsymbol{x}^{(t-1)})) - (\nabla_\ell f(\boldsymbol{x}^{(t)}) - \nabla_\ell f(\boldsymbol{x}^{(t-1)}))\|_F^2] \\
\leq&\left(1 + \frac{\delta_\ell \beta_1}{4}\right)\mathbb{E}[\|\tilde{\boldsymbol{M}}_\ell^{(t-1)} - \nabla_\ell f(\boldsymbol{x}^{(t-1)})\|_F^2] + \left(1 + \frac{4}{\delta_\ell \beta_1}\right)\mathbb{E}[\|\nabla_\ell f(\boldsymbol{x}^{(t)}) - \nabla_\ell f(\boldsymbol{x}^{(t-1)})\|_F^2].
\end{aligned} \tag{64}$$

Applying (63)(64) and Lemma 2 to (62) yields (57).

When $t = k\tau + r$, $k \in \mathbb{N}$, $1 \leq r < \tau$, we have

$$\begin{aligned}
&\mathbb{E}[\|\tilde{\boldsymbol{M}}_\ell^{(t)} - \nabla_\ell f(\boldsymbol{x}^{(t)})\|_F^2] \\
=&\mathbb{E}[\|(1-\beta_1)(\tilde{\boldsymbol{M}}_\ell^{(t-1)} - \nabla_\ell f(\boldsymbol{x}^{(t)})) + \beta_1(\boldsymbol{P}_\ell^{(t)}(\boldsymbol{P}_\ell^{(t)})^\top \boldsymbol{G}_\ell^{(t)} - \nabla_\ell f(\boldsymbol{x}^{(t)}))\|_F^2] \\
=&\mathbb{E}[\|(1-\beta_1)(\tilde{\boldsymbol{M}}_\ell^{(t-1)} - \nabla_\ell f(\boldsymbol{x}^{(t)})) + \beta_1(\boldsymbol{P}_\ell^{(t)}(\boldsymbol{P}_\ell^{(t)})^\top - \boldsymbol{I})\nabla_\ell f(\boldsymbol{x}^{(t)})\|_F^2] \\
&+ \beta_1^2 \mathbb{E}[\boldsymbol{P}_\ell^{(t)}(\boldsymbol{P}_\ell^{(t)})^\top(\boldsymbol{G}_\ell^{(t)} - \nabla_\ell f(\boldsymbol{x}^{(t)}))\|_F^2] \\
\leq&(1-\beta_1)\mathbb{E}[\|\tilde{\boldsymbol{M}}_\ell^{(t-1)} - \nabla_\ell f(\boldsymbol{x}^{(t)})\|_F^2] + \beta_1 \mathbb{E}[\|(\boldsymbol{I} - \boldsymbol{P}_\ell^{(t)}(\boldsymbol{P}_\ell^{(t)})^\top)\nabla_\ell f(\boldsymbol{x}^{(t)})\|_F^2 \\
&+ \beta_1^2 \mathbb{E}[\boldsymbol{P}_\ell^{(t)}(\boldsymbol{P}_\ell^{(t)})^\top(\boldsymbol{G}_\ell^{(t)} - \nabla_\ell f(\boldsymbol{x}^{(t)}))\|_F^2],
\end{aligned} \tag{65}$$

where the second equality uses the unbiasedness of $\boldsymbol{G}_\ell^{(t)}$ and the independence implied by $\boldsymbol{P}_\ell^{(t)} = \boldsymbol{P}_\ell^{(t-1)}$, the inequality uses Jensen's inequality. The first term is similarly bounded as (64). For the second term, we have

$$\begin{aligned}
&\mathbb{E}[\|(\boldsymbol{I} - \boldsymbol{P}_\ell^{(k\tau)}(\boldsymbol{P}_\ell^{(k\tau)})^\top)\nabla_\ell f(\boldsymbol{x}^{(t)})\|_F^2] \\
\leq&\left(1 + \frac{\delta_\ell}{4}\right)\mathbb{E}[\|(\boldsymbol{I} - \boldsymbol{P}_\ell^{(k\tau)}(\boldsymbol{P}_\ell^{(k\tau)})^\top)\nabla_\ell f(\boldsymbol{x}^{(k\tau)})\|_F^2]
\end{aligned}$$

$$+ \left(1 + \frac{4}{\delta_\ell}\right) \mathbb{E}[\|(\boldsymbol{I} - \boldsymbol{P}_\ell^{(k\tau)}(\boldsymbol{P}_\ell^{(k\tau)})^\top)(\nabla_\ell f(\boldsymbol{x}^{(t)}) - \nabla_\ell f(\boldsymbol{x}^{(k\tau)}))\|_F^2]$$

$$\leq \left(1 - \frac{3\delta_\ell}{4}\right) \mathbb{E}[\|\nabla_\ell f(\boldsymbol{x}^{(k\tau)})\|_F^2] + \left(1 + \frac{4}{\delta_\ell}\right) \mathbb{E}[\|\nabla_\ell f(\boldsymbol{x}^{(t)}) - \nabla_\ell f(\boldsymbol{x}^{(k\tau)})\|_F^2], \tag{66}$$

where the first inequality uses Young's inequality, the second inequality uses Lemma 5 and $\|\boldsymbol{I} - \boldsymbol{P}_\ell^{(k\tau)}(\boldsymbol{P}_\ell^{(k\tau)})^\top\|_2 = 1$. By Young's inequality, we have

$$\mathbb{E}[\|\nabla_\ell f(\boldsymbol{x}^{(k\tau)})\|_F^2] \leq \left(1 + \frac{\delta_\ell}{4}\right) \mathbb{E}[\|\nabla_\ell f(\boldsymbol{x}^{(t)})\|_F^2] + \left(1 + \frac{4}{\delta_\ell}\right) \mathbb{E}[\|\nabla_\ell f(\boldsymbol{x}^{(t)}) - \nabla_\ell f(\boldsymbol{x}^{(k\tau)})\|_F^2]. \tag{67}$$

Applying (67) to (66) and applying Cauchy's inequality yields

$$\mathbb{E}[\|(\boldsymbol{I} - \boldsymbol{P}_\ell^{(k\tau)}(\boldsymbol{P}_\ell^{(k\tau)})^\top)\nabla_\ell f(\boldsymbol{x}^{(t)})\|_F^2]$$

$$\leq \left(1 - \frac{\delta_\ell}{2}\right) \mathbb{E}[\|\nabla_\ell f(\boldsymbol{x}^{(t)})\|_F^2] + \frac{10r}{\delta_\ell} \sum_{i=1}^r \mathbb{E}[\|\nabla_\ell f(\boldsymbol{x}^{(k\tau+i)}) - \nabla_\ell f(\boldsymbol{x}^{(k\tau+i-1)})\|_F^2]. \tag{68}$$

For the third term, we have

$$\mathbb{E}[\|\boldsymbol{P}_\ell^{(k\tau)}(\boldsymbol{P}_\ell^{(k\tau)})^\top(\boldsymbol{G}_\ell^{(t)} - \nabla_\ell f(\boldsymbol{x}^{(t)}))\|_F^2] \leq \mathbb{E}[\|\boldsymbol{G}_\ell^{(t)} - \nabla_\ell f(\boldsymbol{x}^{(t)})\|_F^2] \leq \sigma_\ell^2, \tag{69}$$

where the first inequality uses $\|\boldsymbol{P}_\ell^{(k\tau)}(\boldsymbol{P}_\ell^{(k\tau)})^\top\|_2 = 1$, the second inequality uses Assumption 3.

Applying (64)(68)(69) to (65) yields (58). $\qquad\square$

**Lemma 11** (Momentum error): Under Assumption 2-3, if $0 < \beta_1 \leq 1$ in GoLore using MSGD and MP (Alg. 5), it holds for any $K \geq 1$ that

$$\sum_{t=0}^{K\tau-1} \mathbb{E}[\|\tilde{\boldsymbol{m}}^{(t)} - \nabla f(\boldsymbol{x}^{(t)})\|_2^2]$$

$$\leq \left(\frac{5(1-\beta_1)}{(1-\underline{\delta}/4)\underline{\delta}\beta_1^2} + \frac{5\tau(\tau-1)}{(1-\underline{\delta}/4)\underline{\delta}} + \frac{\tau-1}{(1-\overline{\delta}/4)\beta_1}\right) L^2 \sum_{t=0}^{K\tau-2} \mathbb{E}[\|\boldsymbol{x}^{(t+1)} - \boldsymbol{x}^{(t)}\|_2^2]$$

$$+ \left(\frac{1-\underline{\delta}/2}{1-\underline{\delta}/4} + \frac{2}{(1-\overline{\delta}/4)\tau\beta_1}\right) \sum_{t=0}^{K\tau-1} \mathbb{E}[\|\nabla f(\boldsymbol{x}^{(t)})\|_2^2] + \frac{K\tau\beta_1\sigma^2}{1-\overline{\delta}/4}. \tag{70}$$

*Proof.* By Lemma 10 we have

$$\sum_{t=0}^{K\tau-1} \mathbb{E}[\|\tilde{\boldsymbol{M}}_\ell^{(t)} - \nabla_\ell f(\boldsymbol{x}^{(t)})\|_F^2] - \left(1 - \left(1 - \frac{\delta_\ell}{4}\right)\beta_1\right) \sum_{t=0}^{K\tau-2} \mathbb{E}[\|\tilde{\boldsymbol{M}}_\ell^{(t)} - \nabla_\ell f(\boldsymbol{x}^{(t)})\|_F^2]$$

$$\leq \left(\frac{5(1-\beta_1)}{\delta_\ell\beta_1} + \frac{5\tau(\tau-1)\beta_1}{\delta_\ell} + (\tau-1)\right) \sum_{t=0}^{K\tau-2} \mathbb{E}[\|\nabla_\ell f(\boldsymbol{x}^{(t+1)}) - \nabla_\ell f(\boldsymbol{x}^{(t)})\|_F^2]$$

$$+ \left(\frac{2}{\tau} + \left(1 - \frac{\delta_\ell}{2}\right)\beta_1\right) \sum_{t=0}^{K\tau-1} \mathbb{E}[\|\nabla_\ell f(\boldsymbol{x}^{(t)})\|_F^2] + K\tau\beta_1^2\sigma_\ell^2,$$

which implies

$$\sum_{t=0}^{K\tau-1} \mathbb{E}[\|\tilde{\boldsymbol{M}}_\ell^{(t)} - \nabla_\ell f(\boldsymbol{x}^{(t)})\|_F^2]$$

$$\leq \left(\frac{5(1-\beta_1)}{(1-\delta_\ell/4)\delta_\ell\beta_1^2} + \frac{5\tau(\tau-1)}{(1-\delta_\ell/4)\delta_\ell} + \frac{\tau-1}{(1-\delta_\ell/4)\beta_1}\right) \sum_{t=0}^{K\tau-2} \mathbb{E}[\|\nabla_\ell f(\boldsymbol{x}^{(t+1)}) - \nabla_\ell f(\boldsymbol{x}^{(t)})\|_F^2]$$

$$+ \left( \frac{1 - \delta_\ell/2}{1 - \delta_\ell/4} + \frac{2}{(1 - \delta_\ell/4)\tau\beta_1} \right) \sum_{t=0}^{K\tau-1} \mathbb{E}[\|\nabla_\ell f(\boldsymbol{x}^{(t)})\|_F^2] + \frac{K\tau\beta_1\sigma_\ell^2}{1 - \delta_\ell/4}. \tag{71}$$

Summing (71) for $\ell = 1, \cdots, N_L$ and applying Assumption 2-3 yields (70). $\qquad\square$

Now we are ready to prove the convergence of Alg. 5.

**Theorem 14** (Convergence of Golore)**:** Under Assumptions 1-3, if hyperparameters

$$0 < \beta_1 \le 1, \quad \tau \ge \frac{64}{3\beta_1\underline{\delta}}, \quad 0 < \eta \le \min \left\{ \frac{1}{4L}, \sqrt{\frac{3\underline{\delta}\beta_1^2}{80L^2}}, \sqrt{\frac{3\underline{\delta}}{80\tau^2 L^2}}, \sqrt{\frac{3\beta_1}{16\tau L^2}} \right\}, \tag{72}$$

GoLore using small-batch stochastic gradients and MSGD with MP (Alg. 5) converges as

$$\frac{1}{K\tau} \sum_{t=0}^{K\tau-1} \mathbb{E}\|\nabla f(\boldsymbol{x}^{(t)})\|_2^2] \le \frac{16\Delta}{\underline{\delta}\eta K\tau} + \frac{32\beta_1\sigma^2}{3\underline{\delta}} \tag{73}$$

for any $K \ge 1$, where $\Delta = f(\boldsymbol{x}^{(0)}) - \inf_{\boldsymbol{x}} f(\boldsymbol{x})$.

*Proof.* By Lemma 4 we have

$$\sum_{t=0}^{K\tau-1} \mathbb{E}[\|\nabla f(\boldsymbol{x}^{(t)})\|_2^2] \le \frac{2[f(\boldsymbol{x}^{(0)}) - \mathbb{E}[f(\boldsymbol{x}^{(K\tau)})]]}{\eta} + \sum_{t=0}^{K\tau-1} \mathbb{E}[\|\tilde{\boldsymbol{m}}^{(t)} - \nabla f(\boldsymbol{x}^{(t)})\|_2^2]$$
$$- \left( \frac{1}{\eta^2} - \frac{L}{\eta} \right) \sum_{t=0}^{K\tau-1} \mathbb{E}[\|\boldsymbol{x}^{(t+1)} - \boldsymbol{x}^{(t)}\|_2^2]. \tag{74}$$

Applying Lemma 11 to (74) and using $\underline{\delta} \le \overline{\delta} < 1$ yields

$$\left( \frac{\delta}{4} - \frac{8}{3\tau\beta_1} \right) \sum_{t=0}^{K\tau-1} \mathbb{E}[\|\nabla f(\boldsymbol{x}^{(t)})\|_2^2]$$
$$\le \frac{2}{\eta}\mathbb{E}[f(\boldsymbol{x}^{(0)}) - f(\boldsymbol{x}^{(K\tau)})] + \frac{4K\tau\beta_1\sigma^2}{3}$$
$$- \left( \frac{1}{\eta^2} - \frac{L}{\eta} - \frac{20(1 - \beta_1)L^2}{3\underline{\delta}\beta_1^2} - \frac{20\tau(\tau - 1)L^2}{3\underline{\delta}} - \frac{4(\tau - 1)L^2}{3\beta_1} \right) \sum_{t=0}^{K\tau-1} \mathbb{E}[\|\boldsymbol{x}^{(t+1)} - \boldsymbol{x}^{(t)}\|_2^2]. \tag{75}$$

By (72) we have

$$\frac{\delta}{4} - \frac{8}{3\tau\beta_1} \ge \frac{\delta}{8}, \quad \text{and} \quad \frac{1}{4\eta^2} \ge \max \left\{ \frac{L}{\eta}, \frac{20(1 - \beta_1)L^2}{3\underline{\delta}\beta_1^2}, \frac{20\tau(\tau - 1)L^2}{3\underline{\delta}}, \frac{4(\tau - 1)L^2}{3\beta_1} \right\}. \tag{76}$$

Applying (76) to (75) yields (73). $\qquad\square$

We now prove Theorem 8, which is restated as follows.

**Corollary 15** (Convergence complexity of GoLore)**.** *Under Assumptions 1-3, if* $T \ge 2 + 128/(3\underline{\delta}) + (128\sigma)^2/(9\sqrt{\underline{\delta}}L\Delta)$ *and we choose*

$$\beta_1 = \left( 1 + \sqrt{\frac{\underline{\delta}^{3/2}\sigma^2 T}{L\Delta}} \right)^{-1},$$

$$\tau = \left\lceil \frac{64}{3\underline{\delta}\beta_1} \right\rceil,$$

$$\eta = \left( 4L + \sqrt{\frac{80L^2}{3\underline{\delta}\beta_1^2}} + \sqrt{\frac{80\tau^2 L^2}{3\underline{\delta}}} + \sqrt{\frac{16\tau L^2}{3\beta_1}} \right)^{-1},$$

*GoLore using small-batch stochastic gradients and MSGD with MP (Alg. 5) converges as*

$$\frac{1}{T} \sum_{t=0}^{T-1} \mathbb{E}[\|\nabla f(\boldsymbol{x}^{(t)})\|_2^2] = \mathcal{O}\left( \frac{L\Delta}{\underline{\delta}^{5/2}T} + \sqrt{\frac{L\Delta\sigma^2}{\underline{\delta}^{7/2}T}} \right), \tag{77}$$

*where $\Delta = f(\boldsymbol{x}^{(0)}) - \inf_{\boldsymbol{x}} f(\boldsymbol{x})$. Consequently, the computation complexity to reach an $\varepsilon$-accurate solution $\boldsymbol{x}$ such that $\|\nabla f(\boldsymbol{x})\|_2^2 \leq \varepsilon$ is $\mathcal{O}\left( \frac{L\Delta\sigma^2}{\underline{\delta}^{7/2}\varepsilon^2} + \frac{L\Delta}{\underline{\delta}^{5/2}\varepsilon} + \frac{\sigma^2}{\underline{\delta}^{1/2}L\Delta} + \frac{1}{\underline{\delta}} \right)$.*

*Proof.* $T \geq 2 + 128/(3\underline{\delta}) + (128\sigma)^2/(9\sqrt{\underline{\delta}}L\Delta)$ guarantees $T \geq \tau$. Let $T = K\tau + r$, where $K \in \mathbb{N}^*$ and $0 \leq r < \tau$. If $r = 0$, (77) is a direct result of Theorem 14. If $r > 0$, applying Theorem 14 to $\tilde{K} := K + 1$ yields

$$\frac{1}{T} \sum_{t=0}^{T-1} \mathbb{E}[\|\nabla f(\boldsymbol{x}^{(t)})\|_2^2] \leq \frac{\tilde{K}\tau}{T} \cdot \frac{1}{\tilde{K}\tau} \sum_{t=0}^{\tilde{K}\tau-1} \mathbb{E}[\|\nabla f(\boldsymbol{x}^{(t)})\|_2^2] = \mathcal{O}\left( \frac{L\Delta}{\underline{\delta}^{5/2}T} + \sqrt{\frac{L\Delta\sigma^2}{\underline{\delta}^{7/2}T}} \right).$$

$\square$

## C. Convergence of GaLore under isotropic noise assumptions

Based on the anisotropic gradient noise we use to construct the counter-example in the proof of GaLore's non-convergence under standard assumptions, an interesting open question is whether GaLore is guaranteed to converge if the noise are further assumed isotropic. In this section, we consider the following additional assumption:

**Assumption 16** (Isotropic noise). The distribution of stochastic noise for each gradient matrix is invariant under orthogonal transformations, *i.e.*, it holds for any layer $\ell = 1, \cdots, N_L$, parameter $\boldsymbol{x} \in \mathbb{R}^d$ and orthogonal matrix $\boldsymbol{O}_1 \in \mathbb{R}^{m_\ell \times m_\ell}$, $\boldsymbol{O}_2 \in \mathbb{R}^{n_\ell \times n_\ell}$ that

$$\nabla_\ell F(\boldsymbol{x};\xi) - \nabla_\ell f(\boldsymbol{x}) \overset{\text{dist}}{=} \boldsymbol{O}_1 [\nabla_\ell F(\boldsymbol{x};\xi) - \nabla_\ell f(\boldsymbol{x})] \boldsymbol{O}_2,$$

where $A \overset{\text{dist}}{=} B$ represents $A$ and $B$ shares the same distribution.

**Remark 7.** The property in Assumption 16 can be satisfied by multivariate Gaussian distribution, *e.g.*, $\text{vec}(\nabla_\ell F(\boldsymbol{x};\xi) - \nabla_\ell f(\boldsymbol{x})) \sim \mathcal{N}(0, \frac{\sigma_\ell^2}{m_\ell n_\ell} \cdot \boldsymbol{I}_{m_\ell \times n_\ell})$.

Besides Assumption 16, we consider an additional assumption, which is crucial in analyzing the projection error.

**Assumption 17** (Leading property). Let $\mathcal{D}_\ell(\boldsymbol{x})$ denotes the distribution of gradient noise $\nabla_\ell F(\boldsymbol{x};\xi) - \nabla_\ell f(\boldsymbol{x})$. We assume $\mathcal{D}_\ell(\boldsymbol{x})$ satisfies the following "leading property": if $\boldsymbol{A} \sim \mathcal{D}_\ell(\boldsymbol{x})$, $\boldsymbol{B} \in \mathbb{R}^{m_\ell \times n_\ell}$ satisfies $B_{11} \geq B_{22} \geq \cdots \geq B_{\min\{m_\ell,n_\ell\},\min\{m_\ell,n_\ell\}} \geq 0$ and $B_{ij} = 0$ for $i \neq j$, the SVD decomposition $\boldsymbol{U}\boldsymbol{\Sigma}\boldsymbol{V}^\top = \boldsymbol{A} + \boldsymbol{B}$ satisfies

$$\begin{cases} \frac{1}{r} \sum_{i=1}^{k} \sum_{j=1}^{r} \mathbb{E}[U_{ij}^2] \geq \frac{k}{m_\ell}, & \forall 1 \leq k, r \leq m_\ell, \quad \text{if } m_\ell \leq n_\ell; \\ \frac{1}{r} \sum_{i=1}^{k} \sum_{j=1}^{r} \mathbb{E}[V_{ij}^2] \geq \frac{k}{n_\ell}, & \forall 1 \leq k, r \leq n_\ell, \quad \text{if } m_\ell > n_\ell. \end{cases}$$

Though not fully established in theory, we can empirically validate that multivariate Gaussian distribution may satisfy Assumption 17.

Specifically, we consider the following experiment setup. Let $\text{vec}(\boldsymbol{A}) \sim \mathcal{N}(0, \sigma^2 \cdot \boldsymbol{I}_{32\times 32})$ for some noise scale $\sigma > 0$ and select a fixed matrix $\boldsymbol{B}$ with $B_{11} \geq B_{22} \geq \cdots \geq B_{32,32} \geq 0$. In order to validate the properties in expectation, we sample matrix $\boldsymbol{A}$ for 200,000 times and uses the empirical expectations $\hat{\mathbb{E}}[U_{ij}]$'s to estimate the true expectations $\mathbb{E}[U_{ij}]$'s. Figures 7, 8, 9 represent results under different noise scales $\sigma = 10, 1, 0.1$, respectively, where "$r = r_0$" in each figure plots the line connecting points $(k, \frac{1}{r_0} \sum_{i=1}^{k} \sum_{j=1}^{r_0} \hat{\mathbb{E}}[U_{ij}^2])$ for $k = 1, 2, \cdots, 32$. As presented, all lines "$r = r_0$" with $r_0 < 32$ are above the line "$r = 32$", which is guaranteed to pass through the points $(k, \frac{k}{32})$, $k = 1, 2, \cdots, 32$, in theory. Consequently, we have good reason to believe that multivariate Gaussian distribution can empirically satisfy Assumption 17.

With Assumptions 16 and 17, we can establish new error bounds for GaLore's SVD projection.

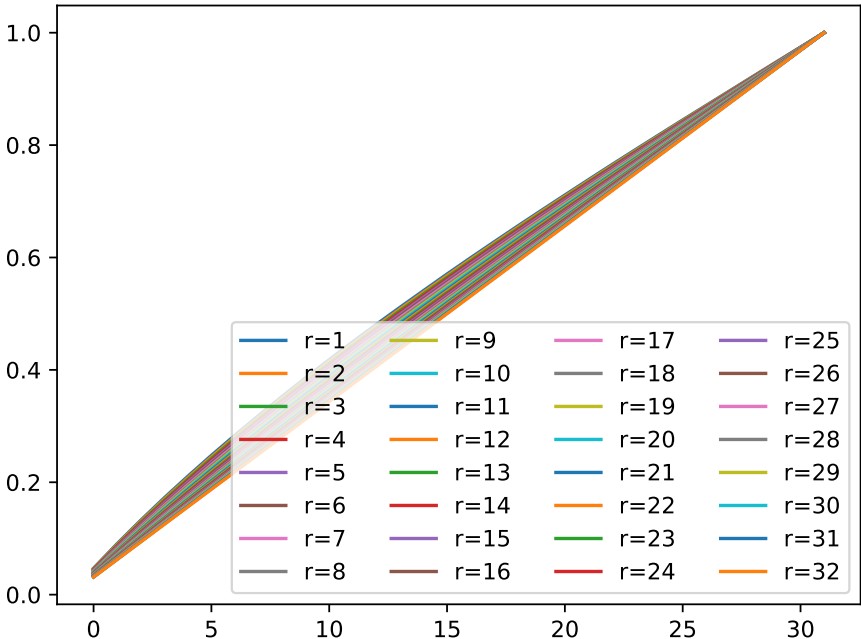

*Figure 7.* Observations with a small noise scale $\sigma = 0.1$.

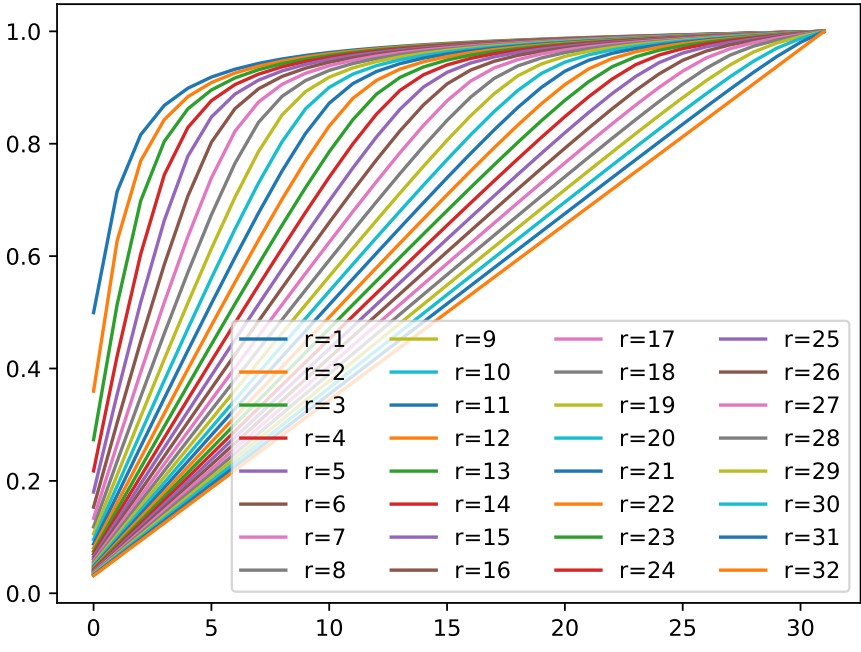

*Figure 8.* Observations with a medium noise scale $\sigma = 1$.

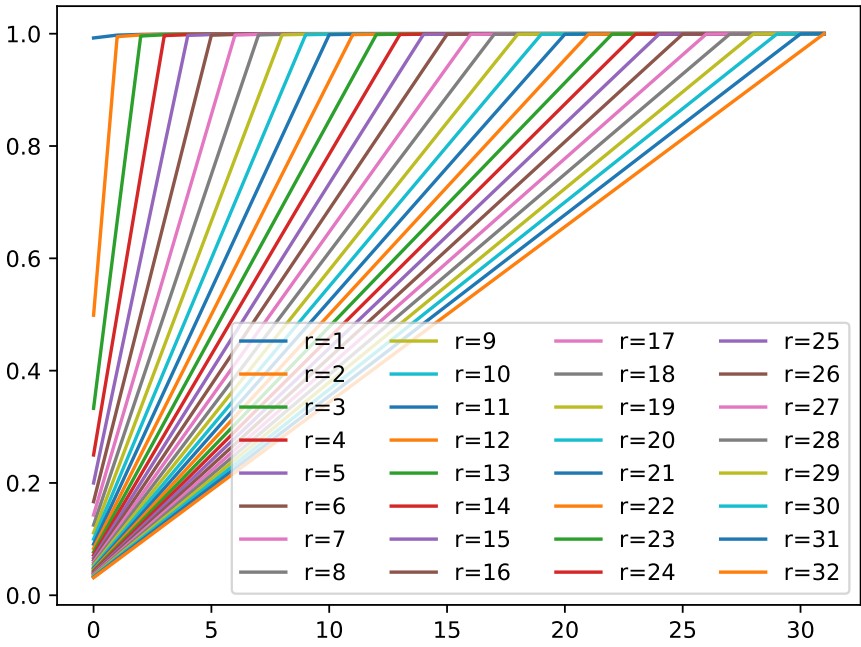

*Figure 9.* Observations with a large noise scale $\sigma = 10$.

**Lemma 12** (Error of GaLore's projection under isotropic noise)**:** Let $\boldsymbol{G} = \nabla_\ell f(\boldsymbol{x})$ and $\boldsymbol{E} = \nabla_\ell F(\boldsymbol{x}; \xi) - \nabla_\ell f(\boldsymbol{x})$, projection matrix $\boldsymbol{P} = \boldsymbol{U}[:,:r_\ell]$, $\boldsymbol{Q} = \boldsymbol{V}[:,:r_\ell]$ where $\boldsymbol{U\Sigma V}^\top = \boldsymbol{G} + \boldsymbol{E}$ is the SVD of stochastic gradient $\nabla_\ell F(\boldsymbol{x}; \xi)$, it holds under Assumptions 16 and 17 for $m_\ell \leq n_\ell$ that

$$\mathbb{E}[\|\boldsymbol{P}\boldsymbol{P}^\top \boldsymbol{G} - \boldsymbol{G}\|_F^2] \leq \left(1 - \frac{r_\ell}{m_\ell}\right) \|\boldsymbol{G}\|_F^2,$$

and for $m_\ell > n_\ell$ that

$$\mathbb{E}[\|\boldsymbol{G}\boldsymbol{Q}\boldsymbol{Q}^\top - \boldsymbol{G}\|_F^2] \leq \left(1 - \frac{r_\ell}{n_\ell}\right) \|\boldsymbol{G}\|_F^2.$$

*Proof.* We only consider the case where $m_\ell < n_\ell$, as the proof for the other case is similar. We first conduct SVD of $\boldsymbol{G}$ and get $\boldsymbol{G} = \boldsymbol{U}_0 \boldsymbol{\Sigma}_0 \boldsymbol{V}_0^\top$. It holds that

$$\begin{aligned}
\|\boldsymbol{P}\boldsymbol{P}^\top \boldsymbol{G}\|_F^2 &= \operatorname{tr}(\boldsymbol{G}^\top \boldsymbol{P}\boldsymbol{P}^\top \boldsymbol{G}) \\
&= \operatorname{tr}(\boldsymbol{V}_0 \boldsymbol{\Sigma}_0^\top \boldsymbol{U}_0^\top \boldsymbol{P}\boldsymbol{P}^\top \boldsymbol{U}_0 \boldsymbol{\Sigma}_0 \boldsymbol{V}_0^\top) \\
&= \operatorname{tr}(\boldsymbol{\Sigma}_0 \boldsymbol{\Sigma}_0^\top \boldsymbol{U}_0^\top \boldsymbol{P}\boldsymbol{P}^\top \boldsymbol{U}_0).
\end{aligned} \tag{78}$$

Denote $\tilde{\boldsymbol{U}} = \boldsymbol{U}_0^\top \boldsymbol{U}$ and $\tilde{\boldsymbol{V}} = \boldsymbol{V}_0^\top \boldsymbol{V}$, it holds that $\tilde{\boldsymbol{U}}\boldsymbol{\Sigma}_0\tilde{\boldsymbol{V}}^\top = (\boldsymbol{U}_0^\top \boldsymbol{U})\boldsymbol{\Sigma}_0(\boldsymbol{V}_0^\top \boldsymbol{V})^\top$ is SVD of $\boldsymbol{U}_0^\top(\boldsymbol{G} + \boldsymbol{E})\boldsymbol{V}_0 = \boldsymbol{U}_0^\top \boldsymbol{E}\boldsymbol{V}_0 + \boldsymbol{\Sigma}_0 \overset{\text{dist}}{=} \boldsymbol{E} + \boldsymbol{\Sigma}_0$. By Assumption 17 we have

$$\frac{1}{r_\ell} \sum_{i=1}^{k} \sum_{j=1}^{r_\ell} \mathbb{E}[\tilde{U}_{ij}^2] \geq \frac{k}{m_\ell}, \quad k = 1, 2, \cdots, m_\ell. \tag{79}$$

Let $\sigma_1 \geq \sigma_2 \geq \cdots \sigma_{m_\ell} \geq 0$ represent the singular values of $\boldsymbol{G}$, taking expectations of (78) yields

$$\mathbb{E}[\|\boldsymbol{P}\boldsymbol{P}^\top \boldsymbol{G}\|_F^2] = \operatorname{tr}(\boldsymbol{\Sigma}_0 \boldsymbol{\Sigma}_0^\top \mathbb{E}[\boldsymbol{U}_0^\top \boldsymbol{P}\boldsymbol{P}^\top \boldsymbol{U}_0])$$

$$= \sum_{i=1}^{m_\ell} \sigma_i^2 \sum_{j=1}^{r_\ell} \mathbb{E}[\tilde{U}_{ij}^2]$$

$$\geq \sum_{i=1}^{m_\ell} \sigma_i^2 \cdot \frac{r_\ell}{m_\ell} = \frac{r_\ell}{m_\ell} \cdot \|\boldsymbol{G}\|_F^2, \tag{80}$$

where the inequality applies $\sigma_1^2 \geq \sigma_2^2 \geq \cdots \sigma_{m_\ell}^2$ and (79). Based on (80), we have

$$\mathbb{E}[\|\boldsymbol{PP}^\top \boldsymbol{G} - \boldsymbol{G}\|_F^2] = \|\boldsymbol{G}\|_F^2 - \mathbb{E}[\|\boldsymbol{PP}^\top \boldsymbol{G}\|_F^2] \leq \left(1 - \frac{r_\ell}{m_\ell}\right) \|\boldsymbol{G}\|_F^2,$$

which completes the proof. $\qquad\square$

**Lemma 13** (Momentum contraction)**:** Under Assumptions 3,16-17, in GaLore using MSGD with MP, if $0 < \beta_1 \leq 1$, term $\tilde{\boldsymbol{M}}_\ell^{(t)}$ has the following contraction properties:

- When $t = 0$, it holds that

$$\mathbb{E}[\|\tilde{\boldsymbol{M}}_\ell^{(0)} - \nabla_\ell f(\boldsymbol{X}^{(0)})\|_F^2] \leq (\tau - 1)(2 - \delta_\ell) \sum_{r=0}^{\tau-2} \mathbb{E}[\|\nabla_\ell f(\boldsymbol{x}^{(r+1)}) - \nabla_\ell f(\boldsymbol{x}^{(r)})\|_F^2]$$

$$+ \frac{2(2 - \delta_\ell)}{\tau} \sum_{r=0}^{\tau-1} \mathbb{E}[\|\nabla_\ell f(\boldsymbol{x}^{(r)})\|_F^2] + \beta_1^2 \sigma_\ell^2; \tag{81}$$

- When $t = k\tau$, $k \in \mathbb{N}^*$, it holds that

$$\mathbb{E}[\|\tilde{\boldsymbol{M}}_\ell^{(t)} - \nabla_\ell f(\boldsymbol{x}^{(t)})\|_F^2] - \left(1 - \left(1 - \frac{\delta_\ell}{4}\right)\beta_1\right) \mathbb{E}[\|\tilde{\boldsymbol{M}}_\ell^{(t-1)} - \nabla_\ell f(\boldsymbol{x}^{(t-1)})\|_F^2]$$

$$\leq \frac{2(1 - \delta_\ell)}{\tau} \sum_{r=0}^{\tau-1} \mathbb{E}[\|\nabla_l f(\boldsymbol{x}^{(k\tau+r)})\|_F^2] + \frac{5(1 - \beta_1)}{\delta_\ell \beta_1} \mathbb{E}[\|\nabla_\ell f(\boldsymbol{x}^{(t)}) - \nabla_\ell f(\boldsymbol{x}^{(t-1)})\|_F^2]$$

$$+ (\tau - 1)(1 - \delta_\ell) \sum_{r=0}^{\tau-2} \mathbb{E}[\|\nabla_\ell f(\boldsymbol{x}^{(k\tau+r+1)}) - \nabla_\ell f(\boldsymbol{x}^{(k\tau+r)})\|_F^2] + \beta_1^2 \sigma_\ell^2; \tag{82}$$

- When $t = k\tau + r$, $k \in \mathbb{N}$, $1 \leq r < \tau$, it holds that

$$\mathbb{E}[\|\tilde{\boldsymbol{M}}_\ell^{(t)} - \nabla_\ell f(\boldsymbol{x}^{(t)})\|_F^2] - \left(1 - \left(1 - \frac{\delta_\ell}{4}\right)\beta_1\right) \mathbb{E}[\|\tilde{\boldsymbol{M}}_\ell^{(t-1)} - \nabla_\ell f(\boldsymbol{x}^{(t-1)})\|_F^2]$$

$$\leq \left(1 - \frac{\delta_\ell}{2}\right)\beta_1 \mathbb{E}[\|\nabla_\ell f(\boldsymbol{x}^{(t)})\|_F^2] + \frac{5(1 - \beta_1)}{\delta_\ell \beta_1} \mathbb{E}[\|\nabla_\ell f(\boldsymbol{x}^{(t)}) - \nabla_\ell f(\boldsymbol{x}^{(t-1)})\|_F^2]$$

$$+ \frac{10r\beta_1}{\delta_\ell} \sum_{i=1}^{r} \mathbb{E}[\|\nabla_\ell f(\boldsymbol{x}^{(k\tau+i)}) - \nabla_\ell f(\boldsymbol{x}^{(k\tau+i-1)})\|_F^2] + \beta_1^2 \sigma_\ell^2. \tag{83}$$

*Proof.* Without loss of generality assume $m_\ell \leq n_\ell$ (the other case can be proved similarly). When $t = 0$, (81) is direct result of Lemma 8 by letting $\mathcal{B} = 1$. When $t = 0$, we have

$$\mathbb{E}[\|\tilde{\boldsymbol{M}}_\ell^{(0)} - \nabla_\ell f(\boldsymbol{x}^{(0)})\|_F^2]$$

$$= \mathbb{E}[\|\beta_1 \boldsymbol{P}_\ell^{(0)} (\boldsymbol{P}_\ell^{(0)})^\top \boldsymbol{G}_\ell^{(0)} - \nabla_\ell f(\boldsymbol{x}^{(0)})\|_F^2]$$

$$= \mathbb{E}[\|(\beta_1 \boldsymbol{P}_\ell^{(0)} (\boldsymbol{P}_\ell^{(0)})^\top - \boldsymbol{I}) \nabla_\ell f(\boldsymbol{x}^{(0)})\|_F^2] + \beta_1^2 \mathbb{E}[\|\boldsymbol{P}_\ell^{(0)} (\boldsymbol{P}_\ell^{(0)})^\top (\boldsymbol{G}_\ell^{(0)} - \nabla_\ell f(\boldsymbol{x}^{(0)}))\|_F^2], \tag{84}$$

For the first term, we have

$$\mathbb{E}[\|(\beta_1 \boldsymbol{P}_\ell^{(0)} (\boldsymbol{P}_\ell^{(0)})^\top - \boldsymbol{I}) \nabla_\ell f(\boldsymbol{x}^{(0)})\|_F^2]$$

$$=(1-\beta_1)^2\mathbb{E}[\|\boldsymbol{P}_\ell^{(0)}(\boldsymbol{P}_\ell^{(0)})^\top\nabla_\ell f(\boldsymbol{x}^{(0)})\|_F^2] + \mathbb{E}[\|(\boldsymbol{I}-\boldsymbol{P}_\ell^{(0)}(\boldsymbol{P}_\ell^{(0)})^\top)\nabla_\ell f(\boldsymbol{x}^{(0)})\|_F^2]$$
$$\leq\left((1-\beta_1)^2 + (1-\delta_\ell)\right)\|\nabla_\ell f(\boldsymbol{x}^{(0)})\|_F^2 \leq (2-\delta_\ell)\|\nabla_\ell f(\boldsymbol{x}^{(0)})\|_F^2, \tag{85}$$

where the first inequality uses Lemma 12. For the second term, we have

$$\mathbb{E}[\|\boldsymbol{P}_\ell^{(0)}(\boldsymbol{P}_\ell^{(0)})^\top(\boldsymbol{G}_\ell^{(0)} - \nabla_\ell f(\boldsymbol{x}^{(0)}))\|_F^2] \leq \mathbb{E}[\|\boldsymbol{G}_\ell^{(0)} - \nabla_\ell f(\boldsymbol{x}^{(0)})\|_F^2] \leq \sigma_\ell^2. \tag{86}$$

Applying (85)(86) to (84) and using Lemma 2 yields (81).

When $t = k\tau$, $k \in \mathbb{N}^*$, according to the proof of Lemma 8, we have

$$\mathbb{E}[\|\tilde{\boldsymbol{M}}_\ell^{(t-1)} - \nabla_\ell f(\boldsymbol{x}^{(t)})\|_F^2]$$
$$\leq\left(1 + \frac{\delta_\ell\beta_1}{4}\right)\mathbb{E}[\|\tilde{\boldsymbol{M}}_\ell^{(t-1)} - \nabla_\ell f(\boldsymbol{x}^{(t-1)})\|_F^2] + \left(1 + \frac{4}{\delta_\ell\beta_1}\right)\mathbb{E}[\|\nabla_\ell f(\boldsymbol{x}^{(t)}) - \nabla_\ell f(\boldsymbol{x}^{(t-1)})\|_F^2], \tag{87}$$

and

$$\mathbb{E}[\|\tilde{\boldsymbol{M}}_\ell^{(t)} - \nabla_\ell f(\boldsymbol{x}^{(t)})\|_F^2]$$
$$=\mathbb{E}[\|\boldsymbol{P}_\ell^{(t)}(\boldsymbol{P}_\ell^{(t)})^\top[(1-\beta_1)\tilde{\boldsymbol{M}}_\ell^{(t-1)} + \beta_1\boldsymbol{G}_\ell^{(t)} - \nabla_\ell f(\boldsymbol{x}^{(t)})]\|_F^2]$$
$$\quad + \mathbb{E}[\|(\boldsymbol{I}-\boldsymbol{P}_\ell^{(t)}(\boldsymbol{P}_\ell^{(t)})^\top)\nabla_\ell f(\boldsymbol{x}^{(t)})\|_F^2]$$
$$\leq\mathbb{E}[\|(1-\beta_1)(\tilde{\boldsymbol{M}}_\ell^{(t-1)} - \nabla_\ell f(\boldsymbol{x}^{(t)}))\|_F^2] + \beta_1^2\sigma_\ell^2 + (1-\delta_\ell)\mathbb{E}[\|\nabla_\ell f(\boldsymbol{x}^{(t)})\|_F^2], \tag{88}$$

where the last inequality applies Lemma 12. Applying (87) to (88) and using Lemma 2 yields (82).

When $t = k\tau + r$, $k \in \mathbb{N}$, $1 \leq r < \tau$, we have the following results according to the proof of Lemma 8:

$$\mathbb{E}[\|\tilde{\boldsymbol{M}}_\ell^{(t)} - \nabla_\ell f(\boldsymbol{x}^{(t)})\|_F^2]$$
$$\leq(1-\beta_1)\mathbb{E}[\|\tilde{\boldsymbol{M}}_\ell^{(t-1)} - \nabla_\ell f(\boldsymbol{x}^{(t)})\|_F^2] + \beta_1\mathbb{E}[\|(\boldsymbol{I}-\boldsymbol{P}_\ell^{(t)}(\boldsymbol{P}_\ell^{(t)})^\top)\nabla_\ell f(\boldsymbol{x}^{(t)})\|_F^2$$
$$\quad + \beta_1^2\mathbb{E}[\boldsymbol{P}_\ell^{(t)}(\boldsymbol{P}_\ell^{(t)})^\top(\boldsymbol{G}_\ell^{(t)} - \nabla_\ell f(\boldsymbol{x}^{(t)}))\|_F^2]$$
$$\leq(1-\beta_1)\mathbb{E}[\|\tilde{\boldsymbol{M}}_\ell^{(t-1)} - \nabla_\ell f(\boldsymbol{x}^{(t)})\|_F^2] + \beta_1\mathbb{E}[\|(\boldsymbol{I}-\boldsymbol{P}_\ell^{(t)}(\boldsymbol{P}_\ell^{(t)})^\top)\nabla_\ell f(\boldsymbol{x}^{(t)})\|_F^2 + \beta_1^2\sigma_\ell^2, \tag{89}$$

For the second term, we have

$$\mathbb{E}[\|(\boldsymbol{I}-\boldsymbol{P}_\ell^{(k\tau)}(\boldsymbol{P}_\ell^{(k\tau)})^\top)\nabla_\ell f(\boldsymbol{x}^{(t)})\|_F^2]$$
$$\leq\left(1+\frac{\delta_\ell}{4}\right)\mathbb{E}[\|(\boldsymbol{I}-\boldsymbol{P}_\ell^{(k\tau)}(\boldsymbol{P}_\ell^{(k\tau)})^\top)\nabla_\ell f(\boldsymbol{x}^{(k\tau)})\|_F^2]$$
$$\quad + \left(1+\frac{4}{\delta_\ell}\right)\mathbb{E}[\|(\boldsymbol{I}-\boldsymbol{P}_\ell^{(k\tau)}(\boldsymbol{P}_\ell^{(k\tau)})^\top)(\nabla_\ell f(\boldsymbol{x}^{(t)}) - \nabla_\ell f(\boldsymbol{x}^{(k\tau)})\|_F^2]$$
$$\leq\left(1-\frac{3\delta_\ell}{4}\right)\mathbb{E}[\|\nabla_\ell f(\boldsymbol{x}^{(k\tau)})\|_F^2] + \left(1+\frac{4}{\delta_\ell}\right)\mathbb{E}[\|\nabla_\ell f(\boldsymbol{x}^{(t)}) - \nabla_\ell f(\boldsymbol{x}^{(k\tau)})\|_F^2]$$
$$\leq\left(1-\frac{\delta_\ell}{2}\right)\mathbb{E}[\|\nabla_\ell f(\boldsymbol{x}^{(t)})\|_F^2] + 2\left(1+\frac{4}{\delta_\ell}\right)\mathbb{E}[\|\nabla_\ell f(\boldsymbol{x}^{(t)}) - \nabla_\ell f(\boldsymbol{x}^{(k\tau)})\|_F^2]$$
$$\leq\left(1-\frac{\delta_\ell}{2}\right)\mathbb{E}[\|\nabla_\ell f(\boldsymbol{x}^{(t)})\|_F^2] + \frac{10r}{\delta_\ell}\sum_{i=1}^{r}\mathbb{E}[\|\nabla_\ell f(\boldsymbol{x}^{(k\tau+i)}) - \nabla_\ell f(\boldsymbol{x}^{(k\tau+i-1)})\|_F^2], \tag{90}$$

where the first inequality applies Young's inequality, the second inequality applies Lemma 12, the third inequality applies Young's inequality, the last inequality applies Cauchy's inequality. Applying (87)(90) to (89) yields (83). □

**Lemma 14** (Momentum error)**:** Under Assumptions 2-3,16-17, if $0 < \beta_1 \leq 1$ in GaLore using MSGD and MP, it holds for any $K \geq 1$ that

$$\sum_{t=0}^{K\tau-1}\mathbb{E}[\|\tilde{\boldsymbol{m}}^{(t)} - \nabla f(\boldsymbol{x}^{(t)})\|_2^2]$$

$$\leq \left( \frac{5(1-\beta_1)}{(1-\underline{\delta}/4)\underline{\delta}\beta_1^2} + \frac{5\tau(\tau-1)}{(1-\overline{\delta}/4)\underline{\delta}} + \frac{2(\tau-1)}{(1-\overline{\delta}/4)\beta_1} \right) L^2 \sum_{t=0}^{K\tau-2} \mathbb{E}[\|\boldsymbol{x}^{(t+1)} - \boldsymbol{x}^{(t)}\|_2^2]$$

$$+ \left( \frac{1-\underline{\delta}/2}{1-\underline{\delta}/4} + \frac{4}{(1-\overline{\delta}/4)\tau\beta_1} \right) \sum_{t=0}^{K\tau-1} \mathbb{E}[\|\nabla f(\boldsymbol{x}^{(t)})\|_2^2] + \frac{K\tau\beta_1\sigma^2}{1-\overline{\delta}/4}. \tag{91}$$

*Proof.* By Lemma 13 we have

$$\sum_{t=0}^{K\tau-1} \mathbb{E}[\|\tilde{\boldsymbol{M}}_\ell^{(t)} - \nabla_\ell f(\boldsymbol{x}^{(t)})\|_F^2] - \left( 1 - \left( 1 - \frac{\delta_\ell}{4} \right)\beta_1 \right) \sum_{t=0}^{K\tau-2} \mathbb{E}[\|\tilde{\boldsymbol{M}}_\ell^{(t)} - \nabla_\ell f(\boldsymbol{x}^{(t)})\|_F^2]$$

$$\leq \left( \frac{5(1-\beta_1)}{\delta_\ell\beta_1} + \frac{5\tau(\tau-1)\beta_1}{\delta_\ell} + 2(\tau-1) \right) \sum_{t=0}^{K\tau-2} \mathbb{E}[\|\nabla_\ell f(\boldsymbol{x}^{(t+1)}) - \nabla_\ell f(\boldsymbol{x}^{(t)})\|_F^2]$$

$$+ \left( \frac{4}{\tau} + \left( 1 - \frac{\delta_\ell}{2} \right)\beta_1 \right) \sum_{t=0}^{K\tau-1} \mathbb{E}[\|\nabla_\ell f(\boldsymbol{x}^{(t)})\|_F^2] + K\tau\beta_1^2\sigma_\ell^2,$$

which implies

$$\sum_{t=0}^{K\tau-1} \mathbb{E}[\|\tilde{\boldsymbol{M}}_\ell^{(t)} - \nabla_\ell f(\boldsymbol{x}^{(t)})\|_F^2]$$

$$\leq \left( \frac{5(1-\beta_1)}{(1-\delta_\ell/4)\delta_\ell\beta_1^2} + \frac{5\tau(\tau-1)}{(1-\delta_\ell/4)\delta_\ell} + \frac{2(\tau-1)}{(1-\delta_\ell/4)\beta_1} \right) \sum_{t=0}^{K\tau-2} \mathbb{E}[\|\nabla_\ell f(\boldsymbol{x}^{(t+1)}) - \nabla_\ell f(\boldsymbol{x}^{(t)})\|_F^2]$$

$$+ \left( \frac{1-\delta_\ell/2}{1-\delta_\ell/4} + \frac{4}{(1-\delta_\ell/4)\tau\beta_1} \right) \sum_{t=0}^{K\tau-1} \mathbb{E}[\|\nabla_\ell f(\boldsymbol{x}^{(t)})\|_F^2] + \frac{K\tau\beta_1\sigma_\ell^2}{1-\delta_\ell/4}. \tag{92}$$

Summing (92) for $\ell = 1, \cdots, N_L$ and applying Assumptions 2-3 yields (91). $\qquad\square$

Now we are ready to prove the convergence of GaLore with small-batch stochastic gradients under isotropic noise assumptions.

**Theorem 18** (Convergence of Galore under isotropic noise assumptions): Under Assumptions 1-3,16-17, if hyperparameters

$$0 < \beta_1 \leq 1, \quad \tau \geq \frac{128}{3\beta_1\underline{\delta}}, \quad 0 < \eta \leq \min\left\{ \frac{1}{4L}, \sqrt{\frac{3\underline{\delta}\beta_1^2}{80L^2}}, \sqrt{\frac{3\underline{\delta}}{80\tau^2L^2}}, \sqrt{\frac{3\beta_1}{32\tau L^2}} \right\}, \tag{93}$$

GaLore using small-batch stochastic gradients and MSGD with MP converges as

$$\frac{1}{K\tau} \sum_{t=0}^{K\tau-1} \mathbb{E}\|\nabla f(\boldsymbol{x}^{(t)})\|_2^2 \leq \frac{16\Delta}{\underline{\delta}\eta K\tau} + \frac{32\beta_1\sigma^2}{3\underline{\delta}} \tag{94}$$

for any $K \geq 1$, where $\Delta = f(\boldsymbol{x}^{(0)}) - \inf_{\boldsymbol{x}} f(\boldsymbol{x})$.

*Proof.* By Lemma 4 we have

$$\sum_{t=0}^{K\tau-1} \mathbb{E}[\|\nabla f(\boldsymbol{x}^{(t)})\|_2^2] \leq \frac{2[f(\boldsymbol{x}^{(0)}) - \mathbb{E}[f(\boldsymbol{x}^{(K\tau)})]]}{\eta} + \sum_{t=0}^{K\tau-1} \mathbb{E}[\|\tilde{\boldsymbol{m}}^{(t)} - \nabla f(\boldsymbol{x}^{(t)})\|_2^2]$$

$$- \left( \frac{1}{\eta^2} - \frac{L}{\eta} \right) \sum_{t=0}^{K\tau-1} \mathbb{E}[\|\boldsymbol{x}^{(t+1)} - \boldsymbol{x}^{(t)}\|_2^2]. \tag{95}$$

Applying Lemma 14 to (95) and using $\underline{\delta} \leq \overline{\delta} < 1$ yields

$$
\left(\frac{\delta}{4} - \frac{16}{3\tau\beta_1}\right) \sum_{t=0}^{K\tau-1} \mathbb{E}[\|\nabla f(\boldsymbol{x}^{(t)})\|_2^2]
$$

$$
\leq \frac{2}{\eta} \mathbb{E}[f(\boldsymbol{x}^{(0)}) - f(\boldsymbol{x}^{(K\tau)})] + \frac{4K\tau\beta_1\sigma^2}{3}
$$

$$
- \left(\frac{1}{\eta^2} - \frac{L}{\eta} - \frac{20(1-\beta_1)L^2}{3\underline{\delta}\beta_1^2} - \frac{20\tau(\tau-1)L^2}{3\underline{\delta}} - \frac{8(\tau-1)L^2}{3\beta_1}\right) \sum_{t=0}^{K\tau-1} \mathbb{E}[\|\boldsymbol{x}^{(t+1)} - \boldsymbol{x}^{(t)}\|_2^2]. \tag{96}
$$

By (93) we have

$$
\frac{\delta}{4} - \frac{16}{3\tau\beta_1} \geq \frac{\delta}{8}, \quad \text{and} \quad \frac{1}{4\eta^2} \geq \max\left\{\frac{L}{\eta}, \frac{20(1-\beta_1)L^2}{3\underline{\delta}\beta_1^2}, \frac{20\tau(\tau-1)L^2}{3\underline{\delta}}, \frac{8(\tau-1)L^2}{3\beta_1}\right\}. \tag{97}
$$

Applying (97) to (96) yields (94). $\qquad\square$

**Corollary 19** (Convergence complexity of GaLore under isotropic noise assumptions). *Under Assumptions 1-17, if $T \geq 2 + 256/(3\underline{\delta}) + (256\sigma)^2/(9\sqrt{\underline{\delta}}L\Delta)$ and we choose*

$$
\beta_1 = \left(1 + \sqrt{\frac{\delta^{3/2}\sigma^2 T}{L\Delta}}\right)^{-1},
$$

$$
\tau = \left\lceil \frac{128}{3\underline{\delta}\beta_1} \right\rceil,
$$

$$
\eta = \left(4L + \sqrt{\frac{80L^2}{3\underline{\delta}\beta_1^2}} + \sqrt{\frac{80\tau^2 L^2}{3\underline{\delta}}} + \sqrt{\frac{32\tau L^2}{3\beta_1}}\right)^{-1},
$$

*GaLore using small-batch stochastic gradients and MSGD with MP converges as*

$$
\frac{1}{T}\sum_{t=0}^{T-1} \mathbb{E}[\|\nabla f(\boldsymbol{x}^{(t)})\|_2^2] = \mathcal{O}\left(\frac{L\Delta}{\underline{\delta}^{5/2}T} + \sqrt{\frac{L\Delta\sigma^2}{\underline{\delta}^{7/2}T}}\right), \tag{98}
$$

*where $\Delta = f(\boldsymbol{x}^{(0)}) - \inf_{\boldsymbol{x}} f(\boldsymbol{x})$. Consequently, the computation complexity to reach an $\varepsilon$-accurate solution $\boldsymbol{x}$ such that $\|\nabla f(\boldsymbol{x})\|_2^2 \leq \varepsilon$ is $\mathcal{O}\left(\frac{L\Delta\sigma^2}{\underline{\delta}^{7/2}\varepsilon^2} + \frac{L\Delta}{\underline{\delta}^{5/2}\varepsilon} + \frac{\sigma^2}{\underline{\delta}^{1/2}L\Delta} + \frac{1}{\underline{\delta}}\right)$.*

*Proof.* $T \geq 2 + 256/(3\underline{\delta}) + (256\sigma)^2/(9\sqrt{\underline{\delta}}L\Delta)$ guarantees $T \geq \tau$. Let $T = K\tau + r$, where $K \in \mathbb{N}^*$ and $0 \leq r < \tau$. If $r = 0$, (98) is a direct result of Theorem 18. If $r > 0$, applying Theorem 18 to $\tilde{K} := K + 1$ yields

$$
\frac{1}{T}\sum_{t=0}^{T-1} \mathbb{E}[\|\nabla f(\boldsymbol{x}^{(t)})\|_2^2] \leq \frac{\tilde{K}\tau}{T} \cdot \frac{1}{\tilde{K}\tau}\sum_{t=0}^{\tilde{K}\tau-1} \mathbb{E}[\|\nabla f(\boldsymbol{x}^{(t)})\|_2^2] = \mathcal{O}\left(\frac{L\Delta}{\underline{\delta}^{5/2}T} + \sqrt{\frac{L\Delta\sigma^2}{\underline{\delta}^{7/2}T}}\right).
$$

$\qquad\square$

# D. Results for sparse subspace optimization

In this section, we illustrate how to transfer the main results of this paper to sparse subspace optimization algorithms. We first present the detailed algorithm formulation, then present the theoretical results corresponding to GaLore/GoLore. Although it only requires little effort to transfer results in GaLore/GoLore to sparse subspace optimization, we still include proofs for completeness.

### D.1. Algorithm design

While low-rank subspace optimization algorithms like GaLore/GoLore project full-parameter gradient $G \in \mathbb{R}^{(m \times n)}$ into low-rank subspaces via projection like $P^\top G$, sparse subspace optimization algorithms use a sparse mask $S$ to get $S \odot G$. Specifically, consider the following set

$$\mathrm{Sp}_{m,n}^k = \{S \in \{0,1\}^{m \times n} \mid \|S\|_F^2 = k\},$$

*i.e.*, a set of $m \times n$ matrices contains $k$ ones and $(mn - k)$ zeros. Corresponding to the subspace selecting strategy in GaLore, we consider a Top-$k$ strategy which places the $k$ ones at indices corresponding to $G$'s elements with the $k$ largest absolute values. We also consider a Rand-$k$ strategy which samples the sparse mask matrix $S$ from the uniform distribution on $\mathrm{SP}_{m,n}^k$ corresponding to GoLore. For convenience, we name the algorithm using Top-$k$ strategy as GaSare (**Gra**dient **Spar**se proj**e**ction), and the one using Rand-$k$ strategy as GoSare (**Gr**adient rand**o**m **Spar**se proj**e**ction). The concerned sparse subspace descent algorithms are described as in Alg. 6

### D.2. Notations and useful lemmas

We assume the model parameters consist of $N_L$ weight matrices. We use $X_\ell \in \mathbb{R}^{m_\ell \times n_\ell}$ to denote the $\ell$-th weight matrix and $x \in \mathbb{R}^d = (\mathrm{vec}(X_1)^\top, \cdots, \mathrm{vec}(X_{N_L})^\top)^\top$ to denote the vector collecting all the parameters, $d = \sum_{\ell=1}^{N_L} m_\ell n_\ell$. We assume GaSare/GoSare applies sparse mask in $\mathrm{Sp}_{m_\ell,n_\ell}^{k_\ell}$ to the $\ell$-th weight matrix and denote

$$\delta_\ell = \frac{k_\ell}{m_\ell n_\ell}, \quad \underline{\delta} = \min_{1 \le \ell \le N_L} \delta_\ell, \quad \overline{\delta} = \max_{1 \le \ell \le N_l} \delta_\ell.$$

We define $\tilde{M}_\ell^{(t)} = S_\ell^{(t)} \odot M_\ell^{(t)}$ and $\tilde{m} = (\mathrm{vec}(\tilde{M}_1)^\top, \cdots, \mathrm{vec}(\tilde{M}_{N_L})^\top)^\top$. While using Alg. 6 with MSGD, it holds that

$$\tilde{M}_\ell^{(t)} = \begin{cases} \beta_1 S_\ell^{(0)} \odot G_\ell^{(0)}, & t = 0; \\ S_\ell^{(t)} \odot \left( (1 - \beta_1) \tilde{M}_\ell^{(t-1)} + \beta_1 G_\ell^{(t)} \right), & t = k\tau, \ k \in \mathbb{N}^*; \\ (1 - \beta_1) \tilde{M}_\ell^{(t-1)} + \beta_1 S_\ell^{(t)} \odot G_\ell^{(t)}, & t = k\tau + r, \ k \in \mathbb{N}, \ 1 \le r < \tau; \end{cases}$$

and that

$$X_\ell^{(t+1)} = X_\ell^{(t)} - \eta \tilde{M}_\ell^{(t)}.$$

We use $E_{m,n}$ to denote the all-one $m \times n$ matrix, *i.e.*,

$$E_{m,n} = \begin{pmatrix} 1 & 1 & \cdots & 1 \\ 1 & 1 & \cdots & 1 \\ \vdots & \vdots & \ddots & \vdots \\ 1 & 1 & \cdots & 1 \end{pmatrix} \in \mathbb{R}^{m \times n}.$$

**Lemma 15** (Error of GaSare's projection)**:** Let $S$ be the Top-$k$ mask of $G \in \mathbb{R}^{m \times n}$, it holds that

$$\|S \odot G - G\|_F^2 \le \left(1 - \frac{k}{mn}\right) \|G\|_F^2.$$

*Proof.* Let $g_1, g_2, \cdots, g_{mn}$ be elements of $G$ such that $|g_1| \ge |g_2| \ge \cdots \ge |g_{mn}|$. It holds that

$$\begin{aligned} \|S \odot G - G\|_F^2 &= \sum_{i=1}^k (g_k - g_k)^2 + \sum_{i=k+1}^{mn} (0 - g_k)^2 \\ &= \sum_{i=k+1}^{mn} g_k^2 \\ &\le \left(1 - \frac{k}{mn}\right) \sum_{i=1}^{mn} g_k^2 \end{aligned}$$

$$= \left(1 - \frac{k}{mn}\right) \|\boldsymbol{G}\|_F^2,$$

where the inequality uses $\frac{1}{mn-k} \sum_{i=k+1}^{mn} g_i^2 \leq \frac{1}{k} \sum_{i=1}^{k} g_i^2$. □

**Lemma 16** (Error of GoSare's projection): Let $\boldsymbol{S} \sim \mathcal{U}(\mathrm{Sp}_{m,n}^k)$, it holds for all $\boldsymbol{G} \in \mathbb{R}^{m \times n}$ that

$$\mathbb{E}[\boldsymbol{S}] = \frac{k}{mn} \cdot \boldsymbol{E}_{m,n}, \tag{99}$$

and

$$\mathbb{E}[\|\boldsymbol{S} \odot \boldsymbol{G} - \boldsymbol{G}\|_F^2] = \left(1 - \frac{k}{mn}\right) \|\boldsymbol{G}\|_F^2. \tag{100}$$

*Proof.* To prove (99), it suffices to note that for any element $S_{i,j}$ in $\boldsymbol{S}$, it holds that

$$\mathbb{E}[S_{i,j}] = \mathbb{P}[S_{i,j} = 1] = \frac{(mn-1)!/[(mn-k)!(k-1)!]}{(mn)!/[(mn-k)!k!]} = \frac{k}{mn}.$$

To prove (100), we have

$$\mathbb{E}[\|\boldsymbol{S} \odot \boldsymbol{G} - \boldsymbol{G}\|_F^2] = \sum_{1 \leq i \leq m, 1 \leq j \leq n} \mathbb{P}[S_{i,j} = 0]\boldsymbol{G}_{i,j}^2 = \left(1 - \frac{k}{mn}\right) \|\boldsymbol{G}\|_F^2.$$

□

### D.3. Non-convergence of GaSare

In this subsection, we present the non-convergence result of GaSare, similar to that of GaLore.

**Theorem 20** (Non-convergence of GaSare): There exists an objective function $f : \mathbb{R}^d \to \mathbb{R}$ satisfying Assumptions 1, 2, a stochastic gradient oracle $(F, \mathcal{D})$ satisfying Assumption 3, an initial point $\boldsymbol{x}^{(0)}$, a constant $\epsilon_0 > 0$ such that for GaSare with any sparsity level $k_\ell < m_\ell n_\ell$, subspace changing frequency $\tau$ and any subspace optimizer $\rho$ with arbitrary hyperparameters and any $t > 0$, it holds that

$$\|\nabla f(\boldsymbol{x}^{(t)})\|_2^2 \geq \epsilon_0.$$

*Proof.* Consider target function $f(\boldsymbol{X}) = \frac{L}{2}\|(\boldsymbol{pp}^\top)\odot\boldsymbol{X}\|_F^2$ where $L > 0$, $\boldsymbol{X} \in \mathbb{R}^{n \times n}$ with $n > 1$ and $\boldsymbol{p} = (1, 0, \cdots, 0)^\top \in \mathbb{R}^n$. It holds that

$$f(\boldsymbol{X}) = \frac{LX_{1,1}^2}{2} \geq 0,$$

thus $f$ satisfies Assumption 1. Since $\nabla f(\boldsymbol{X}) = L(\boldsymbol{pp}^\top) \odot \boldsymbol{X}$, it holds that

$$\|\nabla f(\boldsymbol{X}) - \nabla f(\boldsymbol{Y})\|_F = L\|(\boldsymbol{pp}^\top) \odot (\boldsymbol{X} - \boldsymbol{Y})\|_F \leq L\|\boldsymbol{X} - \boldsymbol{Y}\|_F,$$

thus $f$ satisfies Assumption 2.

Consider the following stochastic gradient oracle:

$$F(\boldsymbol{X};\xi) = f(\boldsymbol{X}) + \xi\tilde{\sigma} \cdot \mathrm{tr}(\boldsymbol{QX}), \quad \text{and} \quad \mathbb{P}_{\xi\sim\mathcal{D}}[\xi = 1] = \mathbb{P}_{\xi\sim\mathcal{D}}[\xi = -1] = 0.5,$$

where $\tilde{\sigma} = \sigma/\sqrt{n^2(n^2 - 1)/2}$ and

$$\boldsymbol{Q} = \begin{pmatrix} 0 & \sqrt{n} & \cdots & \sqrt{n^2 - n} \\ \sqrt{1} & \sqrt{n+1} & \cdots & \sqrt{n^2 - n + 1} \\ \vdots & \vdots & \ddots & \vdots \\ \sqrt{n-1} & \sqrt{2n-1} & \cdots & \sqrt{n^2 - 1} \end{pmatrix} \in \mathbb{R}^{n \times n}.$$

Note that $\nabla F(\boldsymbol{X};\xi) = \nabla f(\boldsymbol{X}) + \xi\tilde{\sigma}\boldsymbol{Q}$, it holds for any $\boldsymbol{X} \in \mathbb{R}^{n \times n}$ that

$$\mathbb{E}_{\xi \sim \mathcal{D}}[\nabla F(\boldsymbol{X};\xi)] = \nabla f(\boldsymbol{X})$$

$$\mathbb{E}_{\xi \sim \mathcal{D}}[\|\nabla F(\boldsymbol{X};\xi) - \nabla f(\boldsymbol{X})\|_F^2] = \tilde{\sigma}^2 \|\boldsymbol{Q}\|_F^2 = \frac{\sigma^2}{n^2(n^2-1)/2} \cdot \sum_{i=1}^{n^2-1} i = \sigma^2,$$

thus oracle $(F, \mathcal{D})$ satisfies Assumption 3.

Consider the initial point $\boldsymbol{X}^{(0)}$ with $X_{1,1}^{(0)} = \lambda$, where $0 < \lambda < \tilde{\sigma}/L$ is a scalar. We show that GaSare with the above objective function $f$, stochastic gradient oracle $(F, \mathcal{D})$, initial point $\boldsymbol{X}^{(0)}$, arbitrary sparsity level $0 < k < n^2$, arbitrary subspace changing frequency $\tau$ and arbitrary subspace optimizer $\rho$, can only output points $\boldsymbol{X}^{(t)}$ with $\|\nabla f(\boldsymbol{X}^{(t)})\|_F^2 \geq \epsilon_0$ for $\epsilon_0 = L^2\lambda^2 > 0$.

When $\tau \mid t$, GaSare recomputes the sparse mask matrix at iteration $t$. If $X_{1,1}^{(t)} = \lambda$, the stochastic gradient is given by

$$\boldsymbol{G}^{(t)} = L(\boldsymbol{pp}^\top) \odot \boldsymbol{X} + \xi^{(t)}\tilde{\sigma}\boldsymbol{Q}.$$

since $L\lambda < \tilde{\sigma}$, the Top-$k$ mask $\boldsymbol{S} \in \mathbb{R}^{n \times n}$ satisfies

$$\text{vec}(\boldsymbol{S}) = (\underbrace{0, 0, \cdots, 0}_{(n^2-k)\times}, \underbrace{1, 1, \cdots, 1}_{k\times})^\top \in \mathbb{R}^{n^2},$$

Using this mask matrix, the subspace updates in the following $\tau$ iterations is as

$$\boldsymbol{X}^{(t+\Delta_t)} = \boldsymbol{X}^{(t)} + \boldsymbol{S}^{(t)} \odot \left(\sum_{s=0}^{\Delta_t - 1} \rho^{(t+s)}(\boldsymbol{S}^{(t)} \odot \boldsymbol{G}^{(t)})\right) \quad \Rightarrow \quad X_{1,1}^{(t+\Delta_t)} = X_{1,1}^{(t)} = \lambda,$$

for $\Delta_t = 1, 2, \cdots, \tau$. Since $X_{1,1}^{(0)} = \lambda$, it holds for all $t > 0$ that $X_{1,1}^{(t)} = \lambda$ and thus

$$\|\nabla f(\boldsymbol{X}^{(t)})\|_F^2 = L^2\lambda^2 = \epsilon_0.$$

$\square$

## D.4. Convergence of deterministic GaSare

In this subsection, we prove the convergence properties of GaSare with deterministic gradients. The results and proofs are similar to those of deterministic GaLore in Appendix B.3.

**Lemma 17** (Momentum contraction)**:** In deterministic GaSare using MSGD (Alg. 6), if $0 < \beta_1 \leq 1$, term $\tilde{\boldsymbol{M}}_\ell^{(t)}$ has the following contraction properties:

- When $t = 0$, it holds that

$$\|\tilde{\boldsymbol{M}}_\ell^{(0)} - \nabla_\ell f(\boldsymbol{X}^{(0)})\|_F^2 \leq (\tau - 1)(1 - \delta_\ell\beta_1) \sum_{r=0}^{\tau-2} \|\nabla_\ell f(\boldsymbol{x}^{(r+1)}) - \nabla_\ell f(\boldsymbol{x}^{(r)})\|_F^2$$

$$+ \frac{2(1 - \delta_\ell\beta_1)}{\tau} \sum_{r=0}^{\tau-1} \|\nabla_\ell f(\boldsymbol{x}^{(r)})\|_F^2; \tag{101}$$

- When $t = k\tau$, $k \in \mathbb{N}^*$, it holds that

$$\|\tilde{\boldsymbol{M}}_\ell^{(t)} - \nabla_\ell f(\boldsymbol{x}^{(t)})\|_F^2 - \left(1 - \left(1 - \frac{\delta_\ell}{4}\right)\beta_1\right) \|\tilde{\boldsymbol{M}}_\ell^{(t-1)} - \nabla_\ell f(\boldsymbol{x}^{(t-1)})\|_F^2$$

$$\leq \frac{2(1 - \delta_\ell)}{\tau} \sum_{r=0}^{\tau-1} \|\nabla_l f(\boldsymbol{x}^{(k\tau+r)})\|_F^2 + \frac{5(1 - \beta_1)}{\delta_\ell\beta_1} \|\nabla_\ell f(\boldsymbol{x}^{(t)}) - \nabla_\ell f(\boldsymbol{x}^{(t-1)})\|_F^2$$

$$+ (\tau - 1)(1 - \delta_\ell) \sum_{r=0}^{\tau-2} \|\nabla_\ell f(\boldsymbol{x}^{(k\tau+r+1)}) - \nabla_\ell f(\boldsymbol{x}^{(k\tau+r)})\|_F^2; \tag{102}$$

- When $t = k\tau + r$, $k \in \mathbb{N}$, $1 \le r < \tau$, it holds that

$$\mathbb{E}[\|\tilde{\boldsymbol{M}}_\ell^{(t)} - \nabla_\ell f(\boldsymbol{x}^{(t)})\|_F^2] - \left(1 - \left(1 - \frac{\delta_\ell}{4}\right)\beta_1\right)\mathbb{E}[\|\tilde{\boldsymbol{M}}_\ell^{(t-1)} - \nabla_\ell f(\boldsymbol{x}^{(t-1)})\|_F^2]$$

$$\le \left(1 - \frac{\delta_\ell}{2}\right)\beta_1\mathbb{E}[\|\nabla_\ell f(\boldsymbol{x}^{(t)})\|_F^2] + \frac{5(1-\beta_1)}{\delta_\ell\beta_1}\mathbb{E}[\|\nabla_\ell f(\boldsymbol{x}^{(t)}) - \nabla_\ell f(\boldsymbol{x}^{(t-1)})\|_F^2]$$

$$+ \frac{10r\beta_1}{\delta_\ell}\sum_{i=1}^r \mathbb{E}[\|\nabla_\ell f(\boldsymbol{x}^{(k\tau+i)}) - \nabla_\ell f(\boldsymbol{x}^{(k\tau+i-1)})\|_F^2]. \tag{103}$$

*Proof.* For convenience we use $\boldsymbol{E}$ to denote $\boldsymbol{E}_{m_\ell, n_\ell}$. When $t = 0$, we have

$$\|\tilde{\boldsymbol{M}}_\ell^{(0)} - \nabla_\ell f(\boldsymbol{x}^{(0)})\|_F^2 = \|\beta_1(\boldsymbol{S}_\ell^{(0)} - \boldsymbol{E}) \odot \nabla_\ell f(\boldsymbol{x}^{(0)}) - (1-\beta_1)\nabla_\ell f(\boldsymbol{x}^{(0)})\|_F^2$$

$$\le \beta_1(1-\delta_\ell)\|\nabla_\ell f(\boldsymbol{x}^{(0)})\|_F^2 + (1-\beta_1)\|\nabla_\ell f(\boldsymbol{x}^{(0)})\|_F^2$$

$$= (1 - \delta_\ell\beta_1)\|\nabla_\ell f(\boldsymbol{x}^{(0)})\|_F^2, \tag{104}$$

where the inequality uses Lemma 15 and Jensen's inequality. Applying Lemma 2 to (104) yields (101).

When $t = k\tau$, $k \in \mathbb{N}^*$, we have

$$\|\tilde{\boldsymbol{M}}_\ell^{(t)} - \nabla_\ell f(\boldsymbol{x}^{(t)})\|_F^2$$

$$= \|\boldsymbol{S}_\ell^{(t)} \odot [(1-\beta_1)\tilde{\boldsymbol{M}}_\ell^{(t-1)} + \beta_1\boldsymbol{G}_\ell^{(t)} - \nabla_\ell f(\boldsymbol{x}^{(t)})] - (\boldsymbol{E} - \boldsymbol{S}_\ell^{(t)}) \odot \nabla_\ell f(\boldsymbol{x}^{(t)})\|_F^2$$

$$= \|\boldsymbol{S}_\ell^{(t)} \odot [(1-\beta_1)(\tilde{\boldsymbol{M}}_\ell^{(t-1)} - \nabla_\ell f(\boldsymbol{x}^{(t)}))]\|_F^2 + \|(\boldsymbol{E} - \boldsymbol{S}_\ell^{(t)}) \odot \nabla_\ell f(\boldsymbol{x}^{(t)})\|_F^2$$

$$\le \|(1-\beta_1)(\tilde{\boldsymbol{M}}_\ell^{(t-1)} - \nabla_\ell f(\boldsymbol{x}^{(t)}))\|_F^2 + (1-\delta_\ell)\|\nabla_\ell f(\boldsymbol{x}^{(t)})\|_F^2, \tag{105}$$

where the inequality uses Lemma 15. By Young's inequality, we have

$$\|\tilde{\boldsymbol{M}}_\ell^{(t-1)} - \nabla_\ell f(\boldsymbol{x}^{(t)})\|_F^2$$

$$= \|(\tilde{\boldsymbol{M}}_\ell^{(t-1)} - \nabla_\ell f(\boldsymbol{x}^{(t-1)})) - (\nabla_\ell f(\boldsymbol{x}^{(t)}) - \nabla_\ell f(\boldsymbol{x}^{(t-1)}))\|_F^2$$

$$\le \left(1 + \frac{\delta_\ell\beta_1}{4}\right)\|\tilde{\boldsymbol{M}}_\ell^{(t-1)} - \nabla_\ell f(\boldsymbol{x}^{(t-1)})\|_F^2 + \left(1 + \frac{4}{\delta_\ell\beta_1}\right)\|\nabla_\ell f(\boldsymbol{x}^{(t)}) - \nabla_\ell f(\boldsymbol{x}^{(t-1)})\|_F^2. \tag{106}$$

Applying Lemma 2 and (106) to (105) yields (102).

When $t = k\tau + r$, $k \in \mathbb{N}$, $1 \le r < \tau$, we have

$$\|\tilde{\boldsymbol{M}}_\ell^{(t)} - \nabla_\ell f(\boldsymbol{x}^{(t)})\|_F^2$$

$$= \|(1-\beta_1)(\tilde{\boldsymbol{M}}_\ell^{(t-1)} - \nabla_\ell f(\boldsymbol{x}^{(t)})) + \beta_1(\boldsymbol{S}_\ell^{(t)} - \boldsymbol{E}) \odot \nabla_\ell f(\boldsymbol{x}^{(t)})\|_F^2$$

$$\le (1-\beta_1)\|\tilde{\boldsymbol{M}}_\ell^{(t-1)} - \nabla_\ell f(\boldsymbol{x}^{(t)})\|_F^2 + \beta_1\|(\boldsymbol{E} - \boldsymbol{S}_\ell^{(k\tau)}) \odot \nabla_\ell f(\boldsymbol{x}^{(t)})\|_F^2, \tag{107}$$

where the inequality uses Jensen's inequality and $\boldsymbol{S}_\ell^{(t)} = \boldsymbol{S}_\ell^{(t-1)} = \cdots = \boldsymbol{S}_\ell^{(k\tau)}$. The first term can be similarly upper bounded as (106). For the second term, we have

$$(\boldsymbol{E} - \boldsymbol{S}_\ell^{(k\tau)}) \odot \nabla_\ell f(\boldsymbol{x}^{(t)})\|_F^2$$

$$\le \left(1 + \frac{\delta_\ell}{4}\right)\|(\boldsymbol{E} - \boldsymbol{S}_\ell^{(k\tau)}) \odot \nabla_\ell f(\boldsymbol{x}^{(k\tau)})\|_F^2$$

$$+ \left(1 + \frac{4}{\delta_\ell}\right)\|(\boldsymbol{E} - \boldsymbol{S}_\ell^{(k\tau)}) \odot (\nabla_\ell f(\boldsymbol{x}^{(t)}) - \nabla_\ell f(\boldsymbol{x}^{(k\tau)}))\|_F^2$$

$$\le \left(1 + \frac{\delta_\ell}{4}\right)(1-\delta_\ell)\|\nabla_\ell f(\boldsymbol{x}^{(k\tau)})\|_F^2 + \frac{5}{\delta_\ell}\|\nabla_\ell f(\boldsymbol{x}^{(t)}) - \nabla_\ell f(\boldsymbol{x}^{(k\tau)})\|_F^2, \tag{108}$$

where the first inequality uses Young's inequality and the second inequality uses Lemma 15. By Young's inequality, we have

$$\|\nabla_\ell f(\boldsymbol{x}^{(k\tau)})\|_F^2 \leq \left(1 + \frac{\delta_\ell}{4}\right)\|\nabla_\ell f(\boldsymbol{x}^{(t)})\|_F^2 + \left(1 + \frac{4}{\delta_\ell}\right)\|\nabla_\ell f(\boldsymbol{x}^{(t)}) - \nabla_\ell f(\boldsymbol{x}^{(k\tau)})\|_F^2. \tag{109}$$

Note that $t = k\tau + r$, we further have

$$\|\nabla_\ell f(\boldsymbol{x}^{(t)}) - \nabla_\ell f(\boldsymbol{x}^{(k\tau)})\|_F^2 = \left\|\sum_{i=1}^{r} \nabla_\ell f(\boldsymbol{x}^{(k\tau+i)}) - \nabla_\ell f(\boldsymbol{x}^{(k\tau+i-1)})\right\|_F^2$$

$$\leq r \sum_{i=1}^{r} \|\nabla_\ell f(\boldsymbol{x}^{(k\tau+i)}) - \nabla_\ell f(\boldsymbol{x}^{(k\tau+i-1)})\|_F^2, \tag{110}$$

where the inequality uses Cauchy's inequality. Applying (109)(110) to (108) yields

$$(\boldsymbol{E} - \boldsymbol{S}_\ell^{(k\tau)}) \odot \nabla_\ell f(\boldsymbol{x}^{(t)})\|_F^2$$

$$\leq \left(1 - \frac{\delta_\ell}{2}\right)\|\nabla_\ell f(\boldsymbol{x}^{(t)})\|_F^2 + \frac{10r}{\delta_\ell} \sum_{i=1}^{r} \|\nabla_\ell f(\boldsymbol{x}^{(k\tau+i)}) - \nabla_\ell f(\boldsymbol{x}^{(k\tau+i-1)})\|_F^2. \tag{111}$$

Applying (106)(111) to (107) yields (103). $\qquad\square$

Based on Lemma 17, we can prove the convergence properties of deterministic GaSare similarly as the proofs of Lemma 7, Theorem 10 and Corollary 11. Below we directly present the final convergence results.

**Theorem 21** (Convergence of deterministic GaSare)**:** Under Assumptions 1-2, if hyperparameters

$$0 < \beta_1 \leq 1, \quad \tau \geq \frac{64}{3\beta_1\underline{\delta}}, \quad 0 < \eta \leq \min\left\{\frac{1}{4L}, \sqrt{\frac{3\underline{\delta}\beta_1^2}{80L^2}}, \sqrt{\frac{3\underline{\delta}}{80\tau^2 L^2}}, \sqrt{\frac{3\beta_1}{16\tau L^2}}\right\},$$

GaSare using deterministic gradients and MSGD (Alg. 6) converges as

$$\frac{1}{K\tau} \sum_{t=0}^{K\tau-1} \|\nabla f(\boldsymbol{x}^{(t)})\|_2^2 \leq \frac{16\Delta}{\underline{\delta}\eta K\tau}$$

for any $K \geq 1$, where $\Delta = f(\boldsymbol{x}^{(0)}) - \inf_{\boldsymbol{x}} f(\boldsymbol{x})$. If $T \geq 64/(3\underline{\delta})$ and we further choose

$$\beta_1 = 1$$

$$\tau = \left\lceil \frac{64}{3\underline{\delta}\beta_1} \right\rceil$$

$$\eta = \left(4L + \sqrt{\frac{80L^2}{3\underline{\delta}\beta_1^2}} + \sqrt{\frac{80\tau^2 L^2}{3\underline{\delta}}} + \sqrt{\frac{16\tau L^2}{3\beta_1}}\right)^{-1},$$

GaSare using deterministic gradients and MSGD (Alg. 6) converges as

$$\frac{1}{T} \sum_{t=0}^{T-1} \|\nabla f(\boldsymbol{x}^{(t)})\|_2^2 = \mathcal{O}\left(\frac{L\Delta}{\underline{\delta}^{5/2}T}\right).$$

Consequently, the computation complexity to reach an $\varepsilon$-accurate solution $\boldsymbol{x}$ such that $\|\nabla f(\boldsymbol{x})\|_2^2 \leq \varepsilon$ is $\mathcal{O}\left(\frac{L\Delta}{\underline{\delta}^{5/2}\varepsilon} + \frac{1}{\underline{\delta}}\right)$.

### D.5. Convergence of large-batch GaSare

In this subsection, we present the convergence properties of GaSare with large-batch stochastic gradients. The results and proofs are similar to those of large-batch GaLore in Appendix B.4.

**Lemma 18** (Momentum contraction)**:** Under Assumption 3, in large-batch GaSare using MSGD (Alg. 6), if $0 < \beta_1 \leq 1$, term $\tilde{\boldsymbol{M}}_\ell^{(t)}$ has the following contraction properties:

- When $t = 0$, it holds that

$$
\mathbb{E}[\|\tilde{\boldsymbol{M}}_\ell^{(0)} - \nabla_\ell f(\boldsymbol{X}^{(0)})\|_F^2] \leq 2(\tau-1)(1-\delta_\ell\beta_1)\sum_{r=0}^{\tau-2}\mathbb{E}[\|\nabla_\ell f(\boldsymbol{x}^{(r+1)}) - \nabla_\ell f(\boldsymbol{x}^{(r)})\|_F^2]
$$
$$
+ \frac{4(1-\delta_\ell\beta_1)}{\tau}\sum_{r=0}^{\tau-1}\mathbb{E}[\|\nabla_\ell f(\boldsymbol{x}^{(r)})\|_F^2] + \frac{4\beta_1\sigma_\ell^2}{\mathcal{B}}; \tag{112}
$$

- When $t = k\tau$, $k \in \mathbb{N}^*$, it holds that

$$
\mathbb{E}[\|\tilde{\boldsymbol{M}}_\ell^{(t)} - \nabla_\ell f(\boldsymbol{x}^{(t)})\|_F^2] - \left(1 - \left(1 - \frac{\delta_\ell}{4}\right)\beta_1\right)\mathbb{E}[\|\tilde{\boldsymbol{M}}_\ell^{(t-1)} - \nabla_\ell f(\boldsymbol{x}^{(t-1)})\|_F^2]
$$
$$
\leq \frac{4(1-\delta_\ell)}{\tau}\sum_{r=0}^{\tau-1}\mathbb{E}[\|\nabla_l f(\boldsymbol{x}^{(k\tau+r)})\|_F^2] + \frac{5(1-\beta_1)}{\delta_\ell\beta_1}\mathbb{E}[\|\nabla_\ell f(\boldsymbol{x}^{(t)}) - \nabla_\ell f(\boldsymbol{x}^{(t-1)})\|_F^2]
$$
$$
+ 2(\tau-1)(1-\delta_\ell)\sum_{r=0}^{\tau-2}\mathbb{E}[\|\nabla_\ell f(\boldsymbol{x}^{(k\tau+r+1)}) - \nabla_\ell f(\boldsymbol{x}^{(k\tau+r)})\|_F^2] + \frac{5\sigma_\ell^2}{\mathcal{B}}; \tag{113}
$$

- When $t = k\tau + r$, $k \in \mathbb{N}$, $1 \leq r < \tau$, it holds that

$$
\mathbb{E}[\|\tilde{\boldsymbol{M}}_\ell^{(t)} - \nabla_\ell f(\boldsymbol{x}^{(t)})\|_F^2] - \left(1 - \left(1 - \frac{\delta_\ell}{4}\right)\beta_1\right)\mathbb{E}[\|\tilde{\boldsymbol{M}}_\ell^{(t-1)} - \nabla_\ell f(\boldsymbol{x}^{(t-1)})\|_F^2]
$$
$$
\leq \left(1 - \frac{\delta_\ell}{2}\right)\beta_1\mathbb{E}[\|\nabla_\ell f(\boldsymbol{x}^{(t)})\|_F^2] + \frac{5(1-\beta_1)}{\delta_\ell\beta_1}\mathbb{E}[\|\nabla_\ell f(\boldsymbol{x}^{(t)}) - \nabla_\ell f(\boldsymbol{x}^{(t-1)})\|_F^2]
$$
$$
+ \frac{15r\beta_1}{\delta_\ell}\sum_{i=1}^{r}\mathbb{E}[\|\nabla_\ell f(\boldsymbol{x}^{(k\tau+i)}) - \nabla_\ell f(\boldsymbol{x}^{(k\tau+i-1)})\|_F^2] + \left(\frac{11\beta_1}{\delta_\ell\mathcal{B}} + \beta_1^2\right)\sigma_\ell^2. \tag{114}
$$

*Proof.* For convenience we use $\boldsymbol{E}$ to denote $\boldsymbol{E}_{m_\ell, n_\ell}$. When $t = 0$, we have

$$
\mathbb{E}[\|\tilde{\boldsymbol{M}}_\ell^{(0)} - \nabla_\ell f(\boldsymbol{x}^{(0)})\|_F^2]
$$
$$
= \mathbb{E}[\|\beta_1\boldsymbol{S}_\ell^{(0)} \odot \boldsymbol{G}_\ell^{(0)} - \nabla_\ell f(\boldsymbol{x}^{(0)})\|_F^2]
$$
$$
= \mathbb{E}[\|\beta_1(\boldsymbol{S}_\ell^{(0)} - \boldsymbol{E}) \odot \boldsymbol{G}_\ell^{(0)} + \beta_1(\boldsymbol{G}_\ell^{(0)} - \nabla_\ell f(\boldsymbol{x}^{(0)})) - (1-\beta_1)\nabla_\ell f(\boldsymbol{x}^{(0)})\|_F^2]
$$
$$
\leq \beta_1\mathbb{E}[\|(\boldsymbol{S}_\ell^{(0)} - \boldsymbol{E}) \odot \boldsymbol{G}_\ell^{(0)} + \boldsymbol{G}_\ell^{(0)} - \nabla_\ell f(\boldsymbol{x}^{(0)})\|_F^2] + (1-\beta_1)\|\nabla_\ell f(\boldsymbol{x}^{(0)})\|_F^2, \tag{115}
$$

where the inequality uses Jensen's inequality. For the first term we have

$$
\mathbb{E}[\|(\boldsymbol{S}_\ell^{(0)} - \boldsymbol{E}) \odot \boldsymbol{G}_\ell^{(0)} + \boldsymbol{G}_\ell^{(0)} - \nabla_\ell f(\boldsymbol{x}^{(0)})\|_F^2]
$$
$$
\leq 2\mathbb{E}[\|(\boldsymbol{E} - \boldsymbol{S}_\ell^{(0)}) \odot \boldsymbol{G}_\ell^{(0)}\|_F^2] + 2\mathbb{E}[\|\boldsymbol{G}_\ell^{(0)} - \nabla_\ell f(\boldsymbol{x}^{(0)})\|_F^2]
$$
$$
\leq 2(1-\delta_\ell)\mathbb{E}[\|\boldsymbol{G}_\ell\|_F^2] + 2\mathbb{E}[\|\boldsymbol{G}_\ell^{(0)} - \nabla_\ell f(\boldsymbol{x}^{(0)})\|_F^2]
$$
$$
\leq 2(1-\delta_\ell)\|\nabla_\ell f(\boldsymbol{x}^{(0)})\|_F^2 + \frac{(4-2\delta_\ell)\sigma_\ell^2}{\mathcal{B}}, \tag{116}
$$

where the first inequality uses Cauchy's inequality, the second inequality uses Lemma 15, the third inequality uses $\mathbb{E}[\|\boldsymbol{G}_\ell^{(0)} - \nabla_\ell f(\boldsymbol{x}^{(0)})\|_F^2] \leq \sigma_\ell^2/\mathcal{B}$ (Assumption 3). Applying (116) and Lemma 2 to (115) yields (112).

When $t = k\tau$, $k \in \mathbb{N}^*$, we have

$$
\mathbb{E}[\|\tilde{\boldsymbol{M}}_\ell^{(t)} - \nabla_\ell f(\boldsymbol{x}^{(t)})\|_F^2]
$$

$$=\mathbb{E}[\|\boldsymbol{S}_\ell^{(t)} \odot [(1-\beta_1)\tilde{\boldsymbol{M}}_\ell^{(t-1)} + \beta_1 \boldsymbol{G}_\ell^{(t)} - \nabla_\ell f(\boldsymbol{x}^{(t)})] - (\boldsymbol{E} - \boldsymbol{S}_\ell^{(t)}) \odot \nabla_\ell f(\boldsymbol{x}^{(t)})\|_F^2]$$

$$=\mathbb{E}[\|\boldsymbol{S}_\ell^{(t)} \odot [(1-\beta_1)\tilde{\boldsymbol{M}}_\ell^{(t-1)} + \beta_1 \boldsymbol{G}_\ell^{(t)} - \nabla_\ell f(\boldsymbol{x}^{(t)})]\|_F^2] + \mathbb{E}[\|(\boldsymbol{E} - \boldsymbol{S}_\ell^{(t)}) \odot \nabla_\ell f(\boldsymbol{x}^{(t)})\|_F^2]. \tag{117}$$

We further have

$$\mathbb{E}[\|\boldsymbol{S}_\ell^{(t)} \odot [(1-\beta_1)\tilde{\boldsymbol{M}}_\ell^{(t-1)} + \beta_1 \boldsymbol{G}_\ell^{(t)} - \nabla_\ell f(\boldsymbol{x}^{(t)})]\|_F^2]$$

$$\leq \mathbb{E}[\|(1-\beta_1)\tilde{\boldsymbol{M}}_\ell^{(t-1)} + \beta_1 \boldsymbol{G}_\ell^{(t)} - \nabla_\ell f(\boldsymbol{x}^{(t)})\|_F^2]$$

$$= \mathbb{E}[\|(1-\beta_1)(\tilde{\boldsymbol{M}}_\ell^{(t-1)} - \nabla_\ell f(\boldsymbol{x}^{(t)})) + \beta_1(\boldsymbol{G}_\ell^{(t)} - \nabla_\ell f(\boldsymbol{x}^{(t)}))\|_F^2]$$

$$\leq \mathbb{E}[\|(1-\beta_1)(\tilde{\boldsymbol{M}}_\ell^{(t-1)} - \nabla_\ell f(\boldsymbol{x}^{(t)}))\|_F^2] + \beta_1^2 \mathbb{E}[\|\boldsymbol{G}_\ell^{(t)} - \nabla_\ell f(\boldsymbol{x}^{(t)})\|_F^2], \tag{118}$$

where the last inequality uses the unbiasedness of $\boldsymbol{G}_\ell^{(t)}$ (Assumption 3). By Young's inequality, we have

$$\mathbb{E}[\|\tilde{\boldsymbol{M}}_\ell^{(t-1)} - \nabla_\ell f(\boldsymbol{x}^{(t)})\|_F^2]$$

$$=\mathbb{E}[\|(\tilde{\boldsymbol{M}}_\ell^{(t-1)} - \nabla_\ell f(\boldsymbol{x}^{(t-1)})) - (\nabla_\ell f(\boldsymbol{x}^{(t)}) - \nabla_\ell f(\boldsymbol{x}^{(t-1)}))\|_F^2]$$

$$\leq \left(1 + \frac{\delta_\ell \beta_1}{4}\right) \mathbb{E}[\|\tilde{\boldsymbol{M}}_\ell^{(t-1)} - \nabla_\ell f(\boldsymbol{x}^{(t-1)})\|_F^2] + \left(1 + \frac{4}{\delta_\ell \beta_1}\right) \mathbb{E}[\|\nabla_\ell f(\boldsymbol{x}^{(t)}) - \nabla_\ell f(\boldsymbol{x}^{(t-1)})\|_F^2]. \tag{119}$$

Applying (119) to (118) yields

$$\mathbb{E}[\|\boldsymbol{S}_\ell^{(t)} \odot [(1-\beta_1)\tilde{\boldsymbol{M}}_\ell^{(t-1)} + \beta_1 \boldsymbol{G}_\ell^{(t)} - \nabla_\ell f(\boldsymbol{x}^{(t)})]\|_F^2]$$

$$\leq \left(1 - \left(1 - \frac{\delta_\ell}{4}\right)\beta_1\right) \mathbb{E}[\|\tilde{\boldsymbol{M}}_\ell^{(t-1)} - \nabla_\ell f(\boldsymbol{x}^{(t-1)})\|_F^2] + \frac{\beta_1^2 \sigma^2}{\mathcal{B}}$$

$$+ \frac{5(1-\beta_1)}{\delta_\ell \beta_1} \mathbb{E}[\|\nabla_\ell f(\boldsymbol{x}^{(t)}) - \nabla_\ell f(\boldsymbol{x}^{(t-1)})\|_F^2]. \tag{120}$$

For the second term in (117), we have

$$\mathbb{E}[\|(\boldsymbol{E} - \boldsymbol{S}_\ell^{(t)}) \odot \nabla_\ell f(\boldsymbol{x}^{(t)})\|_F^2]$$

$$\leq 2\mathbb{E}[\|(\boldsymbol{E} - \boldsymbol{S}_\ell^{(t)}) \odot \boldsymbol{G}_\ell^{(t)}\|_F^2] + 2\mathbb{E}[\|(\boldsymbol{E} - \boldsymbol{S}_\ell^{(t)}) \odot (\boldsymbol{G}_\ell^{(t)} - \nabla_\ell f(\boldsymbol{x}^{(t)}))\|_F^2]$$

$$\leq 2(1 - \delta_\ell)\mathbb{E}[\|\boldsymbol{G}_\ell^{(t)}\|_F^2] + 2\mathbb{E}[\|\boldsymbol{G}_\ell^{(t)} - \nabla_\ell f(\boldsymbol{x}^{(t)})\|_F^2]$$

$$\leq 2(1 - \delta_\ell)\mathbb{E}[\|\nabla_\ell f(\boldsymbol{x}^{(t)})\|_F^2] + \frac{4\sigma_\ell^2}{\mathcal{B}}, \tag{121}$$

where the first inequality uses Cauchy's inequality, the second inequality uses Lemma 15, the third inequality uses Assumption 3. Applying (120)(121) to (117) and using Lemma 2 yields (113).

When $t = k\tau + r$, $k \in \mathbb{N}$, $1 \leq r < \tau$, we have

$$\mathbb{E}[\|\tilde{\boldsymbol{M}}_\ell^{(t)} - \nabla_\ell f(\boldsymbol{x}^{(t)})\|_F^2]$$

$$=\mathbb{E}[\|(1-\beta_1)(\tilde{\boldsymbol{M}}_\ell^{(t-1)} - \nabla_\ell f(\boldsymbol{x}^{(t)})) + \beta_1(\boldsymbol{S}_\ell^{(t)} \odot \boldsymbol{G}_\ell^{(t)} - \nabla_\ell f(\boldsymbol{x}^{(t)}))\|_F^2]$$

$$=\mathbb{E}[\|(1-\beta_1)(\tilde{\boldsymbol{M}}_\ell^{(t-1)} - \nabla_\ell f(\boldsymbol{x}^{(t)})) + \beta_1(\boldsymbol{S}_\ell^{(t)} - \boldsymbol{E}) \odot \nabla_\ell f(\boldsymbol{x}^{(t)})\|_F^2]$$

$$+ \beta_1^2 \mathbb{E}[\|\boldsymbol{S}_\ell^{(t)} \odot (\boldsymbol{G}_\ell^{(t)} - \nabla_\ell f(\boldsymbol{x}^{(t)}))\|_F^2]$$

$$\leq (1-\beta_1)\mathbb{E}[\|\tilde{\boldsymbol{M}}_\ell^{(t-1)} - \nabla_\ell f(\boldsymbol{x}^{(t)})\|_F^2] + \beta_1 \mathbb{E}[\|(\boldsymbol{E} - \boldsymbol{S}_\ell^{(t)}) \odot \nabla_\ell f(\boldsymbol{x}^{(t)})\|_F^2$$

$$+ \beta_1^2 \mathbb{E}[\|\boldsymbol{S}_\ell^{(t)} \odot (\boldsymbol{G}_\ell^{(t)} - \nabla_\ell f(\boldsymbol{x}^{(t)}))\|_F^2], \tag{122}$$

where the second equality uses the unbiasedness of $\boldsymbol{G}_\ell^{(t)}$ and the independence implied by $\boldsymbol{S}_\ell^{(t)} = \boldsymbol{S}_\ell^{(t-1)}$, the inequality uses Jensen's inequality. The first term is similarly bounded as (119). For the second term, we have

$$\mathbb{E}[\|(\boldsymbol{E} - \boldsymbol{S}_\ell^{(k\tau)}) \odot \nabla_\ell f(\boldsymbol{x}^{(t)})\|_F^2]$$

$$\leq \left(1 + \frac{\delta_\ell}{4}\right) \mathbb{E}[\|(\boldsymbol{E} - \boldsymbol{S}_\ell^{(k\tau)}) \odot \boldsymbol{G}_\ell^{(k\tau)}\|_F^2]$$

$$+ \left(1 + \frac{4}{\delta_\ell}\right) \mathbb{E}[\|(\boldsymbol{E} - \boldsymbol{S}_\ell^{(k\tau)}) \odot (\nabla_\ell f(\boldsymbol{x}^{(t)}) - \boldsymbol{G}_\ell^{(k\tau)})\|_F^2]$$

$$\leq \left(1 - \frac{3\delta_\ell}{4}\right) \mathbb{E}[\|\boldsymbol{G}_\ell^{(k\tau)}\|_F^2] + 2\left(1 + \frac{4}{\delta_\ell}\right) \mathbb{E}[\|\boldsymbol{G}_\ell^{(k\tau)} - \nabla_\ell f(\boldsymbol{x}^{(k\tau)})\|_F^2]$$

$$+ 2\left(1 + \frac{4}{\delta_\ell}\right) \mathbb{E}[\|\nabla_\ell f(\boldsymbol{x}^{(t)}) - \nabla_\ell f(\boldsymbol{x}^{(k\tau)})\|_F^2], \tag{123}$$

where the first inequality uses Young's inequality, the second inequality uses Lemma 15 and Cauchy's inequality. We further have

$$\left(1 - \frac{3\delta_\ell}{4}\right) \mathbb{E}[\|\boldsymbol{G}_\ell^{(k\tau)}\|_F^2] + 2\left(1 + \frac{4}{\delta_\ell}\right) \mathbb{E}[\|\boldsymbol{G}_\ell^{(k\tau)} - \nabla_\ell f(\boldsymbol{x}^{(k\tau)})\|_F^2]$$

$$\leq \left(1 - \frac{3\delta_\ell}{4}\right) \mathbb{E}[\|\nabla_\ell f(\boldsymbol{x}^{(k\tau)})\|_F^2] + \frac{11}{\delta_\ell} \mathbb{E}[\|\boldsymbol{G}_\ell^{(k\tau)} - \nabla_\ell f(\boldsymbol{x}^{(k\tau)})\|_F^2]$$

$$\leq \left(1 - \frac{3\delta_\ell}{4}\right) \mathbb{E}[\|\nabla_\ell f(\boldsymbol{x}^{(k\tau)})\|_F^2] + \frac{11\sigma_\ell^2}{\delta_\ell \mathcal{B}}$$

$$\leq \left(1 - \frac{\delta_\ell}{2}\right) \mathbb{E}[\|\nabla_\ell f(\boldsymbol{x}^{(t)})\|_F^2] + \left(1 + \frac{4}{\delta_\ell}\right) \mathbb{E}[\|\nabla_\ell f(\boldsymbol{x}^{(t)}) - \nabla_\ell f(\boldsymbol{x}^{(k\tau)})\|_F^2] + \frac{11\sigma_\ell^2}{\delta_\ell \mathcal{B}}, \tag{124}$$

where the first inequality uses unbiasedness of $\boldsymbol{G}_\ell^{(k\tau)}$, the second inequality uses Assumption 3, the third inequality uses Young's inequality.

Applying (124) to (123) and applying Cauchy's inequality yields

$$\mathbb{E}[\|(\boldsymbol{E} - \boldsymbol{S}_\ell^{(k\tau)}) \odot \nabla_\ell f(\boldsymbol{x}^{(t)})\|_F^2]$$

$$\leq \left(1 - \frac{\delta_\ell}{2}\right) \mathbb{E}[\|\nabla_\ell f(\boldsymbol{x}^{(t)})\|_F^2] + \frac{11\sigma_\ell^2}{\delta_\ell \mathcal{B}} + \frac{15r}{\delta_\ell} \sum_{i=1}^{r} \mathbb{E}[\|\nabla_\ell f(\boldsymbol{x}^{(k\tau+i)}) - \nabla_\ell f(\boldsymbol{x}^{(k\tau+i-1)})\|_F^2]. \tag{125}$$

For the third term, we have

$$\mathbb{E}[\|\boldsymbol{S}_\ell^{(k\tau)} \odot (\boldsymbol{G}_\ell^{(t)} - \nabla_\ell f(\boldsymbol{x}^{(t)}))\|_F^2] \leq \mathbb{E}[\|\boldsymbol{G}_\ell^{(t)} - \nabla_\ell f(\boldsymbol{x}^{(t)})\|_F^2] \leq \sigma_\ell^2, \tag{126}$$

where the second inequality uses Assumption 3.

Applying (119)(125)(126) to (122) yields (114). $\qquad\square$

Based on Lemma 18, we can prove the convergence properties of large-batch GaSare similarly as the proofs of Lemma 9, Theorem 12 and Corollary 13. Below we directly present the final convergence results.

**Theorem 22** (Convergence of large-batch GaSare)**:** Under Assumptions 1-3, if hyperparameters

$$0 < \beta_1 \leq 1, \quad \tau \geq \frac{128}{3\beta_1 \underline{\delta}}, \quad 0 < \eta \leq \min\left\{\frac{1}{4L}, \sqrt{\frac{3\delta\beta_1^2}{80L^2}}, \sqrt{\frac{\delta}{40\tau^2 L^2}}, \sqrt{\frac{3\beta_1}{32\tau L^2}}\right\},$$

GaSare using large-batch stochastic gradients and MSGD (Alg. 6) converges as

$$\frac{1}{K\tau} \sum_{t=0}^{K\tau-1} \mathbb{E}\|\nabla f(\boldsymbol{x}^{(t)})\|_2^2 \leq \frac{16\Delta}{\underline{\delta}\eta K\tau} + \left(\frac{160}{3\beta_1 \underline{\delta}\tau\mathcal{B}} + \frac{352}{3\underline{\delta}^2 \mathcal{B}} + \frac{32\beta_1}{3\underline{\delta}}\right)\sigma^2$$

for any $K \geq 1$, where $\Delta = f(\boldsymbol{x}^{(0)}) - \inf_{\boldsymbol{x}} f(\boldsymbol{x})$. If $T \geq 2 + 256/(3\underline{\delta}) + (256\sigma)^2/(9\sqrt{\underline{\delta}}L\Delta)$ and we further choose

$$\beta_1 = \left(1 + \sqrt{\frac{\underline{\delta}^{3/2}\sigma^2 T}{L\Delta}}\right)^{-1},$$

$$\tau = \left\lceil \frac{128}{3\underline{\delta}\beta_1} \right\rceil,$$

$$\eta = \left( 4L + \sqrt{\frac{80L^2}{3\underline{\delta}\beta_1^2}} + \sqrt{\frac{40\tau^2 L^2}{\underline{\delta}}} + \sqrt{\frac{32\tau L^2}{3\beta_1}} \right)^{-1},$$

$$\mathcal{B} = \left\lceil \frac{1}{\underline{\delta}\beta_1} \right\rceil,$$

GaSare using large-batch stochastic gradients and MSGD (Alg. 6) converges as

$$\frac{1}{T}\sum_{t=0}^{T-1} \mathbb{E}[\|\nabla f(\boldsymbol{x}^{(t)})\|_2^2] = \mathcal{O}\left( \frac{L\Delta}{\underline{\delta}^{5/2}T} + \sqrt{\frac{L\Delta\sigma^2}{\underline{\delta}^{7/2}T}} \right).$$

Consequently, the computation complexity to reach an $\varepsilon$-accurate solution $\boldsymbol{x}$ such that $\|\nabla f(\boldsymbol{x})\|_2^2 \leq \varepsilon$ is given by $\mathcal{O}\left( \frac{L\Delta\sigma^2}{\underline{\delta}^{7/2}\varepsilon^2} + \frac{L\Delta}{\underline{\delta}^{5/2}\varepsilon} + \frac{\sigma^2}{\underline{\delta}^{1/2}L\Delta} + \frac{1}{\underline{\delta}} \right)$.

### D.6. Convergence of GoSare

In this subsection, we present the convergence properties of GoSare with small-batch stochastic gradients. The results and proofs are similar to those of GoLore in Appendix B.5.

**Lemma 19** (Momentum contraction): Under Assumption 3, in GoSare using MSGD (Alg. 6), if $0 < \beta_1 \leq 1$, term $\tilde{\boldsymbol{M}}_\ell^{(t)}$ has the following contraction properties:

- When $t = 0$, it holds that

$$\mathbb{E}[\|\tilde{\boldsymbol{M}}_\ell^{(0)} - \nabla_\ell f(\boldsymbol{X}^{(0)})\|_F^2] \leq (\tau-1)(1-\delta_\ell\beta_1)\sum_{r=0}^{\tau-2}\mathbb{E}[\|\nabla_\ell f(\boldsymbol{x}^{(r+1)}) - \nabla_\ell f(\boldsymbol{x}^{(r)})\|_F^2]$$
$$+ \frac{2(1-\delta_\ell\beta_1)}{\tau}\sum_{r=0}^{\tau-1}\mathbb{E}[\|\nabla_\ell f(\boldsymbol{x}^{(r)})\|_F^2] + \delta_\ell\beta_1^2\sigma_\ell^2; \tag{127}$$

- When $t = k\tau$, $k \in \mathbb{N}^*$, it holds that

$$\mathbb{E}[\|\tilde{\boldsymbol{M}}_\ell^{(t)} - \nabla_\ell f(\boldsymbol{x}^{(t)})\|_F^2] - \delta_\ell\left(1-\left(1-\frac{\delta_\ell}{4}\right)\beta_1\right)\mathbb{E}[\|\tilde{\boldsymbol{M}}_\ell^{(t-1)} - \nabla_\ell f(\boldsymbol{x}^{(t-1)})\|_F^2]$$

$$\leq \frac{2(1-\delta_\ell)}{\tau}\sum_{r=0}^{\tau-1}\mathbb{E}[\|\nabla_l f(\boldsymbol{x}^{(k\tau+r)})\|_F^2] + \frac{5(1-\beta_1)}{\beta_1}\mathbb{E}[\|\nabla_\ell f(\boldsymbol{x}^{(t)}) - \nabla_\ell f(\boldsymbol{x}^{(t-1)})\|_F^2]$$

$$+ (\tau-1)(1-\delta_\ell)\sum_{r=0}^{\tau-2}\mathbb{E}[\|\nabla_\ell f(\boldsymbol{x}^{(k\tau+r+1)}) - \nabla_\ell f(\boldsymbol{x}^{(k\tau+r)})\|_F^2] + \delta_\ell\beta_1^2\sigma_\ell^2; \tag{128}$$

- When $t = k\tau + r$, $k \in \mathbb{N}$, $1 \leq r < \tau$, it holds that

$$\mathbb{E}[\|\tilde{\boldsymbol{M}}_\ell^{(t)} - \nabla_\ell f(\boldsymbol{x}^{(t)})\|_F^2] - \left(1-\left(1-\frac{\delta_\ell}{4}\right)\beta_1\right)\mathbb{E}[\|\tilde{\boldsymbol{M}}_\ell^{(t-1)} - \nabla_\ell f(\boldsymbol{x}^{(t-1)})\|_F^2]$$

$$\leq \left(1-\frac{\delta_\ell}{2}\right)\beta_1\mathbb{E}[\|\nabla_\ell f(\boldsymbol{x}^{(t)})\|_F^2] + \frac{5(1-\beta_1)}{\delta_\ell\beta_1}\mathbb{E}[\|\nabla_\ell f(\boldsymbol{x}^{(t)}) - \nabla_\ell f(\boldsymbol{x}^{(t-1)})\|_F^2]$$

$$+ \frac{10r\beta_1}{\delta_\ell}\sum_{i=1}^{r}\mathbb{E}[\|\nabla_\ell f(\boldsymbol{x}^{(k\tau+i)}) - \nabla_\ell f(\boldsymbol{x}^{(k\tau+i-1)})\|_F^2] + \beta_1^2\sigma_\ell^2. \tag{129}$$

*Proof.* For convenience we use $\boldsymbol{E}$ to denote $\boldsymbol{E}_{m_\ell,n_\ell}$. When $t = 0$, we have

$$\mathbb{E}[\|\tilde{\boldsymbol{M}}_\ell^{(0)} - \nabla_\ell f(\boldsymbol{x}^{(0)})\|_F^2]$$

$$
\begin{aligned}
&= \mathbb{E}[\|\beta_1 \boldsymbol{S}_\ell^{(0)} \odot \boldsymbol{G}_\ell^{(0)} - \nabla_\ell f(\boldsymbol{x}^{(0)})\|_F^2] \\
&= \mathbb{E}[\|(\beta_1 \boldsymbol{S}_\ell^{(0)} - \boldsymbol{E}) \odot \nabla_\ell f(\boldsymbol{x}^{(0)})\|_F^2] + \beta_1^2 \mathbb{E}[\|\boldsymbol{S}_\ell^{(0)} \odot (\boldsymbol{G}_\ell^{(0)} - \nabla_\ell f(\boldsymbol{x}^{(0)}))\|_F^2],
\end{aligned}
\tag{130}
$$

where the second equality uses unbiasedness of $\boldsymbol{G}_\ell^{(0)}$. By Lemma 5 we have

$$
\begin{aligned}
&\mathbb{E}[\|(\beta_1 \boldsymbol{S}_\ell^{(0)} - \boldsymbol{E}) \odot \nabla_\ell f(\boldsymbol{x}^{(0)})\|_F^2 \\
&= \sum_{1 \le i \le m_\ell, 1 \le j \le n_\ell} \mathbb{E}[(\beta_1 [S_\ell^{(0)}]_{i,j} - 1)^2][\nabla_\ell f(\boldsymbol{x}^{(0)})]_{i,j}^2 \\
&= \sum_{1 \le i \le m_\ell, 1 \le j \le n_\ell} (1 - 2\beta_1 \delta_\ell + \beta_1^2 \delta_\ell)[\nabla_\ell f(\boldsymbol{x}^{(0)})]_{i,j}^2 \\
&\le (1 - \delta_\ell \beta_1)\|\nabla_\ell f(\boldsymbol{x}^{(0)})\|_F^2.
\end{aligned}
\tag{131}
$$

Similarly, by Lemma 5 we have

$$
\begin{aligned}
&\mathbb{E}[\|\boldsymbol{S}_\ell^{(0)} \odot (\boldsymbol{G}_\ell^{(0)} - \nabla_\ell f(\boldsymbol{x}^{(0)}))\|_F^2 \\
&= \sum_{1 \le i \le m_\ell, 1 \le j \le n_\ell} \mathbb{E}[[S_\ell^{(0)}]_{i,j}^2][\boldsymbol{G}_\ell^{(0)} - \nabla_\ell f(\boldsymbol{x}^{(0)})]_{i,j}^2 \\
&= \delta_\ell \mathbb{E}[\|\boldsymbol{G}_\ell^{(0)} - \nabla_\ell f(\boldsymbol{x}^{(0)})\|_F^2] \\
&\le \delta_\ell \sigma_\ell^2,
\end{aligned}
\tag{132}
$$

where the inequality uses Assumption 3. Applying (131)(132) and Lemma 2 to (130) yields (127).

When $t = k\tau$, $k \in \mathbb{N}^*$, we have

$$
\begin{aligned}
&\mathbb{E}[\|\tilde{\boldsymbol{M}}_\ell^{(t)} - \nabla_\ell f(\boldsymbol{x}^{(t)})\|_F^2] \\
&= \mathbb{E}[\|\boldsymbol{S}_\ell^{(t)} \odot [(1 - \beta_1)\tilde{\boldsymbol{M}}_\ell^{(t-1)} + \beta_1 \boldsymbol{G}_\ell^{(t)} - \nabla_\ell f(\boldsymbol{x}^{(t)})] - (\boldsymbol{E} - \boldsymbol{S}_\ell^{(t)}) \odot \nabla_\ell f(\boldsymbol{x}^{(t)})\|_F^2] \\
&= \delta_\ell \mathbb{E}[\|(1 - \beta_1)\tilde{\boldsymbol{M}}_\ell^{(t-1)} + \beta_1 \boldsymbol{G}_\ell^{(t)} - \nabla_\ell f(\boldsymbol{x}^{(t)})\|_F^2] + (1 - \delta_\ell)\mathbb{E}[\|\nabla_\ell f(\boldsymbol{x}^{(t)})\|_F^2],
\end{aligned}
\tag{133}
$$

where the second equality uses Lemma 16. For the first term, we have

$$
\begin{aligned}
&\mathbb{E}[\|(1 - \beta_1)\tilde{\boldsymbol{M}}_\ell^{(t-1)} + \beta_1 \boldsymbol{G}_\ell^{(t)} - \nabla_\ell f(\boldsymbol{x}^{(t)})\|_F^2] \\
&= \mathbb{E}[\|(1 - \beta_1)(\tilde{\boldsymbol{M}}_\ell^{(t-1)} - \nabla_\ell f(\boldsymbol{x}^{(t)})) + \beta_1(\boldsymbol{G}_\ell^{(t)} - \nabla_\ell f(\boldsymbol{x}^{(t)}))\|_F^2] \\
&\le \mathbb{E}[\|(1 - \beta_1)(\tilde{\boldsymbol{M}}_\ell^{(t-1)} - \nabla_\ell f(\boldsymbol{x}^{(t)}))\|_F^2] + \beta_1^2 \mathbb{E}[\|\boldsymbol{G}_\ell^{(t)} - \nabla_\ell f(\boldsymbol{x}^{(t)})\|_F^2] \\
&\le (1 - \beta_1)\mathbb{E}[\|\tilde{\boldsymbol{M}}_\ell^{(t-1)} - \nabla_\ell f(\boldsymbol{x}^{(t)})\|_F^2] + \beta_1^2 \sigma_\ell^2,
\end{aligned}
\tag{134}
$$

where both inequalities use Assumption 3. By Young's inequality, we have

$$
\begin{aligned}
&\mathbb{E}[\|\tilde{\boldsymbol{M}}_\ell^{(t-1)} - \nabla_\ell f(\boldsymbol{x}^{(t)})\|_F^2] \\
&= \mathbb{E}[\|(\tilde{\boldsymbol{M}}_\ell^{(t-1)} - \nabla_\ell f(\boldsymbol{x}^{(t-1)})) - (\nabla_\ell f(\boldsymbol{x}^{(t)}) - \nabla_\ell f(\boldsymbol{x}^{(t-1)}))\|_F^2] \\
&\le \left(1 + \frac{\delta_\ell \beta_1}{4}\right) \mathbb{E}[\|\tilde{\boldsymbol{M}}_\ell^{(t-1)} - \nabla_\ell f(\boldsymbol{x}^{(t-1)})\|_F^2] + \left(1 + \frac{4}{\delta_\ell \beta_1}\right) \mathbb{E}[\|\nabla_\ell f(\boldsymbol{x}^{(t)}) - \nabla_\ell f(\boldsymbol{x}^{(t-1)})\|_F^2].
\end{aligned}
\tag{135}
$$

Applying (134)(135) and Lemma 2 to (133) yields (128).

When $t = k\tau + r$, $k \in \mathbb{N}$, $1 \le r < \tau$, we have

$$
\begin{aligned}
&\mathbb{E}[\|\tilde{\boldsymbol{M}}_\ell^{(t)} - \nabla_\ell f(\boldsymbol{x}^{(t)})\|_F^2] \\
&= \mathbb{E}[\|(1 - \beta_1)(\tilde{\boldsymbol{M}}_\ell^{(t-1)} - \nabla_\ell f(\boldsymbol{x}^{(t)})) + \beta_1(\boldsymbol{S}_\ell^{(t)} \odot \boldsymbol{G}_\ell^{(t)} - \nabla_\ell f(\boldsymbol{x}^{(t)}))\|_F^2] \\
&= \mathbb{E}[\|(1 - \beta_1)(\tilde{\boldsymbol{M}}_\ell^{(t-1)} - \nabla_\ell f(\boldsymbol{x}^{(t)})) + \beta_1(\boldsymbol{S}_\ell^{(t)} - \boldsymbol{E}) \odot \nabla_\ell f(\boldsymbol{x}^{(t)})\|_F^2]
\end{aligned}
$$

$$+ \beta_1^2 \mathbb{E}[\boldsymbol{S}_\ell^{(t)} \odot (\boldsymbol{G}_\ell^{(t)} - \nabla_\ell f(\boldsymbol{x}^{(t)}))\|_F^2]$$

$$\leq (1 - \beta_1) \mathbb{E}[\|\tilde{\boldsymbol{M}}_\ell^{(t-1)} - \nabla_\ell f(\boldsymbol{x}^{(t)})\|_F^2] + \beta_1 \mathbb{E}[\|(\boldsymbol{E} - \boldsymbol{S}_\ell^{(t)}) \odot \nabla_\ell f(\boldsymbol{x}^{(t)})\|_F^2$$

$$+ \beta_1^2 \mathbb{E}[\boldsymbol{S}_\ell^{(t)} \odot (\boldsymbol{G}_\ell^{(t)} - \nabla_\ell f(\boldsymbol{x}^{(t)}))\|_F^2], \tag{136}$$

where the second equality uses the unbiasedness of $\boldsymbol{G}_\ell^{(t)}$ and the independence implied by $\boldsymbol{S}_\ell^{(t)} = \boldsymbol{S}_\ell^{(t-1)}$, the inequality uses Jensen's inequality. The first term is similarly bounded as (135). For the second term, we have

$$\mathbb{E}[\|(\boldsymbol{E} - \boldsymbol{S}_\ell^{(k\tau)}) \odot \nabla_\ell f(\boldsymbol{x}^{(t)})\|_F^2]$$

$$\leq \left(1 + \frac{\delta_\ell}{4}\right) \mathbb{E}[\|(\boldsymbol{E} - \boldsymbol{S}_\ell^{(k\tau)}) \odot \nabla_\ell f(\boldsymbol{x}^{(k\tau)})\|_F^2]$$

$$+ \left(1 + \frac{4}{\delta_\ell}\right) \mathbb{E}[\|(\boldsymbol{E} - \boldsymbol{S}_\ell^{(k\tau)}) \odot (\nabla_\ell f(\boldsymbol{x}^{(t)}) - \nabla_\ell f(\boldsymbol{x}^{(k\tau)}))\|_F^2]$$

$$\leq \left(1 - \frac{3\delta_\ell}{4}\right) \mathbb{E}[\|\nabla_\ell f(\boldsymbol{x}^{(k\tau)})\|_F^2] + \left(1 + \frac{4}{\delta_\ell}\right) \mathbb{E}[\|\nabla_\ell f(\boldsymbol{x}^{(t)}) - \nabla_\ell f(\boldsymbol{x}^{(k\tau)})\|_F^2], \tag{137}$$

where the first inequality uses Young's inequality, the second inequality uses Lemma 16. By Young's inequality, we have

$$\mathbb{E}[\|\nabla_\ell f(\boldsymbol{x}^{(k\tau)})\|_F^2] \leq \left(1 + \frac{\delta_\ell}{4}\right) \mathbb{E}[\|\nabla_\ell f(\boldsymbol{x}^{(t)})\|_F^2] + \left(1 + \frac{4}{\delta_\ell}\right) \mathbb{E}[\|\nabla_\ell f(\boldsymbol{x}^{(t)}) - \nabla_\ell f(\boldsymbol{x}^{(k\tau)})\|_F^2]. \tag{138}$$

Applying (138) to (137) and applying Cauchy's inequality yields

$$\mathbb{E}[\|(\boldsymbol{E} - \boldsymbol{S}_\ell^{(k\tau)}) \odot \nabla_\ell f(\boldsymbol{x}^{(t)})\|_F^2]$$

$$\leq \left(1 - \frac{\delta_\ell}{2}\right) \mathbb{E}[\|\nabla_\ell f(\boldsymbol{x}^{(t)})\|_F^2] + \frac{10r}{\delta_\ell} \sum_{i=1}^r \mathbb{E}[\|\nabla_\ell f(\boldsymbol{x}^{(k\tau+i)}) - \nabla_\ell f(\boldsymbol{x}^{(k\tau+i-1)})\|_F^2]. \tag{139}$$

For the third term, we have

$$\mathbb{E}[\|\boldsymbol{S}_\ell^{(k\tau)} \odot (\boldsymbol{G}_\ell^{(t)} - \nabla_\ell f(\boldsymbol{x}^{(t)}))\|_F^2] \leq \mathbb{E}[\|\boldsymbol{G}_\ell^{(t)} - \nabla_\ell f(\boldsymbol{x}^{(t)})\|_F^2] \leq \sigma_\ell^2, \tag{140}$$

where the second inequality uses Assumption 3.

Applying (135)(139)(140) to (136) yields (129). $\qquad \square$

Based on Lemma 19, we can prove the convergence properties of GoSare similarly as the proofs of Lemma 11, Theorem 14 and Corollary 15. Below we directly present the final convergence results.

**Theorem 23** (Convergence of GoSare): Under Assumptions 1-3, if hyperparameters

$$0 < \beta_1 \leq 1, \quad \tau \geq \frac{64}{3\beta_1 \underline{\delta}}, \quad 0 < \eta \leq \min\left\{\frac{1}{4L}, \sqrt{\frac{3\underline{\delta}\beta_1^2}{80L^2}}, \sqrt{\frac{3\underline{\delta}}{80\tau^2 L^2}}, \sqrt{\frac{3\beta_1}{16\tau L^2}}\right\},$$

GoSare using small-batch stochastic gradients and MSGD (Alg. 6) converges as

$$\frac{1}{K\tau} \sum_{t=0}^{K\tau-1} \mathbb{E}\|\nabla f(\boldsymbol{x}^{(t)})\|_2^2 \leq \frac{16\Delta}{\underline{\delta}\eta K\tau} + \frac{32\beta_1 \sigma^2}{3\underline{\delta}}$$

for any $K \geq 1$, where $\Delta = f(\boldsymbol{x}^{(0)}) - \inf_{\boldsymbol{x}} f(\boldsymbol{x})$. If $T \geq 2 + 128/(3\underline{\delta}) + (128\sigma)^2/(9\sqrt{\underline{\delta}}L\Delta)$ and we further choose

$$\beta_1 = \left(1 + \sqrt{\frac{\underline{\delta}^{3/2}\sigma^2 T}{L\Delta}}\right)^{-1},$$

$$\tau = \left\lceil \frac{64}{3\underline{\delta}\beta_1} \right\rceil,$$

$$\eta = \left(4L + \sqrt{\frac{80L^2}{3\underline{\delta}\beta_1^2}} + \sqrt{\frac{80\tau^2 L^2}{3\underline{\delta}}} + \sqrt{\frac{16\tau L^2}{3\beta_1}}\right)^{-1},$$

GoSare using small-batch stochastic gradients and MSGD (Alg. 6) converges as

$$\frac{1}{T}\sum_{t=0}^{T-1}\mathbb{E}[\|\nabla f(\boldsymbol{x}^{(t)})\|_2^2] = \mathcal{O}\left(\frac{L\Delta}{\underline{\delta}^{5/2}T} + \sqrt{\frac{L\Delta\sigma^2}{\underline{\delta}^{7/2}T}}\right).$$

Consequently, the computation complexity to reach an $\varepsilon$-accurate solution $\boldsymbol{x}$ such that $\|\nabla f(\boldsymbol{x})\|_2^2 \le \varepsilon$ is given by $\mathcal{O}\left(\frac{L\Delta\sigma^2}{\underline{\delta}^{7/2}\varepsilon^2} + \frac{L\Delta}{\underline{\delta}^{5/2}\varepsilon} + \frac{\sigma^2}{\underline{\delta}^{1/2}L\Delta} + \frac{1}{\underline{\delta}}\right)$.

## E. Experimental specifications

In this section, we elaborate on the missing details concerned with the experiments we present in Sec. 6.

**GaLore's non-convergence.** We compared Galore, large-batch GaLore, GoLore and full-parameter training on the constructed quadratic problem defined in (1). We used a batch size of 128 for large-batch GaLore and a batch size of 1 for the others.

**Pre-training tasks on C4 dataset.** We pre-trained LLaMA-60M on C4 dataset for 30,000 iterations on 4 NVIDIA A100 40G GPUs. We use batch size 128, learning rate 1.0e-3, rank 128, scaling factor $\alpha = 1$, subspace changing frequency $\tau = 200$, and a max sequence length of 256.

**Fine-tuning tasks on WinoGrande dataset.** We fine-tune pre-trained LLaMA2-7B model on the WinoGrande dataset for 30 epochs on 4 NVIDIA A100 80G GPUs. We use batch size 1, rank 1024, subspaces changing frequency $\tau = 500$ and a max sequence length of 2048. The learning rate and scaling factor are set as 1.0e-4 and $\alpha = 4$ for GaLore/GoLore, thus corresponds to a learning rate of 4.0e-4 in full-parameter fine-tuning. The test accuracy is presented in Table 4, where GoLore performs similarly to GaLore due to overfitting.

**Fine-tuning tasks on BoolQ dataset.** We fine-tune pre-trained LLaMA2-7B model on the BoolQ dataset on 4 NVIDIA A100 80G GPUs. We use batch size 1, rank 1024, subspaces changing frequency $\tau = 500$ and a max sequence length of 2048. We use MSGD as the subspace optimizer, where the learning rate and scaling factor are set as 1.0e-4 and $\alpha = 4$ for GaLore/GoLore, corresponding to a learning rate of 4.0e-4 in full-parameter fine-tuning. Table 4 presents the test accuracy of different algorithms, where GoLore outperforms GaLore. We further fine-tune pre-trained OPT-13B for 1 epoch using the same experimental setup, whose results are shown in Table 2.

*Table 4.* Evaluating GaLore/GoLore for fine-tuning on WinoGrande and BoolQ using pre-trained LLaMA2-7B.

| Algorithm | BoolQ (1 epoch) | BoolQ (3 epochs) | WinoGrande (80 epochs) |
|---|---|---|---|
| Full Params. | 86.48 | 87.43 | 69.85 |
| GaLore | 84.89 | 86.79 | **68.51** |
| GoLore@20% | **85.81** | **86.88** | **68.51** |

**Fine-tuning tasks on GLUE benchmark.** We fine-tune pre-trained RoBERTa-Base model on the GLUE benchmark for 30 epochs on a single GeForce RTX 4090. Training details including batch size, learning rate, rank, scaling factor $\alpha$ and max sequence length are illustrated in Table 5.

## F. Ablation studies

In this section, we conduct several ablation studies to provide a deeper understanding of the proposed GoLore algorithm.

**Ablation on the switching point.** By *switching point*, we mean the ratio of GaLore to GoLore during training; for instance, GoLore@50% indicates a switching point of 0.5. In our experiments, we choose an earlier switching point if we expect the algorithm to converge more quickly to the solution's neighborhood and a later one if we anticipate slower convergence. To provide a broader guideline, we offer the following insights:

*Table 5.* Hyperparameters used in fine-tuning pre-trained RoBERTa-Base model on the GLUE benchmark.

| Hyperparameter | CoLA | STS-B | MRPC | RTE | SST2 | MNLI | QNLI | QQP |
|---|---|---|---|---|---|---|---|---|
| batch size | 32 | 16 | 16 | 16 | 16 | 16 | 16 | 16 |
| Learning Rate | 2.5e-5 | 2.0e-5 | 3.5e-5 | 7.0e-6 | 1.0e-5 | 1.0e-5 | 1.0e-5 | 1.0e-5 |
| Rank | 4 | 4 | 4 | 4 | 4 | 4 | 4 | 4 |
| GaLore's $\alpha$ | 4 | 4 | 4 | 4 | 4 | 4 | 4 | 4 |
| FLORA'S $\alpha$ | 4 | 4 | 4 | 4 | 4 | 4 | 4 | 4 |
| GoLore's $\alpha$ | 4 | 4 | 4 | 4 | 4 | 4 | 4 | 4 |
| Frequency $\tau$ | 500 | 500 | 500 | 500 | 500 | 500 | 500 | 500 |
| Max Seq. Len. | 512 | 512 | 512 | 512 | 512 | 512 | 512 | 512 |

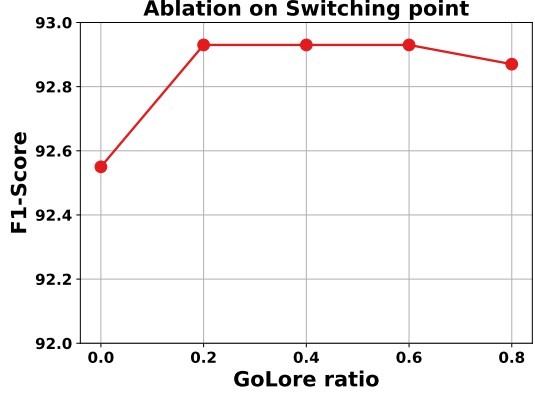

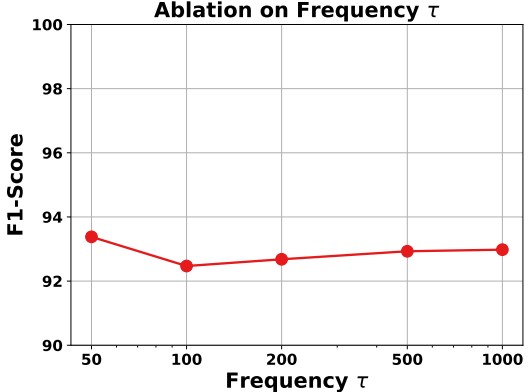

*Figure 10.* Ablation results on the switching point. We report the F1-scores of fine-tuning on the MRPC task in the GLUE benchmark with different GoLore ratios. A GoLore ratio of 0 represents GaLore.

*Figure 11.* Ablation results on the subspace update frequency. We report the F1-scores of fine-tuning on the MRPC task in GLUE benchmark with GoLore@20% with different subspace update frequency $\tau$'s.

- If access to the true gradient is available, a reasonable switching point would be when gradient noise begins to dominate the true gradient.

- If only stochastic gradients are available, and assuming the gradient noise has a roughly constant scale, this dominance can be estimated by monitoring whether the norm of the stochastic gradients falls below a certain threshold. This threshold serves as a hyperparameter that depends on both the training task and the batch size used in the algorithm.

- The optimal switching point is task-dependent. Empirically, it is recommended to switch when the rate of decrease in the loss curve starts to slow down.

  Fig. 10 shows model performance across different switching points. Except for pure GaLore (switching point 0), GoLore achieves comparable performance at all other points.

**Ablation on subspace update frequency.** Consistent with Zhao et al. (2024), which reports stable performance for GaLore across $\tau \in [50, 1000]$, Fig. 11 shows that GoLore@20% exhibits similar robustness to different $\tau$ values in the same range.

**Ablation on subspace sampling strategy.** An important question is whether alternative non-greedy sampling strategies can outperform GoLore. To explore this, we evaluate an importance sampling-based method. Given a stochastic gradient matrix $\boldsymbol{G} \in \mathbb{R}^{m \times n}$ ($m \leq n$), we first perform SVD to obtain $\boldsymbol{G} = \boldsymbol{U}\boldsymbol{\Sigma}\boldsymbol{V}^\top$, following the procedure used in GaLore. However, instead of selecting the top-$r$ columns of $\boldsymbol{U}$ as in GaLore, we sample $r$ columns with probabilities proportional to the corresponding singular values $\sigma_i$. As shown in Fig. 12, this sampling-based strategy achieves performance similar to GaLore, but is consistently outperformed by GoLore. This may be attributed to the fact that importance sampling does not yield unbiased projection matrices and remains susceptible to the bias introduced by stochastic gradient noise.

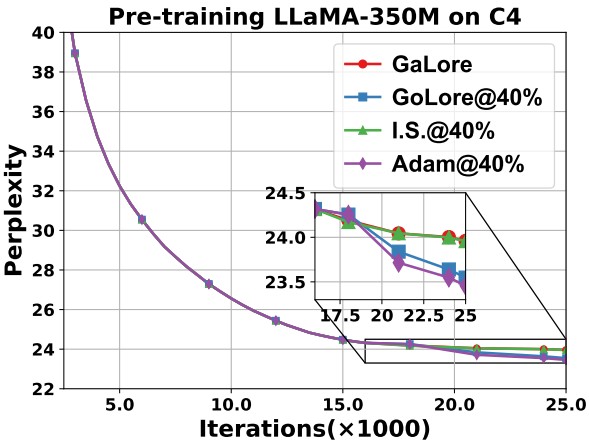

*Figure 12.* Additional results of pre-training LLaMA-350M on C4 dataset. "I.S." represents the alternative importance sampling strategy.

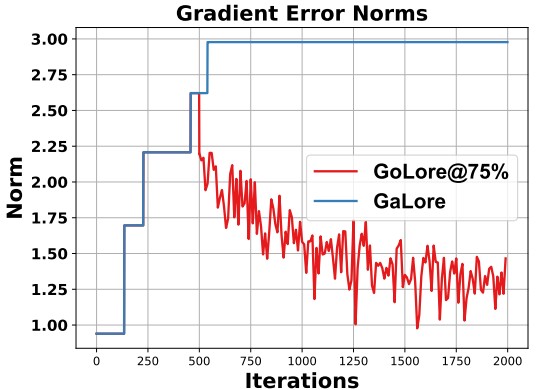

*Figure 13.* Illustration of gradient error norms when training the quadratic target problem with GaLore and GoLore@75% corresponding to Fig. 1 (right).

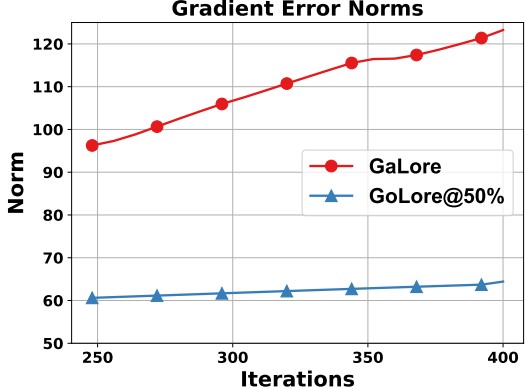

*Figure 14.* Illustration of gradient error norms when fine-tuning with GaLore and GoLore@50% on the MRPC task. Only 32 sequences are used for cheap true-gradient evaluation.

**Does GoLore's projection truly result in smaller compression errors?** To assess whether GoLore achieves lower compression error than GaLore due to its randomized projection strategy, we report the gradient approximation error $\|\nabla f(\boldsymbol{x}^{(t)}) - \hat{\nabla} f(\boldsymbol{x}^{(t)})\|_2$ in Fig.13 and Fig.14, where $\hat{\nabla} f(\boldsymbol{x}^{(t)})$ denotes the approximated stochastic gradient computed using GaLore or GoLore. As shown, GoLore consistently produces more accurate gradient approximations than GaLore across training steps.

---

**Algorithm 4** GaLore using large-batch stochastic gradients and MSGD with MP

---

**Input:** Initial point $\boldsymbol{x}^{(0)}$, data distribution $\mathcal{D}$, learning rate $\eta$, subspace changing frequency $\tau$, rank $\{r_\ell\}_{\ell=1}^{N_L}$, momentum parameter $\beta_1$, large batch size $\mathcal{B}$.
**Output:** $\{\boldsymbol{x}^{(t)}\}_{t=0}^{T}$.
  Initialize optimizer state $\{\boldsymbol{M}_\ell^{(-1)}\}_{\ell=1}^{N_L}$ to zero;
  **for** $t = 0, 1, \cdots, T-1$ **do**
    **if** $t \equiv 0 \pmod{\tau}$ **then**
      Sample $\{\xi^{(t,b)}\}_{b=1}^{\mathcal{B}} \overset{\text{i.i.d.}}{\sim} \mathcal{D}$;
    **else**
      Sample $\xi^{(t)} \sim \mathcal{D}$;
    **end if**
    **for** $\ell = 1, 2, \cdots, N_L$ **do**
      **if** $t \equiv 0 \pmod{\tau}$ **then**
        $\boldsymbol{G}_\ell^{(t)} = \frac{1}{\mathcal{B}} \sum_{b=1}^{\mathcal{B}} \nabla_\ell F(\boldsymbol{x}^{(t)}; \xi^{(t,b)})$;
        $\boldsymbol{U}, \boldsymbol{\Sigma}, \boldsymbol{V} \leftarrow \text{SVD}(\boldsymbol{G}_\ell^{(t)})$;
        **if** $m_\ell \leq n_\ell$ **then**
          $\boldsymbol{P}_\ell^{(t)} \leftarrow \boldsymbol{U}[:, :r_\ell]$;
          $\boldsymbol{M}_\ell^{(t)} \leftarrow (1-\beta_1)(\boldsymbol{P}_\ell^{(t)})^\top \boldsymbol{P}_\ell^{(t-1)} \boldsymbol{M}_\ell^{(t-1)} + \beta_1 (\boldsymbol{P}_\ell^{(t)})^\top \boldsymbol{G}_\ell^{(t)}$;
          $\boldsymbol{X}_\ell^{(t+1)} \leftarrow \boldsymbol{X}_\ell^{(t)} - \eta \boldsymbol{P}_\ell^{(t)} \boldsymbol{M}_\ell^{(t)}$;
        **else**
          $\boldsymbol{Q}_\ell^{(t)} \leftarrow \boldsymbol{V}[:, :r_\ell]$;
          $\boldsymbol{M}_\ell^{(t)} \leftarrow (1-\beta_1)\boldsymbol{M}_\ell^{(t-1)}(\boldsymbol{Q}_\ell^{(t-1)})^\top \boldsymbol{Q}_\ell^{(t)} + \beta_1 \boldsymbol{G}_\ell^{(t)} \boldsymbol{Q}_\ell^{(t)}$;
          $\boldsymbol{X}_\ell^{(t+1)} \leftarrow \boldsymbol{X}_\ell^{(t)} - \eta \boldsymbol{M}_\ell^{(t)}(\boldsymbol{Q}_\ell^{(t)})^\top$;
        **end if**
      **else**
        $\boldsymbol{G}_\ell^{(t)} = \nabla_\ell F(\boldsymbol{x}^{(t)}; \xi^{(t)})$;
        **if** $m_\ell \leq n_\ell$ **then**
          $\boldsymbol{P}_\ell^{(t)} \leftarrow \boldsymbol{P}_\ell^{(t-1)}$;
          $\boldsymbol{M}_\ell^{(t)} \leftarrow (1-\beta_1)\boldsymbol{M}_\ell^{(t-1)} + \beta_1 (\boldsymbol{P}_\ell^{(t)})^\top \boldsymbol{G}_\ell^{(t)}$;
          $\boldsymbol{X}_\ell^{(t+1)} \leftarrow \boldsymbol{X}_\ell^{(t)} - \eta \boldsymbol{P}_\ell^{(t)} \boldsymbol{M}_\ell^{(t)}$;
        **else**
          $\boldsymbol{Q}_\ell^{(t)} \leftarrow \boldsymbol{Q}_\ell^{(t-1)}$;
          $\boldsymbol{M}_\ell^{(t)} \leftarrow (1-\beta_1)\boldsymbol{M}_\ell^{(t-1)} + \beta_1 \boldsymbol{G}_\ell^{(t)} \boldsymbol{Q}_\ell^{(t)}$;
          $\boldsymbol{X}_\ell^{(t+1)} \leftarrow \boldsymbol{X}_\ell^{(t)} - \eta \boldsymbol{M}_\ell^{(t)}(\boldsymbol{Q}_\ell^{(t)})^\top$;
        **end if**
      **end if**
    **end for**
  **end for**

---

---

**Algorithm 5** GoLore using small-batch stochastic gradients and MSGD with MP

---

**Input:** Initial point $\boldsymbol{x}^{(0)}$, data distribution $\mathcal{D}$, learning rate $\eta$, subspace changing frequency $\tau$, rank $\{r_\ell\}_{\ell=1}^{N_L}$, momentum parameter $\beta_1$.

**Output:** $\{\boldsymbol{x}^{(t)}\}_{t=0}^T$.

  Initialize optimizer state $\{\boldsymbol{M}_\ell^{(-1)}\}_{\ell=1}^{N_L}$ to zero;

  **for** $t = 0, 1, \cdots, T-1$ **do**

    Sample $\xi^{(t)} \sim \mathcal{D}$;

    $\boldsymbol{G}_\ell^{(t)} = \nabla_\ell F(\boldsymbol{x}^{(t)}; \xi^{(t)})$;

    **for** $\ell = 1, 2, \cdots, N_L$ **do**

      **if** $t \equiv 0 \ (\mathrm{mod}\ \tau)$ **then**

        **if** $m_\ell \leq n_\ell$ **then**

          Sample $\boldsymbol{P}_\ell^{(t)} \sim \mathcal{U}(\mathrm{St}_{m_\ell, r_\ell})$;

          $\boldsymbol{M}_\ell^{(t)} \leftarrow (1 - \beta_1)(\boldsymbol{P}_\ell^{(t)})^\top \boldsymbol{P}_\ell^{(t-1)} \boldsymbol{M}_\ell^{(t-1)} + \beta_1 (\boldsymbol{P}_\ell^{(t)})^\top \boldsymbol{G}_\ell^{(t)}$;

          $\boldsymbol{X}_\ell^{(t+1)} \leftarrow \boldsymbol{X}_\ell^{(t)} - \eta \boldsymbol{P}_\ell^{(t)} \boldsymbol{M}_\ell^{(t)}$;

        **else**

          Sample $\boldsymbol{Q}_\ell^{(t)} \sim \mathcal{U}(\mathrm{St}_{n_\ell, r_\ell})$;

          $\boldsymbol{M}_\ell^{(t)} \leftarrow (1 - \beta_1) \boldsymbol{M}_\ell^{(t-1)} (\boldsymbol{Q}_\ell^{(t-1)})^\top \boldsymbol{Q}_\ell^{(t)} + \beta_1 \boldsymbol{G}_\ell^{(t)} \boldsymbol{Q}_\ell^{(t)}$;

          $\boldsymbol{X}_\ell^{(t+1)} \leftarrow \boldsymbol{X}_\ell^{(t)} - \eta \boldsymbol{M}_\ell^{(t)} (\boldsymbol{Q}_\ell^{(t)})^\top$;

        **end if**

      **else**

        **if** $m_\ell \leq n_\ell$ **then**

          $\boldsymbol{P}_\ell^{(t)} \leftarrow \boldsymbol{P}_\ell^{(t-1)}$;

          $\boldsymbol{M}_\ell^{(t)} \leftarrow (1 - \beta_1) \boldsymbol{M}_\ell^{(t-1)} + \beta_1 (\boldsymbol{P}_\ell^{(t)})^\top \boldsymbol{G}_\ell^{(t)}$;

          $\boldsymbol{X}_\ell^{(t+1)} \leftarrow \boldsymbol{X}_\ell^{(t)} - \eta \boldsymbol{P}_\ell^{(t)} \boldsymbol{M}_\ell^{(t)}$;

        **else**

          $\boldsymbol{Q}_\ell^{(t)} \leftarrow \boldsymbol{Q}_\ell^{(t-1)}$;

          $\boldsymbol{M}_\ell^{(t)} \leftarrow (1 - \beta_1) \boldsymbol{M}_\ell^{(t-1)} + \beta_1 \boldsymbol{G}_\ell^{(t)} \boldsymbol{Q}_\ell^{(t)}$;

          $\boldsymbol{X}_\ell^{(t+1)} \leftarrow \boldsymbol{X}_\ell^{(t)} - \eta \boldsymbol{M}_\ell^{(t)} (\boldsymbol{Q}_\ell^{(t)})^\top$;

        **end if**

      **end if**

    **end for**

  **end for**

---

---

**Algorithm 6** GaSare / GoSare algorithms using stochastic / deterministic / large-batch gradients

---

**Input:** Initial point $\boldsymbol{x}^{(0)}$, data distribution $\mathcal{D}$, learning rate $\eta$, subspace changing frequency $\tau$, rank $\{r_\ell\}_{\ell=1}^{N_L}$, optimizer hyperparameters $\beta_1, \beta_2, \epsilon$, large batch size $\mathcal{B}$.

**Output:** $\{\boldsymbol{x}^{(t)}\}_{t=0}^{T}$.

  Initialize optimizer state $\{\boldsymbol{M}_\ell^{(-1)}\}_{\ell=1}^{N_L}$ and $\{\boldsymbol{V}_\ell^{(-1)}\}_{\ell=1}^{N_L}$ to zero;

  **for** $t = 0, 1, \cdots, T-1$ **do**

    **for** $\ell = 1, 2, \cdots, N_L$ **do**

      **if** $t \equiv 0 \pmod{\tau}$ **then**

        $\boldsymbol{G}_\ell^{(t)} \leftarrow \nabla_\ell F(\boldsymbol{x}^{(t)}; \xi^{(t)});$   (stochastic)

        $\boldsymbol{G}_\ell^{(t)} \leftarrow \nabla_\ell f(\boldsymbol{x}^{(t)});$   (deterministic)

        $\boldsymbol{G}_\ell^{(t)} \leftarrow \frac{1}{\mathcal{B}} \sum_{b=1}^{\mathcal{B}} \nabla_\ell F(\boldsymbol{x}^{(t)}; \xi^{(t,b)});$   (large-batch)

        $\boldsymbol{S}_\ell^{(t)} \leftarrow \mathrm{Top}_k(\boldsymbol{G}_\ell^{(t)});$   (GaSare)

        Sample $\boldsymbol{S}_\ell^{(t)} \sim \mathcal{U}(\mathrm{Sp}_{m_\ell, n_\ell}^{k_\ell});$   (GoSare)

      **else**

        $\boldsymbol{G}_\ell^{(t)} \leftarrow \nabla_\ell F(\boldsymbol{x}^{(t)}; \xi^{(t)});$   (stochastic)

        $\boldsymbol{G}_\ell^{(t)} \leftarrow \nabla_\ell f(\boldsymbol{x}^{(t)});$   (deterministic)

        $\boldsymbol{G}_\ell^{(t)} \leftarrow \nabla_\ell F(\boldsymbol{x}^{(t)}; \xi^{(t)});$   (large-batch)

        $\boldsymbol{S}_\ell^{(t)} \leftarrow \boldsymbol{S}_\ell^{(t-1)};$

      **end if**

      $\boldsymbol{R}_\ell^{(t)} \leftarrow \boldsymbol{S}_\ell^{(t)} \odot \boldsymbol{G}_\ell^{(t)};$

      $\boldsymbol{M}_\ell^{(t)} \leftarrow (1 - \beta_1) \boldsymbol{S}_\ell^{(t)} \odot \boldsymbol{M}_\ell^{(t-1)} + \beta_1 \boldsymbol{R}_\ell^{(t)};$

      $\boldsymbol{V}_\ell^{(t)} \leftarrow (1 - \beta_2) \boldsymbol{S}_\ell^{(t)} \odot \boldsymbol{V}_\ell^{(t-1)} + \beta_2 \boldsymbol{R}_\ell^{(t)} \odot \boldsymbol{R}_\ell^{(t)};$

      **if** using Adam **then**

        $\boldsymbol{M}_\ell^{(t)} \leftarrow \boldsymbol{M}_\ell^{(t)}/(1 - \beta_1^t), \quad \boldsymbol{V}_\ell^{(t)} \leftarrow \boldsymbol{V}_\ell^{(t)}/(1 - \beta_2^t), \quad \boldsymbol{N}_\ell^{(t)} \leftarrow \boldsymbol{M}_\ell^{(t)}/(\sqrt{\boldsymbol{V}_\ell^{(t)}} + \epsilon);$

      **else if** using MSGD **then**

        $\boldsymbol{N}_\ell^{(t)} \leftarrow \boldsymbol{M}_\ell^{(t)};$

      **end if**

      $\boldsymbol{X}_\ell^{(t+1)} \leftarrow \boldsymbol{X}_\ell^{(t)} - \eta \boldsymbol{S}_\ell^{(t)} \odot \boldsymbol{N}_\ell^{(t)};$

    **end for**

  **end for**

---

