# OpenReview forum: "Subspace Optimization for Large Language Models with Convergence Guarantees"
_ICML.cc/2025/Conference — ICML 2025 poster_

### Official Review · Reviewer_ZeQP · 2025-03-09

**Overall Recommendation:** 4

**Summary:**

The paper critically examines GaLore, highlighting its convergence limitations and proposing GoLore, a robust variant that ensures stochastic convergence. The findings contribute to improving memory-efficient subspace optimization methods for LLM training.

## update after rebuttal
My concerns were addressed in the rebuttal.

**Claims And Evidence:**

The authors claimed GaLore's limitation and provided theoretical justification.

**Essential References Not Discussed:**

Reference is good.

**Experimental Designs Or Analyses:**

I think the experiment is not enough. The improvement is minor.

**Methods And Evaluation Criteria:**

The experiments mainly focus on fine-tuning the GLUE benchmark using pre-trained RoBERTa-Base. In the appendix, they also considers fine-tuning LLaMa 2-7B.

**Other Comments Or Suggestions:**

No

**Other Strengths And Weaknesses:**

Weakness:

From my perspective, GaLore is mainly developed for the training of LLM. The current experiments only consider the fine-tuning, which may not be enough to support the effectiveness of this algorithm.

In the fine-tuning task, the improvement is minor, which weakens the contribution of this work.

The proposed algorithm still relies on GaLore in the initial stage in the practical implementation as mentioned by the authors.

Strengths:

This paper proposed an algorithm for LLM training with convergence guarantee, which enjoys better theoretical properties.

**Questions For Authors:**

It would be better to include important parameters and discussion on these parameters in the main text.

In your analysis, it seems that you didn't consider momentum and Adam cases. It would be better to clarify this.

In the conventional SGD algorithm, it relies on diminishing step size to eliminate the stationary gap. In your counter example and proof, can we reduce the stationary gap via reducing the step size? Is it an issue faced by all stochastic algorithms?

If your claim that SVD decomposition biases the gradient holds, why does the deterministic GaLore converge (Theorem 5)?

**Relation To Broader Scientific Literature:**

They show GaLore's limitation in theory, which is interesting to me.

**Theoretical Claims:**

First, the authors argue that GaLore suffers from convergence issues under general conditions and provide theoretical evidence. Subsequently, they proved GaLore's convergence when the batch size increases in the order of $\sqrt{T}$. Finally, they proposed their own algorithm, GoLore.

---

> ### Author Rebuttal · Authors · 2025-03-31
>
> We thank the reviewer for acknowledging our theoretical findings and for the valuable comments. All questions have been clarified as best as we can, and we are glad to address any further comments or questions. We include all new experiments in this **[anonymous link](https://www.hostize.com/v/pcWmN50BVa)**.
>
> **Weakness 1. GaLore is mainly developed for (pre-)training, current experiments only considering fine-tuning are not enough.**
>
> We would like to respectfully remind the reviewer that we have already included a pre-training result in Figure 4, where the algorithms are applied to pre-train a LLaMA-60M model on the C4 dataset for 30K iterations. To further address this concern, we have conducted additional experiments on pre-training LLaMA-350M models, with results provided in the anonymous link (Figure 1). These results show that transitioning from GaLore to GoLore leads to a faster decrease in the loss curve, further demonstrating the effectiveness of GoLore in pre-training tasks.
>
> **Weakness 2. The improvement in the fine-tuning task is minor.**
>
> We respectfully disagree with the reviewer’s assessment. Notably, in the GLUE benchmark, our method achieves an average score of **86.03**, compared to GaLore’s **85.77**. This represents a **threefold reduction** in the performance gap between GaLore and AdamW (86.15). We believe this result highlights the effectiveness of our method, demonstrating that a simple modification can significantly close the performance gap.
>
> **Weakness 3. The proposed algorithm still relies on GaLore in the initial stage in the practical implementation.**
>
> Our hybrid strategy, which relies on GaLore in the initial stage, aligns with our theoretical insights. The limitation of GaLore arises when anisotropic gradient noise starts to dominate the true gradient, which typically occurs **in the later stages of training**. In the early stage, when true gradients are dominant, GaLore’s greedy strategy remains effective, and no modification is necessary.
>
> Empirically, applying GoLore from the very beginning (i.e., GoLore@100%) results in performance similar to that of GaLore, likely due to a slower initial phase. Therefore, we believe the hybrid strategy is preferable, as it ensures both a fast initial convergence and higher accuracy in the later stages.
>
> **Question 1. It would be better to include important parameters and discussion on these parameters in the main text.**
>
> The primary hyperparameter introduced beyond GaLore is the **switching point**, where we transition to GoLore. We kindly refer the reviewer to our rebuttal to **Reviewer iLWQ (Weakness 1 and 2)** for a detailed discussion, as well as to the **anonymous link (Figure 2 and 3)** for ablation results.
>
> **Question 2. It seems that the analysis didn't consider momentum and Adam cases. It would be better to clarify this.**
>
> We kindly remind the reviewer that GoLore's **convergence analysis** is based on **momentum SGD**, which explicitly accounts for momentum. Additionally, in our **non-convergence analysis**, Theorem 4 holds for **any optimizer $\rho$**, including Adam, AdamW, and momentum SGD.
>
> However, we acknowledge that whether GoLore **converges under Adam** remains an open question. To clarify this, we will revise the conclusion section as follows:
>
> >**Original:** A limitation of this paper is that recent GaLore variants, such as Fira, are not readily covered by our analysis framework.
>
> >**Revised:** A limitation of this paper is that our convergence analysis framework has not readily covered the use of the Adam optimizer and recent GaLore variants such as Fira.
>
> **Question 3. In your counter example and proof, can we reduce the stationary gap via reducing the step size? Is it an issue faced by all stochastic algorithms?**
>
> We cannot reduce the gap by decreasing step size. As demonstrated in **Theorem 4**, the non-convergence result holds for **any optimizer $\rho$**, regardless of the algorithm framework or hyperparameters, including the step size. Therefore, the counterexample illustrates an issue that is inherent to **all stochastic algorithms**.
>
> **Question 4. If your claim that SVD decomposition biases the gradient holds, why does the deterministic GaLore converge (Theorem 5)?**
>
> It may be more accurate to say that GaLore's greedy SVD projection biases the **stochastic gradient**, including the **gradient noise**. Intuitively, the bias introduced by gradient noise will not vanish as the gradient noise itself will not vanish.
>
> On the other hand, the bias in the **true gradient** can be bounded, as shown in **Lemma 1**, and this bias vanishes as the true gradient approaches zero. This is why GaLore can still converge in the deterministic case, despite the bias in the gradient.
>
> We hope these responses can clarify the reviewer's questions and are more than happy to address any further comments or questions.

---

> > ### Comment · Reviewer_ZeQP · 2025-04-05
> >
> > Thanks for the clarification from the authors. I raised my score to 4.
> >
> > But I still have a concern:
> >
> > Could you clarify the advantage or the necessity, or the theoretical insight to update based on SVD decomposition in the early stage? Why is it better to follow this GaLore at the beginning of the training?

---

> > > ### Author Response · Authors · 2025-04-07
> > >
> > > Dear Reviewer ZeQP,
> > >
> > > Thank you again for acknowledging our previous response and for raising your score. We greatly appreciate your engagement with our work and are happy to provide further clarification regarding your question.
> > >
> > > Let $G=G_0+E$ denote the stochastic gradient, where $G_0$ is the true gradient and $E$ is the gradient noise, respectively. In the early stages of training, we typically have $\\|G_0\\|_F\gg\\|E\\|_F$, implying that the gradient is nearly noiseless. In this setting, GaLore's SVD-based projection provides an optimal low-rank approximation of $G$, whereas random projections do not exploit this structure.
> > >
> > > To elaborate, when $\\|G_0\\|_F\gg\\|E\\|_F$, we can approximate $G\approx G_0$. Let $G=U\Sigma V^\top$ be the SVD of $G$, where the singular values satisfy $\sigma_1\ge\sigma_2\ge\cdots\ge\sigma_m$ (assuming $G\in\mathbb{R}^{m\times n} \text{ with } m\le n$). According to the proof of Lemma 1 (Line 801), the GaLore projection matrix $P_a=U[:r]$ satisfies:$$\\|P_aP_a^\top G-G_0\\|_F^2\approx\\|P_aP_a^\top G-G\\|_F^2=\sum\_{i=r+1}^m\sigma_i^2.$$
> > >
> > > In contrast, for a random projection matrix $P_o\sim\mathcal{U}(\mathrm{St}_{m,r})$, Lemma 5 (Line 874) yields: $$\mathbb{E}[\\|P_oP_o^\top G-G_0\\|_F^2]\approx\mathbb{E}[\\|P_oP_o^\top G-G\\|_F^2]=\left(1-\frac{r}{m}\right)\\|G\\|_F^2=\frac{m-r}{m}\sum\_{i=1}^m\sigma_i^2.$$
> > >
> > > The gap between these two approximations is: $$\frac{m-r}{m}\sum_{i=1}^m\sigma_i^2-\sum_{i=r+1}^m\sigma_i^2=\frac{(m-r)r}{m}\cdot\left(\frac{1}{r}\sum_{i=1}^r\sigma_i^2-\frac{1}{m-r}\sum_{i=r+1}^m\sigma_i^2\right)\ge0,$$ with equality only when  $\sigma_1=\cdots=\sigma_m$. As observed in GaLore (Zhao et al., 2024), gradients in deep learning exhibit low-rank structure, meaning $\frac{1}{r}\sum_{i=1}^r\sigma_i^2$ is often significantly larger than $\frac{1}{m-r}\sum_{i=r+1}^m\sigma_i^2$, making GaLore particularly effective in early training when gradients are less noisy.
> > >
> > > We hope this explanation clarifies the theoretical advantage of using SVD-based projection in the early stages. Please let us know if you have further questions—we’d be happy to continue the discussion.
> > >
> > > **Best regards**,
> > >
> > > Authors of Submission 9646

---

### Official Review · Reviewer_Eita · 2025-03-14

**Overall Recommendation:** 3

**Summary:**

This paper examines the convergence properties of subspace optimization algorithms for LLM, focusing on GaLore. GaLore is known for memory efficiency in pre-training and fine-tuning LLM, this paper shows that it does not always converge under standard stochastic optimization settings. The authors substantiate this claim with a counterexample and explore conditions for its convergence, such as large mini-batch sizes or deterministic gradients. To address the limitations of GaLore, the paper introduces a new algorithm called GoLore (Gradient random Low-rank projection). Unlike GaLore's SVD-based projections, GoLore uses random projections, enabling convergence even with small mini-batches in stochastic settings. Theoretical analysis demonstrates GoLore's superior convergence properties under general conditions.

## Update after Rebuttal
As mentioned earlier, I maintained a positive score, and I hope the next revision will incorporate the discussed changes.

**Claims And Evidence:**

The claims in the paper regarding the non-convergence of GaLore and the convergence guarantees of GoLore is supported by a combination of theoretical analysis, illustrative counterexamples and empirical evaluation.

**Essential References Not Discussed:**

References seems to be well cited

**Experimental Designs Or Analyses:**

The author provides a counter example for convergence guarantee, and they validate this with experiments in Figure 1.

**Methods And Evaluation Criteria:**

Yes it does.

**Other Comments Or Suggestions:**

As mentioned before that paper has been well presented and there are no major typos or inconsistencies there.

**Other Strengths And Weaknesses:**

Positives: The paper supports their claim with rigorous theoretical analysis. The paper is well presented.

Negatives: The dependence of the assumptions may limit broader applicability. In a noisy setup, a random projection may add more uncertainty in the final claims.

**Questions For Authors:**

Can a sampling based row rank approximation be used instead of random projection of GOLORE or singular value decomposition of GALORE?

**Relation To Broader Scientific Literature:**

The paper addresses the issue of memory efficient training for LLM along with their guaranteed convergence.

**Theoretical Claims:**

I checked a few convergence theorems, such as theorem 10. 12 and 18, they look fine.

---

> ### Author Rebuttal · Authors · 2025-03-31
>
> We thank the reviewer for acknowledging our theoretical results and appreciate the efforts made to check our convergence proofs. All questions have been clarified as best as we can, and we are glad to address any further comments or questions. We include all new experiments in this **[anonymous link](https://www.hostize.com/v/pcWmN50BVa)**.
>
> **Weakness 1. The dependence of the assumptions may limit broader applicability. In a noisy setup, a random projection may add more uncertainty in the final claims.**
>
> We respectfully disagree with the reviewer’s concern that GoLore's broader applicability is limited by the dependence on assumptions, particularly regarding the noise scale. While introducing random projection may add more uncertainty, we argue that such randomness is, in fact, essential to mitigate the impact of gradient noise in noisy settings.
>
> To see it, we first consider the deterministic projection used by GaLore. Let $G=G_0+E$ represent the stochastic gradient, where $G_0$ is the true gradient and $E$ is the gradient noise. In noisy settings where $\\|E\\|\gg\\|G_0\\|$, GaLore deterministically select the projection matrix $P$ using the information of $G\approx E$. The noisier $E$ is, the noisier $P$ becomes. Moreover, GaLore's Top-K selection can degrade the unbiasedness of $E$, potentially leading to non-convergence.
>
> In contrast, GoLore randomly samples $P$ from the same distribution, regardless of how noisy $E$ is. This approach shields $P$ from being influenced by the dominant gradient noise, ensuring its stability regardless of the noise level in $E$. Furthermore, GoLore's random projection maintains the unbiasedness in subspace selection, as shown by $\mathbb{E}[PP^\top]=\frac{r}{m}\cdot I$ in Lemma 5.
>
> Additionally, the assumptions regarding gradient noise in our analysis are standard in stochastic optimization and are widely accepted in convergence studies. Therefore, we believe that our theoretical results are not unduly restricted by these assumptions and that random projection enhances the algorithm’s robustness in noisy settings.
>
> **Question 1. Can a sampling based row rank approximation be used instead of random projection of GoLore or singular value decomposition of GaLore?**
>
> Thank you for the interesting question. To explore this, we have additionally evaluated the performance of a sampling-based method as follows:
>
> Let $G\in\mathbb{R}^{m\times n}\ (m\le n)$ be the stochastic gradient matrix. Similar to GaLore, we first compute SVD of $G$, obtaining $G=U\Sigma V^\top$. However, instead of selecting the first $r$ columns of $U$ as GaLore does, we  sample $r$ columns with probabilities proportional to the corresponding singular values $\sigma_i$'s. As shown in Figure 1 in our anonymous link, the performance of this sampling-based strategy is similar to that of GaLore's, outperformed by our GoLore method. This is potentially because such an importance sampling strategy cannot lead to unbiased projection matrices, and still suffer from the bias induced by gradient noise.
>
> We thank the reviewer again for the careful comments and valuable suggestions. We hope these responses can clarify the reviewer's questions and are more than happy to address any further comments or questions.

---

> > ### Comment · Reviewer_Eita · 2025-04-03
> >
> > I appreciate the detailed response to the queries/concerns by the authors. I maintain my overall positive score.

---

> > > ### Author Response · Authors · 2025-04-03
> > >
> > > Dear Reviewer Eita,
> > >
> > > Thank you for your time and consideration. We appreciate your thoughtful review and your positive assessment of our work.
> > >
> > > Authors of Submission 9646

---

### Official Review · Reviewer_iLWQ · 2025-03-17

**Overall Recommendation:** 3

**Summary:**

The paper shows that the subspace projection of Galore can be biased when approaching to local minimizer, where the principle component of the projection matrix mainly capture the information of the stochastic noises. Built upon this insight, the paper proposes a method named Golore, which samples the projection matrix from stiefel manifold for ensuring the unbiasness of the gradient when the training is about to converge, i.e. close to local minimizer.

**Claims And Evidence:**

Yes. The claims are supported by the proof and numerical results.

**Essential References Not Discussed:**

N/A

**Experimental Designs Or Analyses:**

Yes.

**Methods And Evaluation Criteria:**

Yes, the C4 pretraining task and GLUE finetuning task are standard benchmarks for evaluating memory-efficient optimization algorithms.

**Other Comments Or Suggestions:**

It may be informative to plot the norm of gradient bias of both Galore and Golore (may be conducted under a relatively small dataset for cheap evaluation of full gradient).

**Other Strengths And Weaknesses:**

**Strengths**

* The paper reveals that Galore's slow convergence in the late training stage is due to that priciple components are dominated by the stochastic noise. The results are justified by (non)-convergence results.

* The proposed strategy for avoiding biasness is relatively easy to implement. Experiments on LLM's pre-training demonstrates that adding Golore phase exhibits faster convergence than running Galore alone.

**Weaknesses**

* The choice of switching point to Golore may have huge impact on the convergence. However, the paper does not provide guideline on how to choose this hyperparameter. The ablation study on the subspace update frequency $\tau$ is missing as well.

**Questions For Authors:**

N/A

**Relation To Broader Scientific Literature:**

The paper explains why Galore can be much slower than Adam in the late training stage from gradient bias perspective. Both theoretical researcher and practioners may find insight from this paper.

**Theoretical Claims:**

I have not went through all the proofs, but I am farmiliar with the related techniques. The theorem make sense under the given assumptions.

---

> ### Author Rebuttal · Authors · 2025-03-31
>
> We thank the reviewer for acknowledging our theoretical results and experimental designs, as well as the detailed comments and suggestions. All questions have been clarified as best as we can, and we are glad to address any further comments or questions. We include all new experiments in this **[anonymous link](https://www.hostize.com/v/pcWmN50BVa)**.
>
> - **Weakness 1. The paper does not provide a guideline on how to choose the switching point.**
>
>     We appreciate the reviewer’s valuable feedback. In our experiments, we manually selected the switching points from $\\{20\\%,30\\%,40\\%,50\\%\\}$ based on intuition and prior experience. Specifically, we choose an earlier switching point if we expect the algorithm to converge more quickly to the solution's neighborhood and a later one if we anticipate slower convergence. To provide a broader guideline, we offer the following insights:
>
>     - If access to the true gradient is available, a reasonable switching point would be when gradient noise begins to dominate the true gradient.
>
>     - If only stochastic gradients are available, and assuming the gradient noise has a roughly constant scale, this dominance can be estimated by monitoring whether the norm of the stochastic gradients falls below a certain threshold. This threshold serves as a hyperparameter that depends on both the training task and the batch size used in the algorithm.
>
>     - The optimal switching point is task-dependent. Empirically, it is recommended to switch when the rate of decrease in the loss curve starts to slow down.
>
> - **Weakness 2. The ablation study on the subspace update frequency is missing.**
>
>     We appreciate the reviewer’s suggestion. We did not initially include an ablation study on the subspace update frequency $\tau$ because the original GaLore paper (Zhao et al., 2024) states that values of $\tau$ between 50 and 1000 have minimal impact on performance. Based on this, we set $\tau=500$ to:
>     - Reduce computational overhead, and
>     - Ensure a sufficient number of GoLore projection steps to enhance performance.
>
>     However, in response to the reviewer's concern, we have provided additional ablation results in Figure 3 in the anonymous link. These results confirm that varying $\tau$ within the range of 50 to 1000 has negligible effect on performance.
>
> - **Suggestion 1. It may be informative to plot the norm of gradient bias of both Galore and Golore.**
>
>    We appreciate the reviewer’s valuable suggestion. In response, Figures 4 and 5 in the anonymous link have additionally plotted the gradient bias norms in our experiments, demonstrating that GoLore achieves a smaller gradient bias.
>
> We thank the reviewer again for the careful comments. We hope these responses can clarify the reviewer's questions and are more than happy to address any further comments or questions.

---

### Decision · Program_Chairs · 2025-05-01

**Decision:**

Accept (poster)

**Comment:**

The paper studies the convergence of GaLore close to the local minima. The key insight is that at these later stages (closer to the local minima) the noise dominates and as a result GaLore may not converge, leading to poor performance. Based on this insight, the paper proposes a randomized projection strategy at the later stages.

**Strengths**: While the gains are modest, this insight is fairly general and interesting and can be used by methods that leverage the low-rank property of the gradient. The paper is overall well-written, and the key insight is corroborated by empirical evaluations.

**Weakness**: While the reviewers appreciated the theoretical analysis. Some parts can be made clearer to improve readability. For instance, in Thm. 4, the paper can be explicit regarding the significance of $\|\nabla f(\mathbf{x}^{(t)}\| \geq \epsilon_0$. That is, the paper can state that since the gradient does not go to zero, this shows nonconvergence of GaLore. This also goes to the proofs, where providing context for each step can further improve the paper and maximize its impact.

The AC believes that these are minor issues which can be handled while preparing the final version of the paper. As a result, the AC recommends acceptance.